# Therapeutic genetic variation revealed in diverse Hsp104 homologs

Zachary M March[1,2], Katelyn Sweeney[3,4,5†], Hanna Kim[6†], Xiaohui Yan[6†], Laura M Castellano[1,7], Meredith E Jackrel[1‡], JiaBei Lin[1], Edward Chuang[1,7], Edward Gomes[1], Corey W Willicott[6], Karolina Michalska[8,9], Robert P Jedrzejczak[8], Andrzej Joachimiak[8,9], Kim A Caldwell[6], Guy A Caldwell[6], Ophir Shalem[3,4,5], James Shorter[1,2,4,7*]

[1]Department of Biochemistry and Biophysics, Perelman School of Medicine at the University of Pennsylvania, Philadelphia, United States; [2]Department of Biochemistry and Molecular Biophysics Graduate Group, Perelman School of Medicine at the University of Pennsylvania, Philadelphia, United States; [3]Department of Genetics, Perelman School of Medicine at the University of Pennsylvania, Philadelphia, United States; [4]Cell and Molecular Biology Graduate Group, Perelman School of Medicine at the University of Pennsylvania, Philadelphia, United States; [5]Center for Cellular and Molecular Therapeutics, Children's Hospital of Philadelphia, Philadelphia, United States; [6]Department of Biological Sciences, The University of Alabama, Tuscaloosa, United States; [7]Pharmacology Graduate Group, Perelman School of Medicine at the University of Pennsylvania, Philadelphia, United States; [8]Structural Biology Center, X-ray Science Division, Argonne National Laboratory, Argonne, United States; [9]Department of Biochemistry and Molecular Biology, University of Chicago, Chicago, United States

*For correspondence:
jshorter@pennmedicine.upenn.edu

†These authors contributed equally to this work

Present address: ‡Department of Chemistry, Washington University in St. Louis, St. Louis, United States

**Abstract** The AAA+ protein disaggregase, Hsp104, increases fitness under stress by reversing stress-induced protein aggregation. Natural Hsp104 variants might exist with enhanced, selective activity against neurodegenerative disease substrates. However, natural Hsp104 variation remains largely unexplored. Here, we screened a cross-kingdom collection of Hsp104 homologs in yeast proteotoxicity models. Prokaryotic ClpG reduced TDP-43, FUS, and α-synuclein toxicity, whereas prokaryotic ClpB and hyperactive variants were ineffective. We uncovered therapeutic genetic variation among eukaryotic Hsp104 homologs that specifically antagonized TDP-43 condensation and toxicity in yeast and TDP-43 aggregation in human cells. We also uncovered distinct eukaryotic Hsp104 homologs that selectively antagonized α-synuclein condensation and toxicity in yeast and dopaminergic neurodegeneration in *C. elegans*. Surprisingly, this therapeutic variation did not manifest as enhanced disaggregase activity, but rather as increased passive inhibition of aggregation of specific substrates. By exploring natural tuning of this passive Hsp104 activity, we elucidated enhanced, substrate-specific agents that counter proteotoxicity underlying neurodegeneration.

## Introduction

Alternative protein folding and aberrant phase transitions underpin fatal neurodegenerative diseases (*Chuang et al., 2018*; *Mathieu et al., 2020*). Diseases such as Parkinson's disease (PD) and amyotrophic lateral sclerosis (ALS) have distinct clinical manifestations but are united by the prominent pathological accumulation of misfolded protein conformers (*Peng et al., 2020*). The proteins implicated in each disease can adopt a range of misfolded conformations (*Peng et al., 2020*; *Shorter, 2017*).

Thus, in PD, alpha-synuclein (αSyn) accumulates in toxic soluble oligomers and amyloid fibers that are a major component of cytoplasmic Lewy bodies in degenerating dopaminergic neurons (*Gallegos et al., 2015*; *Gitler and Shorter, 2007*; *Kim et al., 2014b*; *Recasens and Dehay, 2014*; *Snead and Eliezer, 2014*). Likewise, in ALS, the normally nuclear RNA-binding proteins, TDP-43 and FUS, accumulate in toxic oligomeric structures and cytoplasmic inclusions in different forms of disease (*Gitler and Shorter, 2011*; *Johnson et al., 2009*; *Ling et al., 2013*; *March et al., 2016*; *Sun et al., 2011*).

Protein disaggregation represents an innovative and appealing therapeutic strategy for the treatment of protein-misfolding disorders in that it simultaneously reverses: (a) loss-of-function phenotypes associated with sequestration of functional soluble protein into misfolded oligomers and insoluble aggregates; and (b) any toxic gain-of-function phenotypes associated with the misfolded conformers themselves (*March et al., 2019*; *Shorter, 2008*; *Vashist et al., 2010*). The AAA+ (ATPases Associated with diverse Activities) protein Hsp104 from *Saccharomyces cerevisiae* (ScHsp104) rapidly disassembles a diverse range of misfolded protein conformers, including amorphous aggregates, preamyloid oligomers, and amyloid fibers (*DeSantis et al., 2012*; *DeSantis and Shorter, 2012b*; *Glover and Lindquist, 1998*; *Liu et al., 2011*; *Lo Bianco et al., 2008*; *Shorter and Lindquist, 2004*; *Shorter and Lindquist, 2006*; *Shorter and Southworth, 2019*; *Sweeny and Shorter, 2016*). Hsp104 assembles into asymmetric ring-shaped hexamers that undergo conformational changes upon ATP binding and hydrolysis, which drive substrate translocation across the central channel to power protein disaggregation (*DeSantis et al., 2012*; *Gates et al., 2017*; *Michalska et al., 2019*; *Shorter and Southworth, 2019*; *Sweeny and Shorter, 2016*; *Ye et al., 2019*; *Yokom et al., 2016*). Each protomer is comprised of an N-terminal domain (NTD), nucleotide-binding domain 1 (NBD1), a middle domain (MD), NBD2, and a short C-terminal domain (*Shorter and Southworth, 2019*; *Sweeny and Shorter, 2016*). Hsp104 can disassemble preamyloid oligomers and amyloid conformers of several proteins associated with neurodegenerative disease, including αSyn, polyglutamine, amyloid-beta, and tau (*DeSantis et al., 2012*; *Lo Bianco et al., 2008*). Moreover, these protein-remodeling events result in neuroprotection. For example, Hsp104 suppresses protein-misfolding-induced neurodegeneration in rat and *D. melanogaster* models of polyglutamine-expansion disorders (*Cushman-Nick et al., 2013*; *Vacher et al., 2005*), and a rat model of Parkinson's disease (*Lo Bianco et al., 2008*). Hsp104 is the only factor known to eliminate αSyn fibers and oligomers in vitro, and prevent αSyn-mediated dopaminergic neurodegeneration in rats (*Lo Bianco et al., 2008*). However, these activities have limits and high concentrations of Hsp104 can be required for modest levels of protein remodeling (*DeSantis et al., 2012*; *Jackrel et al., 2014*; *Lo Bianco et al., 2008*).

Previously, we circumvented limitations on Hsp104 disaggregase activity by developing a suite of potentiated Hsp104 variants, differing from wild-type (WT) Hsp104 by one or more missense mutations in the autoregulatory MD (*Jackrel and Shorter, 2015*). These potentiated Hsp104 variants antagonize proteotoxic misfolding of several disease-linked proteins, including TDP-43, FUS, TAF15, FUS-CHOP, EWS-FLI, polyglutamine, and αSyn (*Jackrel et al., 2014*; *Jackrel and Shorter, 2014*; *Jackrel et al., 2015*; *Michalska et al., 2019*; *Ryan et al., 2019*; *Sweeny et al., 2020*; *Tariq et al., 2018*; *Yasuda et al., 2017*). However, in some circumstances, these Hsp104 variants can also exhibit off-target toxicity (*Jackrel et al., 2014*; *Jackrel and Shorter, 2014*; *Jackrel et al., 2015*). Thus, uncovering other therapeutic Hsp104s with diminished propensity for off-target effects is a key objective (*Mack et al., 2020*; *Tariq et al., 2019*).

Hsp104 is conserved among all non-metazoan eukaryotes and eubacteria, and is also found in some archaebacteria (*Erives and Fassler, 2015*; *Sweeny and Shorter, 2016*). We have found that prokaryotic Hsp104 homologs exhibit reduced amyloid-remodeling activity compared to eukaryotic homologs (*Castellano et al., 2015*; *DeSantis et al., 2012*; *Michalska et al., 2019*; *Shorter and Lindquist, 2004*; *Sweeny and Shorter, 2016*). Nonetheless, natural Hsp104 sequence space remains largely unexplored. This lack of exploration raises the possibility that natural Hsp104 sequences may exist with divergent enhanced and selective activity against neurodegenerative disease substrates. Indeed, we reported that an Hsp104 homolog from the thermophilic fungus *Calcarisporiella thermophila* antagonizes toxicity of TDP-43, αSyn, and polyglutamine in yeast without apparent toxic off-target effects (*Michalska et al., 2019*). These findings support our hypothesis that natural Hsp104 homologs may have therapeutically beneficial properties.

Here, we survey a cross-kingdom collection of Hsp104 homologs from diverse lineages for their ability to suppress proteotoxicity from several proteins implicated in human neurodegenerative disease. We discovered that prokaryotic ClpB and hyperactive variants were ineffective, but prokaryotic ClpG could mitigate TDP-43, FUS, and α-synuclein toxicity. Several eukaryotic Hsp104 homologs emerged that selectively suppressed TDP-43 or αSyn toxicity. Mechanistic studies and mutational analysis suggest that these selective activities are not due to enhanced disaggregase activity. Rather, they are due to genetic variation that impacts a passive, aggregation-inhibition activity of Hsp104 homologs for select substrates. Thus, we establish that manipulating passive aggregation-inhibition activity of Hsp104 represents a novel route to enhanced, substrate-specific agents able to counter the deleterious protein misfolding that underlies neurodegenerative disease.

## Results

### Diverse Hsp104 homologs selectively suppress TDP-43 toxicity and aggregation in yeast

In yeast, galactose-inducible expression of several proteins associated with neurodegenerative diseases, including αSyn (*Outeiro and Lindquist, 2003*), TDP-43 (*Johnson et al., 2008*), and FUS (*Sun et al., 2011*) results in their cytoplasmic aggregation and toxicity. These yeast models have proven to be powerful platforms that have enabled the discovery of several important genetic and small-molecule modifiers of disease protein aggregation and toxicity. Importantly, these modifiers have translated to more complex model systems including worm, fly, mouse, and neuronal models of neurodegenerative disease (*Armakola et al., 2012*; *Barmada et al., 2015*; *Becker et al., 2017*; *Caraveo et al., 2014*; *Caraveo et al., 2017*; *Chung et al., 2013*; *Cooper et al., 2006*; *Dhungel et al., 2015*; *Elden et al., 2010*; *Fanning et al., 2019*; *Gitler et al., 2009*; *Jackrel et al., 2014*; *Jackson et al., 2015*; *Ju et al., 2011*; *Khurana et al., 2017*; *Kim et al., 2014a*; *Nuber et al., 2020*; *Su et al., 2010*; *Tardiff et al., 2013*; *Tardiff et al., 2012*; *Vincent et al., 2018*). Indeed, potential therapeutics for ALS (e.g. ataxin 2 antisense oligonucleotide, ION541/BIIB105) and PD (e.g. stearoyl-CoA-desaturase inhibitor, YTX-7739) that have emerged from these studies are now entering phase 1 clinical trials. Coexpression of potentiated Hsp104 variants mitigates αSyn, TDP-43, and FUS toxicity in yeast (*Jackrel et al., 2014*; *Jackrel et al., 2015*; *Tariq et al., 2019*; *Tariq et al., 2018*). Moreover, the natural homolog *Calcarisporiella thermophila* Hsp104 (CtHsp104), but not ScHsp104, reduces TDP-43 and αSyn toxicity in yeast (*Michalska et al., 2019*).

For a deeper exploration of natural Hsp104 sequence space, we screened a collection of 15 additional Hsp104 homologs from diverse eukaryotes spanning ~2 billion years of evolution and encompassing fungi (*Thielavia terrestris, Thermomyces lanuginosus, Dictyostelium discoideum, Chaetomium thermophilum, Lachancea thermotolerans, Myceliophthora thermophila, Scytalidium thermophilum,* and *Thermoascus aurantiacus*), plants (*Arabidopsis thaliana* and *Populus euphratica*), protozoa (*Monosiga brevicollis, Salpingoeca rosetta,* and *Plasmodium falciparum*), and chromista (*Chlamydomonas reinhardtii* and *Galdieria sulphuraria*) (*Figure 1A*; see *Supplementary file 1* for homolog sequences, *Figure 1—figure supplement 1* for an alignment of all homologs). All Hsp104 homologs, except the homolog from *Plasmodium falciparum (Pf)*, were non-toxic to yeast at 30°C or 37°C (*Figure 1—figure supplement 2*). PfHsp104 was even more toxic than the potentiated Hsp104 variant, Hsp104$^{A503V}$, at 37°C (*Figure 1—figure supplement 2*). This toxicity might reflect the very different role played by PfHsp104 in its host where it is not a soluble protein disaggregase. Rather, PfHsp104 has been repurposed as a key component of a membrane-embedded translocon, which transports malarial proteins across a parasite-encasing vacuolar membrane into erythrocytes (*Bullen et al., 2012*; *de Koning-Ward et al., 2009*; *Ho et al., 2018*).

We screened the Hsp104 homologs for suppression of TDP-43, FUS, and αSyn proteotoxicity (*Figures 1B,C* and *2A,B*, and *Figure 1—figure supplement 3*). The toxic Hsp104 homolog, PfHsp104, was unable to reduce TDP-43, FUS, and αSyn proteotoxicity (*Figure 1—figure supplement 3C*). By contrast, in addition to CtHsp104, which suppresses TDP-43 and αSyn toxicity (*Figures 1B* and *2A*), five of these eukaryotic homologs (from protozoa: *Monosiga brevicollis (Mb)* and *Salpingoeca rosetta (Sr)*, from chromista: *Chlamydomonas reinhardtii (Cr)* and *Galdieria sulphuraria (Gs)*, and the plant *Populus euphratica (Pe)*), suppress TDP-43 toxicity (*Figure 1C*; see *Figure 1—figure supplement 4* for an alignment comparing TDP-43-rescuing homologs to ScHsp104).

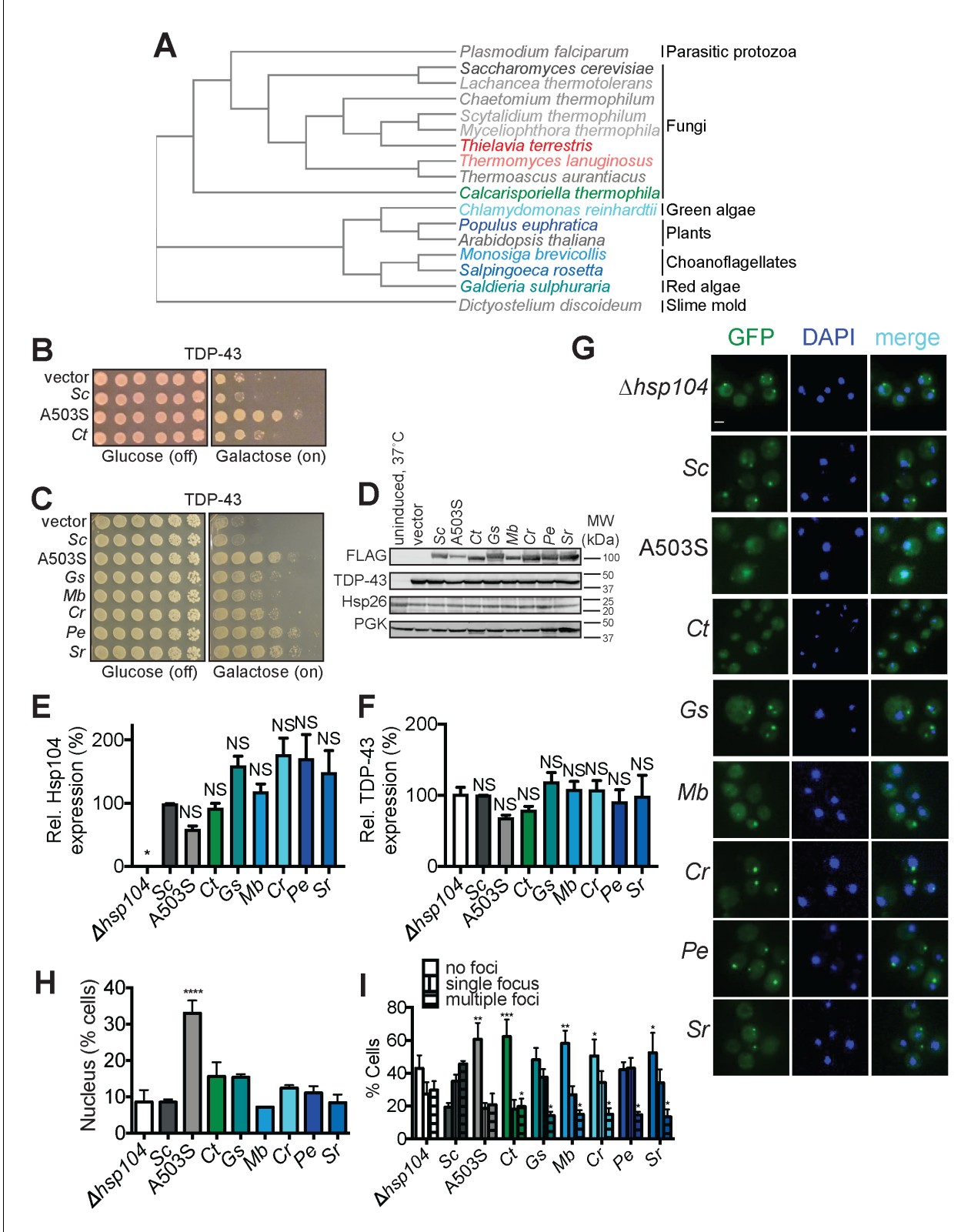

**Figure 1.** Diverse Hsp104 homologs suppress TDP-43 toxicity in yeast. (**A**) Phylogenetic tree constructed using EMBL-EBI Simple Phylogeny tool from a multiple sequence alignment of the indicated Hsp104 homologs generated in Clustal Omega (see also Supplemental Information for alignments) showing evolutionary relationships between Hsp104 homologs studied in this paper. *C. thermophila* is in green, TDP-43-specific homologs are colored in shades of blue, αSyn-specific homologs are colored in red, and non-rescuing homologs are colored in shades of gray. (**B**) Δ*hsp104* yeast transformed

*Figure 1 continued on next page*

*Figure 1 continued*

with plasmids encoding galactose-inducible TDP-43 and the indicated galactose-inducible Hsp104 (either wild-type Hsp104 from *Saccharomyces cerevisiae*, the potentiated variant A503S, or the Hsp104 homolog from *Calcarisporiella thermophila* (Ct)) were serially diluted 5-fold and spotted onto glucose (expression off) or galactose (expression on). (C) Δ*hsp104* yeast transformed with plasmids encoding galactose-inducible TDP-43 and the indicated galactose-inducible Hsp104 (either wild-type Hsp104 from *S. cerevisiae*, the potentiated variant A503S, or homologs from *Galdieria sulphuraria* (Gs), *Monosiga brevicollis* (Mb), *Chlamydomonas reinhardtii* (Cr), *Populus euphratica* (Pe), and *Salpingoeca rosetta* (Sr)) were serially diluted 5-fold and spotted onto glucose (expression off) or galactose (expression on). (D) Western blots confirm consistent expression of FLAG-tagged Hsp104s and proteotoxic protein substrates in yeast, and that neither Hsp104 expression nor TDP-43 expression induces upregulation of the endogenous heat-shock protein Hsp26. The first lane are isogenic yeast that have not been grown in galactose to induce Hsp104 and TDP-43 expression but instead have been pretreated at 37°C for 30 min to upregulate endogenous heat-shock proteins. 3-phosphoglycerate kinase (PGK) is used as a loading control. Molecular weight markers are indicated (right). (E) Expression of the indicated Hsp104-FLAG relative to PGK was quantified for each strain. Values are means ± SEM (n = 3). One-way ANOVA with Dunnett's multiple comparisons test was used to compare expression of ScHsp104-FLAG (*Sc*) to all other conditions. *p<0.05; NS, not significant. (F) TDP-43 expression relative to PGK was quantified for each strain. Values are means ± SEM (n = 3). One-way ANOVA with Dunnett's multiple comparisons test was used to compare TDP-43 levels in the Δ*hsp104* control to all other conditions. NS, not significant. (G) Representative images of yeast co-expressing TDP-43-GFPS11 (and separately GFPS1-10 to promote GFP reassembly) and the indicated Hsp104 homologs. Cells were stained with DAPI to visualize nuclei (blue). Scale bar, 2.5 μm. (H) Quantification of cells where TDP-43 displays nuclear localization. Values represent means ± SEM (n = 3 trials with >200 cells counted per trial). One-way ANOVA with Dunnett's multiple comparisons test was used to compare Δ*hsp104* to all other conditions. ****p<0.0001. (I) Quantification of cells with no, single, or multiple TDP-43 foci. Values represent means ± SEM (n = 3). Two-way ANOVA with Tukey's multiple comparisons test was used to compare the proportion of cells with either no or multiple TDP-43 foci between strains expressing different Hsp104 homologs and a control strain expressing ScHsp104. *p<0.05; **p<0.01; ***p<0.001.

The online version of this article includes the following figure supplement(s) for figure 1:

**Figure supplement 1.** Alignment of all Hsp104 homologs investigated in this study.
**Figure supplement 2.** Hsp104 homologs do not typically induce temperature-dependent toxicity.
**Figure supplement 3.** Suppression of TDP-43 or αSyn toxicity by select Hsp104 homologs is a substrate-specific effect.
**Figure supplement 4.** Alignment comparing ScHsp104 to Hsp104 homologs that rescue TDP-43 toxicity.

Interestingly, the Hsp104 homologs that suppress TDP-43 toxicity have minimal effect on αSyn and FUS toxicity (*Figure 1—figure supplement 3A*). Thus, we describe the first natural or engineered Hsp104 variants that diminish TDP-43 toxicity in a substrate-specific manner.

Expression of Hsp104 homologs did not vary significantly, nor did they significantly affect TDP-43 levels (*Figure 1D–F*), indicating that suppression of toxicity was not due to reduced TDP-43 expression. Moreover, levels of the small heat-shock protein, Hsp26, were similar in all strains and lower than in control strains that had been pretreated at 37°C to mount a heat-shock response (HSR) (*Cashikar et al., 2005*). Thus, expression of heterologous Hsp104s does not indirectly suppress TDP-43 toxicity by inducing a yeast HSR (*Figure 1D*).

Next, we examined how these Hsp104 homologs affect TDP-43 localization in our yeast model. In human cells, TDP-43 normally shuttles between the nucleus and cytoplasm, but in ALS, TDP-43 becomes mislocalized to cytoplasmic inclusions (*Neumann et al., 2006*). Expression of GFP-tagged TDP-43 (TDP43-GFP) in yeast recapitulates this phenotype (*Figure 1G,H*; *Johnson et al., 2008*; *Johnson et al., 2009*). Consistent with previous observations, expression of ScHsp104 does not affect TDP43-GFP cytoplasmic localization (*Figure 1H*) but slightly exacerbates formation of TDP43-GFP foci (*Figure 1I*). By contrast, the potentiated variant, Hsp104[A503S], restores nuclear TDP-43 localization in ~40% of cells (*Jackrel et al., 2014*; *Jackrel and Shorter, 2014*; *Figure 1H*) and suppresses TDP-43 foci formation (*Figure 1I*). Cells expressing Hsp104 homologs that suppress TDP-43 toxicity (e.g. *Ct*, *Gs*, *Mb*, *Cr*, *Pe*, and *Sr*), show at most a modest increase in nuclear TDP43-GFP compared to control strains lacking Hsp104 (Δ*hsp104*) or expressing ScHsp104 (*Figure 1H*). However, formation of cytoplasmic TDP43-GFP foci is suppressed by all these homologs (*Figure 1I*). Thus, TDP-43 toxicity can be mitigated in yeast by limiting TDP-43 inclusion formation without restoring nuclear localization. Indeed, PeHsp104 and SrHsp104 reduced TDP-43 toxicity as effectively as Hsp104[A503S], but without restoring TDP-43 to the nucleus (*Figure 1C,H*).

## Distinct Hsp104 homologs selectively suppress αSyn toxicity and inclusion formation in yeast

In addition to the five Hsp104 homologs that suppress TDP-43 toxicity, we discovered two new Hsp104 homologs (from *Thielavia terrestris* (Tt) and *Thermomyces lanuginosus* (Tl)) that suppress αSyn toxicity (*Figure 2B*; see *Figure 2—figure supplement 1* for an alignment comparing αSyn-

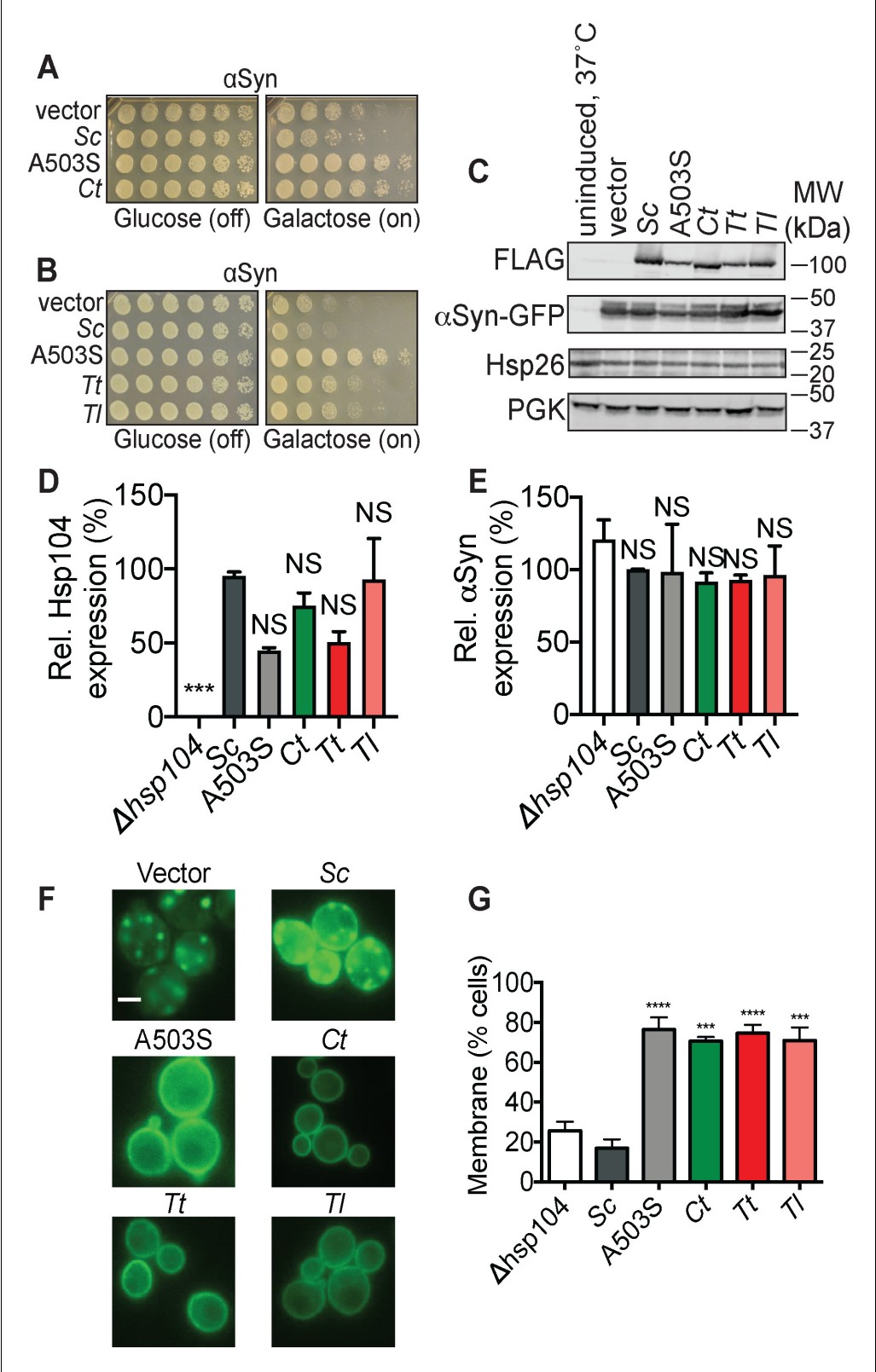

**Figure 2.** Hsp104 homologs from thermophilic fungi suppress αSyn toxicity in yeast. (**A**) Δ*hsp104* yeast transformed with plasmids encoding galactose-inducible αSyn and the indicated galactose-inducible Hsp104 (either wild-type Hsp104 from *Saccharomyces cerevisiae*, the potentiated variant A503S, or the Hsp104 homolog from *Calcarisporiella thermophila* (Ct)) were serially diluted 5-fold and spotted onto glucose (expression off) or galactose (expression on). (**B**) Δ*hsp104* yeast transformed with plasmids encoding galactose-inducible αSyn and the indicated galactose-inducible Hsp104 (either
*Figure 2 continued on next page*

Figure 2 continued

wild-type Hsp104 from *S. cerevisiae*, the potentiated variant A503S, or homologs from *Thielavia terrestris* (Tt), and *Thermomyces lanuginosus* (Tl)) were serially diluted 5-fold and spotted onto glucose (expression off) or galactose (expression on). (C) Western blots confirm consistent expression of FLAG-tagged Hsp104s and proteotoxic protein substrates in yeast, and that neither Hsp104 expression nor αSyn-GFP expression induces upregulation of the endogenous heat-shock protein Hsp26. The first lane are isogenic yeast that have not been grown in galactose to induce Hsp104 and αSyn expression but instead have been pretreated at 37°C for 30 min to upregulate heat-shock proteins. PGK is used as a loading control. Molecular weight markers are indicated (right). (D) Expression of the indicated Hsp104-FLAG relative to PGK was quantified for each strain. Values are means ± SEM (n = 3). One-way ANOVA with Dunnett's multiple comparisons test was used to compare expression of ScHsp104-FLAG (*Sc*) to all other conditions. ***p<0.001; NS, not significant. (E) αSyn-GFP expression relative to PGK was quantified for each strain. Values are means ± SEM (n = 3). One-way ANOVA with Dunnett's multiple comparisons test was used to compare αSyn-GFP levels the Δ*hsp104* control to all other conditions. NS, not significant. (F–G) Fluorescence microscopy of cells coexpressing αSyn-GFP and vector, ScHsp104$^{WT}$, ScHsp104$^{A503S}$, CtHsp104, TtHsp104, or TlHsp104. Scale bar, 2.5 μm. αSyn localization was quantified as the number of cells with fluorescence at the plasma membrane or cytoplasmic inclusions. Values are means ± SEM (n = 3 trials with >200 cells counted per trial). One-way ANOVA with Dunnett's multiple comparisons test was used to compare Δ*hsp104* to all other conditions. ***p<0.001; ****p<0.0001.

The online version of this article includes the following figure supplement(s) for figure 2:

**Figure supplement 1.** Alignment comparing ScHsp104 to Hsp104 homologs that rescue αSyn toxicity.

**Figure supplement 2.** Alignment comparing Hsp104 homologs that rescue TDP-43 toxicity to those that rescue αSyn toxicity.

**Figure supplement 3.** ClpG$_{GI}$ robustly suppresses αSyn toxicity and confers thermotolerance to Δ*hsp104* yeast, whereas ClpB and hyperactive variants do not.

rescuing homologs to ScHsp104 and *Figure 2—figure supplement 2* for an alignment comparing TDP-43-rescuing homologs to αSyn-rescuing homologs). Similar to the Hsp104 homologs that suppress TDP-43 toxicity, these Hsp104 homologs were selective and suppressed αSyn toxicity but had minimal effect on TDP-43 or FUS toxicity (*Figure 1—figure supplement 3B*). Eight of the fifteen Hsp104 homologs tested do not suppress TDP-43, αSyn, or FUS toxicity (*Figure 1—figure supplement 3C*). Expression of Hsp104 homologs did not vary significantly (*Figure 2C,D*). Hsp104 homologs suppressed αSyn toxicity without significantly affecting αSyn expression and without inducing an HSR as indicated by Hsp26 levels (*Figure 2C,E*).

We also examined how Hsp104 homologs that suppress αSyn toxicity affect αSyn localization in yeast. αSyn is a lipid-binding protein that mislocalizes to cytoplasmic Lewy bodies in Parkinson's disease (*Spillantini et al., 1997*). Overexpression of αSyn in yeast recapitulates some aspects of this phenotype (*Outeiro and Lindquist, 2003*). Indeed, αSyn forms toxic cytoplasmic foci that are detergent-insoluble, contain high molecular weight α-syn oligomers, and cluster cytoplasmic vesicles akin to some aspects of Lewy pathology observed in PD patients (*Araki et al., 2019*; *Fanning et al., 2020*; *Gitler et al., 2008*; *Jackrel et al., 2014*; *Jarosz and Khurana, 2017*; *Outeiro and Lindquist, 2003*; *Shahmoradian et al., 2019*; *Soper et al., 2008*; *Tenreiro et al., 2014*). Cytoplasmic αSyn foci in yeast have also been reported to react with the amyloid-diagnostic dye, thioflavin-S (*Franssens et al., 2010*; *Zabrocki et al., 2005*). We observed that all Hsp104 homologs that suppress αSyn toxicity also restore αSyn localization to the plasma membrane in ~75% of cells (*Figure 2F,G*). Cells with membrane-localized αSyn showed no cytoplasmic αSyn foci (*Figure 2F,G*). Taken together, our results demonstrate that Hsp104 homologs that suppress TDP-43 or αSyn toxicity also suppress the formation of TDP-43 or αSyn inclusions.

## Sequence characteristics of Hsp104 homologs

We next examined sequence relatedness among Hsp104 homologs in an effort to understand what sets Hsp104 homologs with particular toxicity-suppression characteristics apart. Hsp104 homologs with particular suppression characteristics cluster together phylogenetically (*Figure 1A*) although there are exceptions. Thus, while *A. thaliana* and *P. euphratica* are closely related species, AtHsp104 does not suppress TDP-43 toxicity while PeHsp104 does. We wondered whether Hsp104 homologs had particular sequence signatures that would predict their toxicity-suppression capacities. Multiple sequence alignments did not reveal any clear patterns to differentiate ScHsp104 from TDP-43-suppressing Hsp104s (*Figure 1—figure supplement 4*) or αSyn-suppressing Hsp104s (*Figure 2—figure supplement 1*). Similarly, there were no clear patterns differentiating TDP-43-suppressing Hsp104 homologs from αSyn-suppressing Hsp104 homologs (*Figure 2—figure supplement 2A*). We next calculated pairwise sequence identities between each possible pair of Hsp104 homologs

(*Supplementary file 2*), and compared the average percent identity of pairs with similar and dissimilar toxicity-suppression profiles (*Figure 2—figure supplement 2B*). The mean pairwise identity between TtHsp104 and TlHsp104, which both suppress αSyn toxicity, was 76%, which was unsurprising given that these homologs are from two closely related species. The ten pairwise identities comparing Hsp104 homologs that both rescue TDP-43 had a mean of 56% (with range of 51–71%), while the ten pairwise identities comparing one Hsp104 homolog that suppresses TDP-43 to another that suppresses αSyn had a mean of 44% (with range of 25–49%). Thus, homologs that suppress TDP-43 or αSyn seem to be more similar to one another than between groups.

## ClpG$_{GI}$ from *Pseudomonas aeruginosa* reduces TDP-43, FUS, and αSyn toxicity

In addition to the eukaryotic Hsp104 homologs discussed above, we also studied two prokaryotic Hsp104 homologs: ClpB from *Escherichia coli* and ClpG$_{GI}$ from the pathogenic bacteria *Pseudomonas aeruginosa*. WT ClpB does not suppress TDP-43, FUS, or αSyn toxicity (*Figure 2—figure supplement 3A*). We wondered whether ClpB activity could be enhanced via missense mutations in the MD, in analogy with Hsp104, so we also selected two previously-described hyperactive ClpB variants, ClpB$^{K476D}$ and ClpB$^{Y503D}$ (*Castellano et al., 2015*; *Oguchi et al., 2012*; *Rizo et al., 2019*), to test for suppression of TDP-43, FUS, and αSyn toxicity. Neither ClpB$^{K476D}$ nor ClpB$^{Y503D}$ suppressed TDP-43, FUS, or αSyn toxicity (*Figure 2—figure supplement 3A*). Thus, despite being able to exert forces of more than 50 pN and translocate polypeptides at speeds of more than 500 residues per second (*Avellaneda et al., 2020*), neither ClpB nor hyperactive ClpB variants are capable of suppressing TDP-43, FUS, or αSyn proteotoxicity in yeast.

Next, we tested ClpG$_{GI}$, which bears significant homology to Hsp104 but is distinguished by an extended NTD and a shorter MD roughly corresponding to loss of MD Motif 2 (*Lee et al., 2018*; *Figure 1—figure supplement 1*). ClpG$_{GI}$ has been reported to be a more effective disaggregase than ClpB from *E. coli* (*Katikaridis et al., 2019*). Furthermore, we previously established that deleting Motif 2 from ScHsp104 yields a potentiated variant able to rescue TDP-43, FUS, or αSyn toxicity (*Jackrel et al., 2015*). Thus, we wondered whether ClpG$_{GI}$ might suppress TDP-43, FUS, or αSyn toxicity. Indeed, we found ClpG$_{GI}$ potently suppresses αSyn toxicity and slightly suppresses TDP-43 and FUS toxicity (*Figure 2—figure supplement 3B*). ClpB, ClpB variants, and ClpG$_{GI}$ all express robustly in yeast and do not affect TDP-43, FUS, or αSyn levels (*Figure 2—figure supplement 3C–E*). ClpB and ClpG$_{GI}$ are also not toxic to yeast when expressed at 37°C (*Figure 2—figure supplement 3F*). Thus, ClpG$_{GI}$ is a prokaryotic disaggregase with therapeutic properties and may represent a natural example of Hsp104 potentiation via loss of Motif 2 from the MD (*Jackrel et al., 2015*).

## Hsp104 homologs prevent TDP-43 aggregation in human cells

We next examined whether Hsp104 homologs that suppress TDP-43 toxicity in yeast would have a beneficial effect in higher model systems. Expression of Hsp104 or potentiated variants in mammalian cells is well tolerated and can be cytoprotective (*Bao et al., 2002*; *Carmichael et al., 2000*; *Mosser et al., 2004*; *Yasuda et al., 2017*). Thus, we transfected human HEK293T cells with an inducible plasmid encoding fluorescently-tagged TDP-43 lacking a functional nuclear-localization sequence (mClover3-TDP43ΔNLS) to enhance cytoplasmic accumulation and aggregation (*Guo et al., 2018*; *Winton et al., 2008*). We co-transfected these cells with an empty vector or inducible plasmids encoding ScHsp104$^{WT}$, the potentiated variant ScHsp104$^{A503S}$, PeHsp104, SrHsp104, or a catalytically-inactive variant ScHsp104$^{DPLA:DWB}$ deficient in both peptide translocation (due to Y257A and Y662A mutations in NBD1 and NBD2 substrate-binding pore loops) and ATP hydrolysis (due to E285Q and E687Q mutations in NBD1 and NBD2 Walker B motifs) (*DeSantis et al., 2012*). PeHsp104 and SrHsp104 display the most potent and selective suppression of TDP-43 toxicity in yeast (*Figure 1C*). We monitored TDP-43 expression and aggregation by pulse-shape analysis of flow cytometry data (see Materials and methods) (*Ramdzan et al., 2013*) over time and quantified the percentage of cells bearing aggregates upon coexpression of each Hsp104 (*Figure 3A*, *Figure 3—figure supplement 1*). Expression of Hsp104 variants in HEK293T cells was confirmed by Western blot (*Figure 3B,C*). At both 24 hr and 48 hr post-transfection, PeHsp104 consistently accumulated to higher levels than other Hsp104 homologs (*Figure 3B,C*). We also monitored mClover3-TDP-43ΔNLS levels by Western blot (*Figure 3B,D*). None of the conditions tested

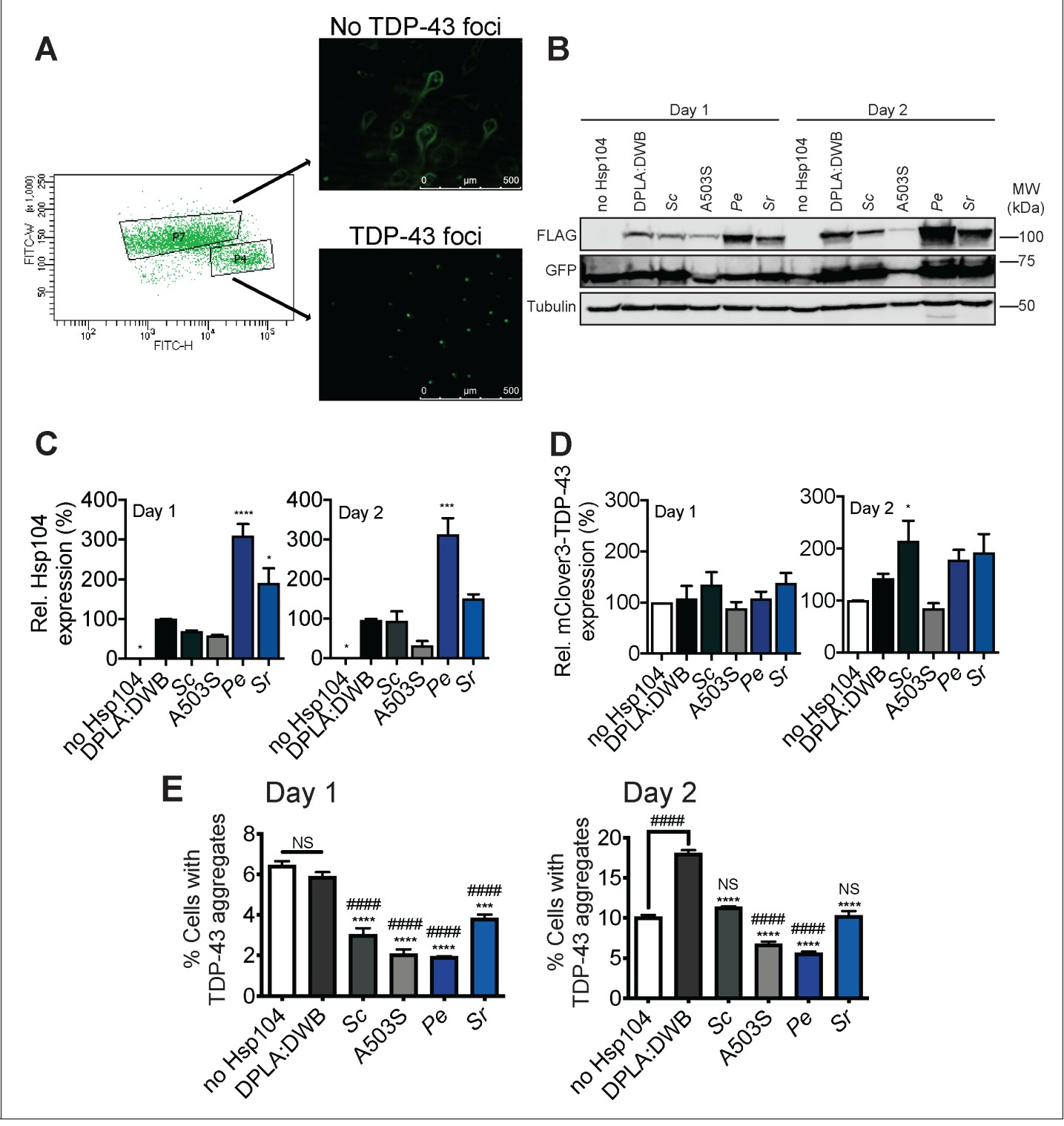

**Figure 3.** Hsp104 homologs reduce TDP-43 aggregation in HEK293T cells. (**A**) HEK293T cells were cotransfected with doxycycline-inducible constructs encoding mClover3-TDP-43ΔNLS. Protein expression was induced with 1 µg/ml doxycycline 6 hr post-transfection. At varying times, cells were sorted by FACS into populations lacking TDP-43 foci (**P7**) or cells with TDP-43 foci (**P4**). Representative fluorescent microscopy of sorted cells is shown at right. Scale bar, 500 µm. (**B**) At days 1 and 2 post-transfection, cells were processed for Western blot to confirm Hsp104-FLAG expression and mClover3-TDP-43ΔNLS (detected with a GFP antibody) expression. Tubulin is used as a loading control. Molecular weight markers are indicated (right). (**C**) Expression of the indicated Hsp104-FLAG relative to tubulin was quantified for each condition at day 1 (left) and day 2 (right) post-transfection. Values are means ± SEM (n = 3). One-way ANOVA with Dunnett's multiple comparisons test was used to compare expression of DPLA:DWB to all other conditions.

*Figure 3 continued on next page*

*Figure 3 continued*

*p<0.01; ***p<0.001. (**D**) mClover3-TDP43ΔNLS expression relative to tubulin was quantified for each condition at day 1 (left) and day 2 (right) post-transfection. Values are means ± SEM (n = 3). One-way ANOVA with Dunnett's multiple comparisons test was used to compare mClover3-TDP43ΔNLS levels in cells expressing no Hsp104 to all other conditions. *p<0.05. (**E**) At day 1 post-transfection, cells were analyzed by flow cytometry to quantify cells bearing TDP-43 aggregates (E, *left*). Cells were also analyzed by flow cytometry at day 2 post-transfection (E, *right*) Values are means ± SEM (n = 3 independent transfections with 10,000 cells counted per trial). One-way ANOVA with Tukey's multiple comparisons test was used to compare no Hsp104 (#) and DPLA:DWB (*) to all other conditions, and to each other. ###/***p<0.001; ####/****p<0.0001; NS, not significant.

The online version of this article includes the following figure supplement(s) for figure 3:

**Figure supplement 1.** Pulse-shape plots for HEK293T TDP-43ΔNLS co-transfection experiments.

led to significant reduction of TDP-43 expression (*Figure 3B,D*), although some conditions exhibited increased TDP-43 levels (*Figure 3B,D*). This finding suggests that Hsp104 variants do not merely affect accumulation of TDP-43 foci by reducing TDP-43 levels. At 24 hr post-transfection, all catalytically active Hsp104 variants significantly decreased the proportion of cells with TDP-43 aggregates compared to cells expressing no Hsp104 or cells expressing the catalytically-inactive ScHsp104$^{DPLA:DWB}$ (*DeSantis et al., 2012*; *Figure 3E*, *left panel,* Day 1). We were surprised that ScHsp104$^{WT}$ reduced the proportion of cells with TDP-43 aggregates at this time point, given that it does not reduce TDP-43 aggregation in yeast (*Figure 1G*; *Figure 1I*). However, at 48 hr post-transfection, cells expressing ScHsp104$^{WT}$ had a similar proportion of cells with TDP-43 aggregates as the vector control (*Figure 3E*, *right panel,* Day 2). By contrast, the catalytically-inactive variant ScHsp104$^{DPLA:DWB}$ had a significantly increased proportion of cells with TDP-43 aggregates compared to cells expressing no Hsp104 or ScHsp104$^{WT}$ (*Figure 3E*, *right panel,* Day 2). Strikingly, cells expressing the potentiated variant ScHsp104$^{A503S}$ or the TDP-43-specific variant PeHsp104 continued to show a significantly lower percentage of cells with TDP-43 aggregates (*Figure 3E*, *right panel,* Day 2). The TDP-43-specific variant SrHsp104, meanwhile, reduced TDP-43 aggregate burden at day 1 but not day 2, suggesting an intermediate effect (*Figure 3E*). Thus, while ScHsp104 and SrHsp104 show a short-lived suppression of TDP-43 foci formation, we define two Hsp104 variants, one engineered (A503S) and one natural (PeHsp104) that show an enduring reduction of TDP-43 foci in human cells.

## αSyn-selective Hsp104 homologs prevent dopaminergic neurodegeneration in *C. elegans*

To test whether TtHsp104, and TlHsp104, which selectively suppress αSyn toxicity in yeast, would likewise suppress αSyn toxicity in animals, we turned to a *C. elegans* model of Parkinson's disease in which the dopamine transporter (*dat-1*) promoter is used to direct expression of αSyn to dopaminergic (DA) neurons (*Cao et al., 2005*). We generated transgenic worms expressing αSyn either alone or in combination with different Hsp104 variants in DA neurons, and confirmed Hsp104 and αSyn expression by qRT-PCR (*Figure 4A,B*). Only ~20% of worms expressing αSyn alone have a full complement of DA neurons at day 7 post-hatching (*Figure 4C,D*). WT Hsp104 from *S. cerevisiae* does not protect *C. elegans* DA neurons in this context (*Jackrel et al., 2014*). Coexpression of TtHsp104 or TlHsp104, which both selectively mitigate αSyn toxicity in yeast, both result in significant protection of DA neurons in *C. elegans* (~40% worms with normal DA neurons in each case) after 7 days post-hatching (*Figure 4C,D*). Additionally, Hsp104 homologs are not intrinsically toxic, as worms expressing DA neuron-localized Hsp104 homologs in the absence of αSyn have a full complement of neurons at 7 days post-hatching (*Figure 4—figure supplement 1*). The level of DA neuron protection conferred by TtHsp104 and TlHsp104 is comparable to that conferred by the potentiated Hsp104 variants, Hsp104$^{A503S}$ and Hsp104$^{DPLF:A503V}$ (*Jackrel et al., 2014*). Thus, our results demonstrate that natural, substrate-specific Hsp104 homologs can function in a wide variety of contexts, including in an intact animal nervous system.

We also assessed how CtHsp104 affects αSyn toxicity in *C. elegans* DA neurons (*Figure 4—figure supplement 2*) and TDP-43 aggregation in HEK293T cells (*Figure 4—figure supplement 3*). Surprisingly, CtHsp104 does not affect *C. elegans* DA neuron survival (*Figure 4—figure supplement 2*), despite robust suppression of αSyn-mediated toxicity and inclusion formation in yeast (*Figure 2A*, F, G; *Michalska et al., 2019*). CtHsp104 likewise fails to suppress TDP-43 aggregation in HEK293T cells (*Figure 4—figure supplement 3*) despite suppression of TDP-43 toxicity in yeast

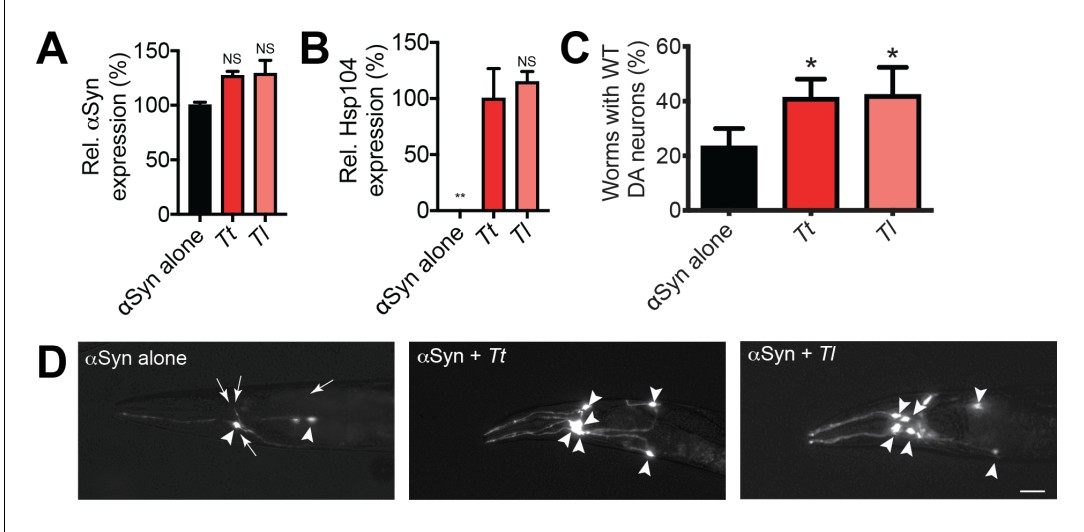

**Figure 4.** Hsp104 homologs protect against αSyn-mediated dopaminergic neurodegeneration in *C. elegans*. (**A**) qRT-PCR for the expression of αSyn and various Hsp104 homologs in transgenic *C. elegans*. αSyn expression was normalized to transgenic worms expressing αSyn alone. Values represent means ± SEM (N = 100 worms per transgenic line, three independent transgenic lines examined for each genotype). The expression of αSyn among all genotypes was not significantly different, as assessed by one-way ANOVA with Dunnett's multiple comparisons test to compare αSyn alone to all other conditions. (**B**) qRT-PCR for the expression of various Hsp104 homologs in transgenic *C. elegans*. Hsp104 expression was normalized to transgenic worms expressing both αSyn and TtHsp104 (*Tt*). Values represent means ± SEM (N = 100 worms per transgenic line, three independent transgenic lines examined for each genotype). One-way ANOVA with Dunnett's multiple comparisons test was used to compare TtHsp104 to all other conditions. **p<0.01; NS, not significant. (**C**) αSyn and the indicated Hsp104 homolog were coexpressed in the dopaminergic (DA) neurons of *C. elegans*. Hermaphrodite nematodes have six anterior DA neurons, which were scored at day seven posthatching. Worms are considered WT if they have all six anterior DA neurons intact (see methods for more details). TtHsp104 and TlHsp104 significantly protect dopaminergic neurons compared to αSyn alone. Values represent means ± SEM (n = 30 worms per genotype per replicate, three independent replicates). One-way ANOVA with Dunnett's multiple comparisons test was used to compare αSyn alone to all other conditions. *p<0.05. (**D**) Photomicrographs of the anterior region of *C. elegans* coexpressing GFP with αSyn. Worms expressing αSyn alone (left) exhibit an age-dependent loss of DA neurons. Worms expressing αSyn plus either *Tt* (middle) or *Tl* (right) exhibit greater neuronal integrity. Arrows indicate degenerating or missing neurons. Arrowheads indicate normal neurons. Scale bar, 10 μm.

The online version of this article includes the following figure supplement(s) for figure 4:

**Figure supplement 1.** Hsp104 homologs are not intrinsically toxic to *C. elegans* DA neurons.

**Figure supplement 2.** CtHsp104 does not protect *C. elegans* DA neurons from αSyn-mediated degeneration.

**Figure supplement 3.** CtHsp104 does not inhibit TDP-43 condensation in human cells.

---

(*Figure 1B*; *Michalska et al., 2019*). The lack of CtHsp104-mediated neuroprotection in *C. elegans* and TDP-43 aggregation-inhibition activity in HEK293T cells may be due to the fact that CtHsp104 is promiscuous. By contrast, the substrate-specific Hsp104 homologs were more effective in metazoan model systems.

## Differential suppression of proteotoxicity by Hsp104 homologs is not due to changes in disaggregase activity

Next, we sought to understand why some Hsp104 homologs suppress TDP-43 or αSyn toxicity while others do not. One possible explanation is that Hsp104 homologs differ in disaggregase activity, as has been the case with potentiated Hsp104 variants (*Jackrel et al., 2014*; *Jackrel and Shorter, 2014*; *Jackrel et al., 2015*; *Tariq et al., 2019*; *Tariq et al., 2018*; *Torrente et al., 2016*). Potentiated disaggregases typically display elevated ATPase and disaggregase activities, including having substantial disaggregase activity even in the absence of Hsp70 and Hsp40 chaperones (*Figure 5—figure supplement 1A–C*; *Jackrel et al., 2014*; *Jackrel and Shorter, 2014*; *Jackrel et al., 2015*; *Tariq et al., 2019*; *Tariq et al., 2018*; *Torrente et al., 2016*). This elevated activity can sometimes manifest as a temperature-dependent toxicity phenotype (*Figure 1—figure supplement 2*; *Jackrel et al., 2014*; *Jackrel et al., 2015*). However, the Hsp104 homologs we assess here (except for PfHsp104) are non-toxic under conditions where some potentiated Hsp104 variants, such as

Hsp104$^{A503V}$, are toxic (*Figure 1—figure supplement 2*). This finding hints that these natural homologs are not potentiated in the same way as engineered variants. To explore this issue further, we directly assessed whether the toxicity-suppression behavior of Hsp104 homologs could be explained by differences in disaggregase activity.

First, we tested how well our Hsp104 homologs conferred thermotolerance (i.e. the ability to survive a 50˚C heat shock) to yeast. Hsp104 is an essential factor for induced thermotolerance in yeast (*Sanchez and Lindquist, 1990*), and Hsp104 homologs in bacteria and plants have similar functions in their respective hosts (*Mogk et al., 1999*; *Queitsch et al., 2000*). The ability of Hsp104 to confer thermotolerance depends on its disaggregase activity, which solubilizes proteins trapped in heat-induced protein assemblies (*Glover and Lindquist, 1998*; *Parsell et al., 1994b*; *Parsell et al., 1991*; *Tessarz et al., 2008*; *Wallace et al., 2015*). Thus, thermotolerance is a convenient in vivo proxy for disaggregase activity among different Hsp104 homologs. Indeed, ~75% of WT yeast survive a 20 min heat shock at 50˚C whereas ~1% of Δ*hsp104* mutants survive the same shock (*Figure 5A,B*). Expressing FLAG-tagged *Sc*Hsp104 from a plasmid effectively complements the thermotolerance defect of Δ*hsp104* yeast (*Figure 5A,B*). We generated transgenic yeast strains in which Hsp104 homologs are expressed under the control of the native *S. cerevisiae HSP104* promoter (except for TtHsp104, which was expressed from p*GAL*-see Materials and Methods), and assessed the thermotolerance phenotypes of these strains. We observed a range of phenotypes. Specifically, 15 of 17 Hsp104s tested conferred some degree of thermotolerance above Δ*hsp104* alone (*Figure 5A,B*). The two exceptions were ClpB from *E. coli* and Hsp104 from *Populus euphratica* (*Figure 5A,B*). Some homologs, such as those from *Thielavia terrestris, Galdieria sulphuraria*, and *Dictyostelium discoideum* strongly complement thermotolerance while others were relatively weak (*Figure 5A,B*). Thermotolerance phenotypes are not explained by the evolutionary divergence time of a particular species from *S. cerevisiae* (*Figure 5C*). Thermotolerance phenotypes for Hsp104 homologs that reduce TDP-43 or αSyn toxicity do not differ noticeably from thermotolerance phenotypes for Hsp104 homologs that rescue neither TDP-43 nor αSyn toxicity (*Figure 5D*). Interestingly, ClpG$_{GI}$ confers a strong thermotolerance phenotype to Δ*hsp104* yeast (*Figure 2—figure supplement 3G*) while ClpB does not (*Parsell et al., 1993*), although both are from prokaryotes. This difference is likely due to the fact that ClpG$_{GI}$ is a stand-alone disaggregase and does not depend on Hsp70 and Hsp40 for disaggregation (*Katikaridis et al., 2019*; *Lee et al., 2018*) while ClpB is incompatible with yeast Hsp70 and Hsp40 (*Krzewska et al., 2001*; *Miot et al., 2011*; *Parsell et al., 1993*). Nevertheless, our results demonstrate that evolutionarily diverse Hsp104 homologs confer thermotolerance to Δ*hsp104* yeast, but differences in thermotolerance activity do not explain differences in suppression of TDP-43 or αSyn toxicity.

Next, we assessed how differences in protein expression between strains may contribute to phenotypic differences. All strains mount an effective heat-shock response, as indicated by Hsp26 levels assessed by Western blot (*Figure 5E,F*). We also measured Hsp104-FLAG expression levels in each strain by Western blot (*Figure 5E,G*). Although expression of Hsp104 homologs is somewhat variable (*Figure 5E,G*), expression level is a poor predictor of thermotolerance (*Figure 5H*, R$^2$ = 0.24).

We next expressed and purified several Hsp104 homologs (MbHsp104, CrHsp104, TtHsp104, TlHsp104, ScHsp104, and CtHsp104) to define their biochemical properties. Hsp104 homologs likely form hexamers (*Figure 6—figure supplement 1*; *Michalska et al., 2019*), are active ATPases (which requires Hsp104 hexamerization [*Mackay et al., 2008*; *Parsell et al., 1994a*; *Parsell et al., 1994b*; *Schirmer et al., 1998*; *Schirmer et al., 2001*]), and display increased ATPase activity with increasing temperature (*Figure 6A*). These findings are consistent with the fact that Hsp104 is a disaggregase induced by thermal stress.

We also investigated the disaggregase activity of Hsp104 homologs (which also requires hexamerization [*Mackay et al., 2008*; *Parsell et al., 1994a*; *Parsell et al., 1994b*; *Schirmer et al., 1998*; *Schirmer et al., 2001*]) by assessing their ability to disaggregate chemically-denatured luciferase aggregates in vitro (*DeSantis et al., 2012*; *Glover and Lindquist, 1998*). These aggregated structures are ~500–2,000 kDa or greater in size and cannot be disaggregated by Hsp70 and Hsp40 alone (*Cupo and Shorter, 2020*; *Glover and Lindquist, 1998*; *Shorter, 2011*). Typically, when combined with Hsp70 and Hsp40, ScHsp104 recovers ~10–30% of the native luciferase activity from these aggregated structures (*Cupo and Shorter, 2020*; *Glover and Lindquist, 1998*; *Shorter, 2011*). First, we compared the disaggregase activity of ScHsp104$^{WT}$ to the potentiated variant ScHsp104$^{A503S}$, which exhibits elevated ATPase activity (*Figure 5—figure supplement 1A*; *Jackrel et al., 2014*). We

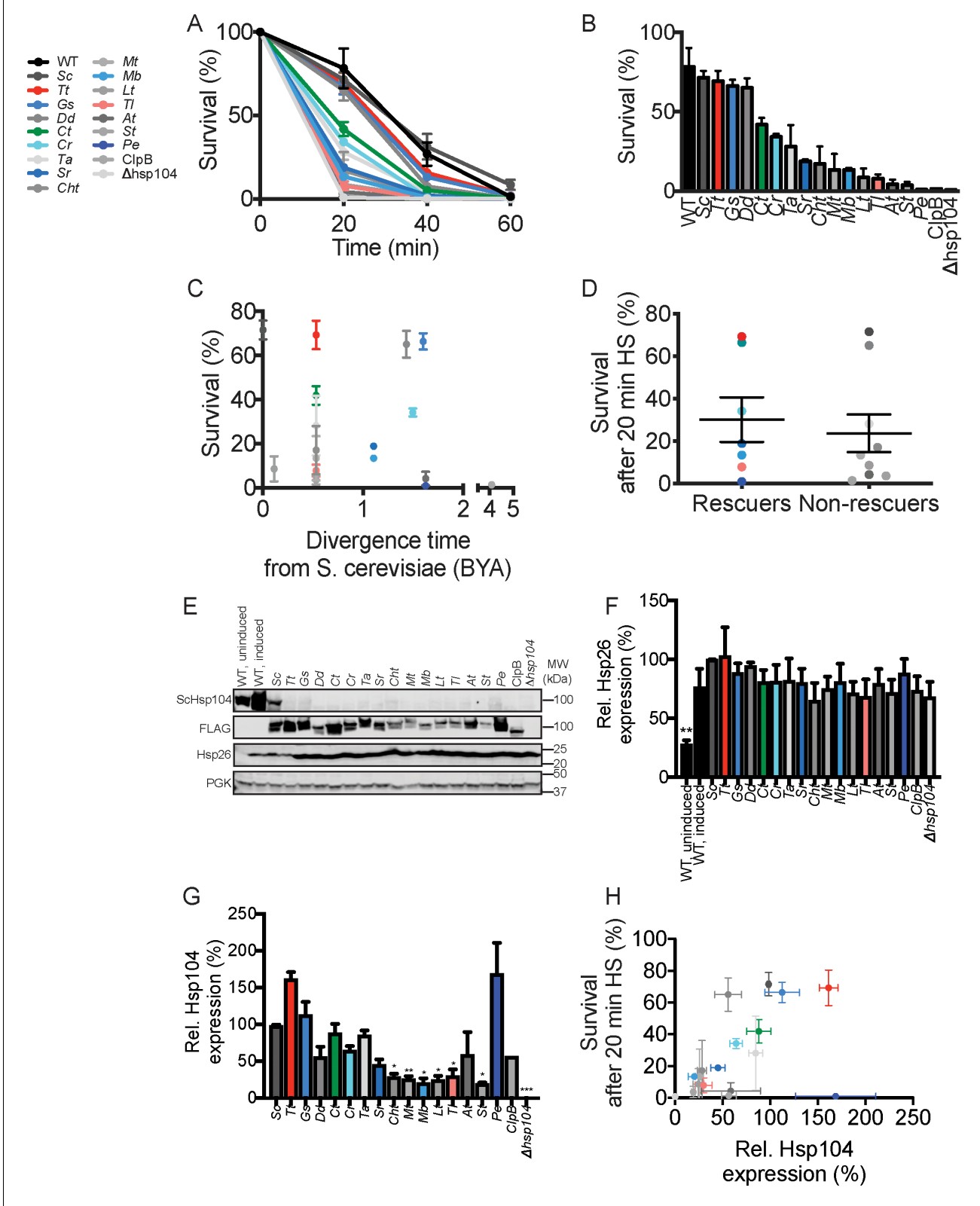

**Figure 5.** Hsp104 homologs function in induced thermotolerance but differences in thermotolerance activity do not explain suppression of TDP-43 or αSyn toxicity. (**A**) WT or Δ*hsp104* yeast carrying a plasmid encoding the indicated Hsp104 homolog under the control of the native *HSP104* promoter (except for TtHsp104, which was expressed from p*GAL*-see Materials and Methods) were pre-treated at 37°C for 30 min, treated at 50°C for 0–60 min, and plated. Surviving colonies were quantified after 2d recovery. Values represent means ± SEM (n = 3 independent transformations). (**B**)

*Figure 5 continued*

Hsp104 homologs ranked by thermotolerance performance after a 20 min heat shock at 50°C. (C) Survival after 20 min heat shock does not correlate with the evolutionary separation between a given species and *S. cerevisiae*. (D) Thermotolerance activity of Hsp104 homologs that suppress TDP-43 or αSyn toxicity ('Rescuers') does not noticeably differ from Hsp104 homologs that do not suppress TDP-43 or αSyn toxicity ('Non-rescuers'). (E) Expression of Hsp104 and Hsp26 before (uninduced) or after pretreatment at 37°C for 30 min (induced) was assessed by Western blot. Molecular weight markers are indicated (right). PGK serves as a loading control. An ScHsp104-specific antibody was used to detect untagged ScHsp104 or ScHsp104-FLAG. A FLAG antibody was used to detect Hsp104-FLAG. (F) Expression of Hsp26 relative to PGK was quantified for each strain. Values are means ± SEM (n = 3). One-way ANOVA with Dunnett's multiple comparisons test was used to compare expression of Hsp26 in the WT, induced strain to all other conditions. **p<0.01. (G) Expression of Hsp104-FLAG relative to PGK was quantified for each strain. Values are means ± SEM (n = 3). One-way ANOVA with Dunnett's multiple comparisons test was used to compare expression of ScHsp104-FLAG (*Sc*) to all other conditions. *p<0.05; **p<0.01; ***p<0.001. (H) Hsp104-FLAG expression is a weak predictor of yeast survival after 20 min heat shock. Values represent means ± SEM (n = 3). A simple linear regression yielded a coefficient of determination, $R^2$ = 0.24.

The online version of this article includes the following figure supplement(s) for figure 5:

**Figure supplement 1.** Biochemical comparison of wild-type ScHsp104 to the potentiated variant ScHsp104$^{A503S}$.

found that ScHsp104$^{A503S}$ has enhanced luciferase disaggregation activity as expected in the absence of Hsp70 and Hsp40 (where ScHsp104 is inactive (*Figure 5—figure supplement 1B*)) and in

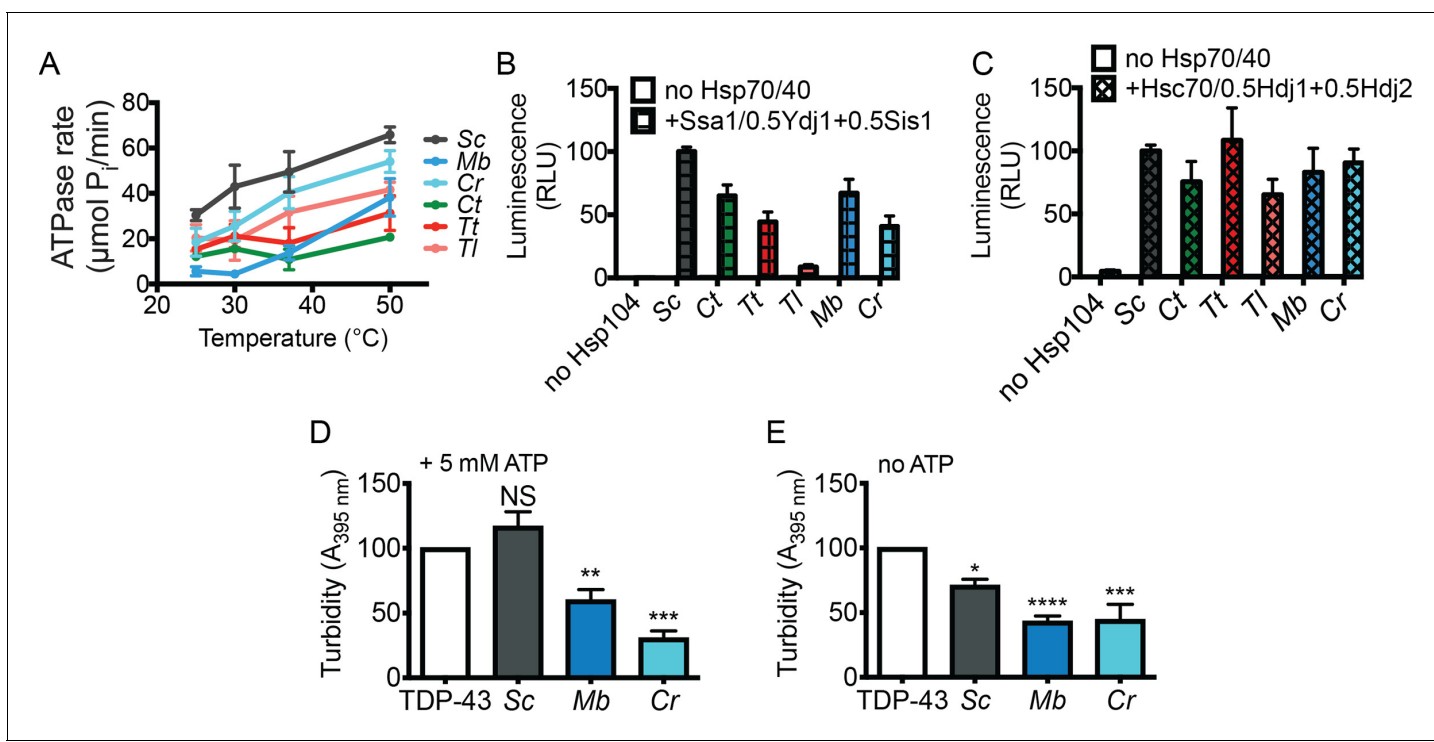

**Figure 6.** Hsp104 homologs are disaggregases in vitro but differences in disaggregase activity do not explain suppression of TDP-43 or αSyn toxicity. (A) ATPase activity of the indicated Hsp104 homologs at different temperatures. Values represent means ± SEM (n = 3). (B) Luciferase aggregates (50 nM) were incubated with the indicated Hsp104 (0.167 µM hexamer) with or without 0.167 µM Ssa1, 0.073 µM Ydj1, and 0.073 µM Sis1 for 90 min at 25°C. Values represent means ± SEM (n = 3). (C) Luciferase aggregates were treated as in (B) but Ssa1, Ydj1, and Sis1 were replaced with Hsc70, Hdj1, and Hdj2. Values represent means ± SEM (n = 3). (D) TDP-43 (3 µM) was incubated in the presence of the indicated Hsp104 (6 µM) and 5 mM ATP, and turbidity was measured at 3 hr relative to TDP-43 aggregation reactions containing no Hsp104. Values represent means ± SEM (n = 3). One-way ANOVA with Dunnett's multiple comparisons test was used to compare TDP-43 alone to all other conditions. NS, not significant; **p<0.01; ***p<0.001. (E) As in (D), except ATP was omitted and turbidity was measured at 2 hr. Values represent means ± SEM (n = 3). One-way ANOVA with Dunnett's multiple comparisons test was used to compare TDP-43 alone to all other conditions. *p<0.05; ****p<0.0001; ***p<0.001.

The online version of this article includes the following figure supplement(s) for figure 6:

**Figure supplement 1.** Hsp104 homologs form hexamers.
**Figure supplement 2.** Hsp104 homologs remodel SEVI fibrils.
**Figure supplement 3.** Hsp104 homologs do not affect MBP-TEV-TDP43 cleavage by TEV protease.

the presence of Hsp70 and Hsp40 (*Figure 5—figure supplement 1C*; *Jackrel et al., 2014*). Next, we assessed the Hsp104 homologs, which could all disassemble and reactivate aggregated luciferase (*Figure 6B,C*). Luciferase disaggregation required the presence of Hsp70 and Hsp40 chaperones, which could be from yeast (Ssa1, Sis1, and Ydj1 *Figure 6B*) or human (Hsc70, Hdj1, and Hdj2 *Figure 6C*). The robust ATPase and disaggregase activity of the Hsp104 homologs provides strong evidence that they assemble in functional hexamers like ScHsp104. We did not observe differences in luciferase disaggregation activity of Hsp104 homologs that would readily explain selective suppression of TDP-43 toxicity versus αSyn toxicity (*Figure 6B,C*).

Next, we tested the activity of several Hsp104 homologs (ScHsp104, CtHsp104, TtHsp104, AtHsp104, and MtHsp104) against an ordered amyloid substrate, semen-derived enhancer of viral infection (SEVI) (*Münch et al., 2007*). As previously reported for ScHsp104 and CtHsp104 (*Castellano et al., 2015*; *Michalska et al., 2019*), all Hsp104 homologs tested rapidly remodeled SEVI fibrils (*Figure 6—figure supplement 2A*). Electron microscopy revealed that Hsp104 homologs remodeled SEVI fibrils into small, amorphous structures (*Figure 6—figure supplement 2B*). Thus, eukaryotic Hsp104 homologs are generally able to remodel amyloid fibrils, unlike prokaryotic ClpB or hyperactive variants that have limited ability to remodel SEVI amyloid (*Castellano et al., 2015*). However, there was not an obvious difference in amyloid-remodeling activity between homologs that suppress αSyn toxicity (CtHsp104 and TtHsp104) and those that do not (ScHsp104, AtHsp104, and MtHsp104). Taken together, these findings suggest that differences in proteotoxicity suppression by Hsp104 homologs is not simply due to differences in their general disaggregase activity.

## Hsp104 homologs can inhibit protein aggregation in an ATP-independent manner

Since differences in general disaggregase activity do not explain differences in proteotoxicity suppression among Hsp104 homologs, we wondered whether Hsp104 homologs may act instead to inhibit protein aggregation. To test this possibility, we reconstituted TDP-43 aggregation in vitro to test how Hsp104 homologs affect TDP-43 aggregation (*Mann et al., 2019*). We performed reactions in the presence (*Figure 6D*) or absence (*Figure 6E*) of ATP. Hsp104 from *Monosiga brevicollis* and *Chlamydomonas reinhardtii*, which selectively suppress TDP-43 toxicity and foci formation in yeast, inhibit TDP-43 aggregation in vitro, whereas Hsp104 from *S. cerevisiae* has limited efficacy (*Figure 6D,E*). Hsp104-mediated inhibition of TDP-43 aggregation occurred in the presence or absence of ATP (*Figure 6D,E*). Hsp104 homologs did not inhibit TDP-43 aggregation merely by inhibiting cleavage of the MBP tag by TEV protease (*Figure 6—figure supplement 3*). Thus, unexpectedly, specific Hsp104 homologs likely suppress TDP-43 toxicity by inhibiting its aggregation in an ATP-independent manner.

## Hsp104 homologs can suppress toxicity of TDP-43 and αSyn in an ATPase-independent manner

Hsp104 homologs inhibited TDP-43 aggregation in vitro in the absence of nucleotide, indicating a passive mechanism of action. We next tested whether Hsp104 homologs also employed a passive mechanism to suppress TDP-43 or αSyn toxicity in yeast. Thus, we generated a series of mutants for each Hsp104 homolog intended to disrupt their disaggregase activity. Hsp104 disaggregase activity is driven by: (1) ATP binding and hydrolysis, which are mediated by Walker A and Walker B motifs, respectively, and which drive conformational changes within the hexamer to support substrate translocation (*DeSantis et al., 2012*; *Parsell et al., 1991*; *Ye et al., 2019*), and (2) substrate translocation through the central pore of the hexamer, which is mediated by tyrosine-bearing pore loops (*DeSantis et al., 2012*; *Gates et al., 2017*; *Lum et al., 2008*; *Lum et al., 2004*). These motifs are highly conserved among all Hsp104 homologs (*Figure 7A* and *Figure 1—figure supplement 1*). We mutated: (1) conserved lysine residues in the Walker A motifs to either alanine or threonine (Hsp104$^{DWA(KA)}$ and Hsp104$^{DWA(KT)}$) in NBD1 and NBD2 to impair ATP binding, (2) conserved glutamate residues in the Walker B motifs to either alanine or glutamine (e.g. Hsp104$^{DWB(EA)}$ or Hsp104$^{DWB(EQ)}$) in NBD1 and NBD2 to impair ATP hydrolysis, and (3) conserved tyrosines in the pore loops to alanine (e.g. Hsp104$^{DPLA}$) in NBD1 and NBD2 to impair substrate threading through the central hexamer pore (*DeSantis et al., 2012*). We also generated mutants lacking the NTD, which also plays a role in substrate binding and processing (*Sweeny et al., 2015*; *Sweeny et al., 2020*).

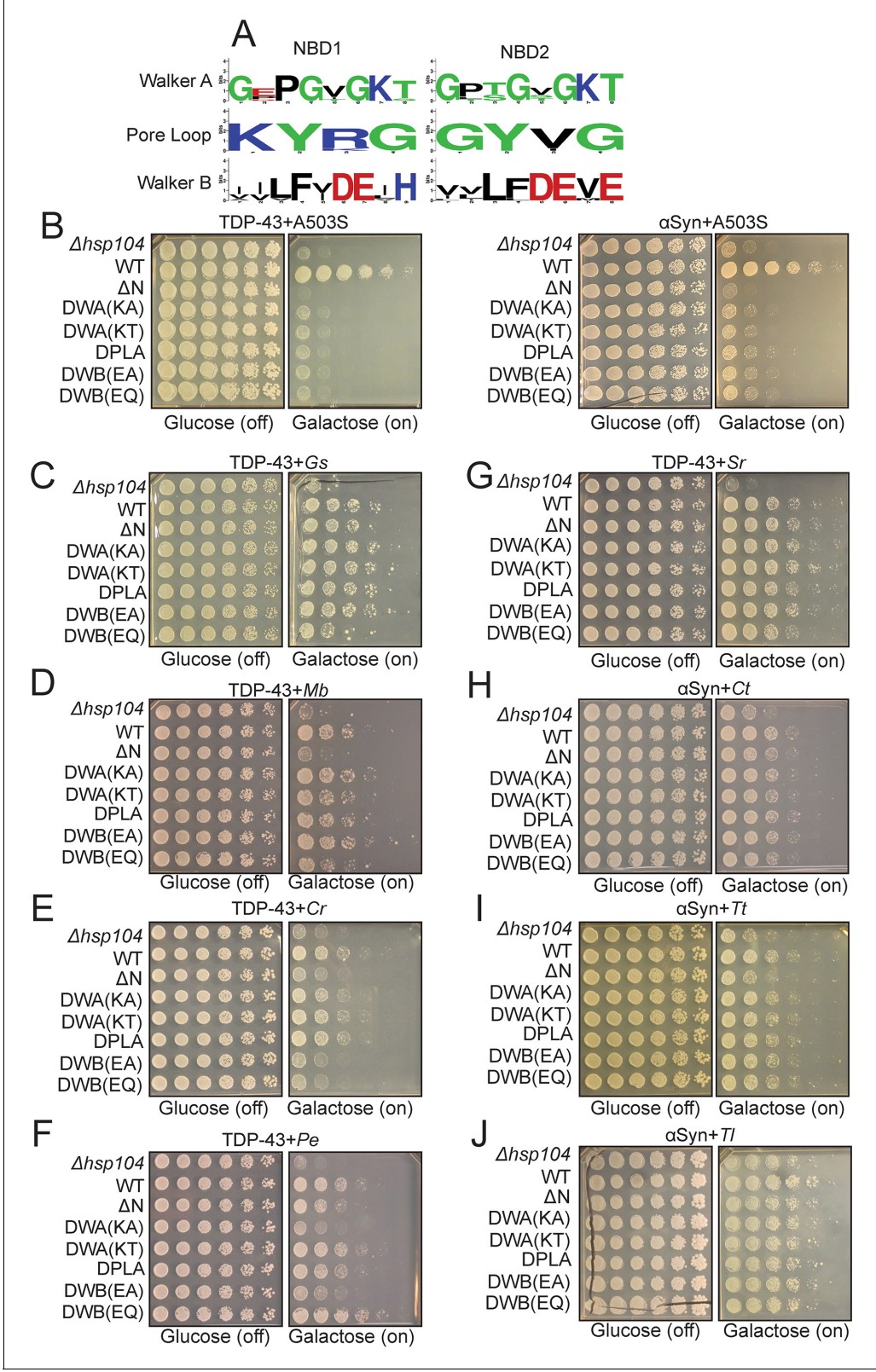

**Figure 7.** Hsp104 homologs can suppress toxicity of TDP-43 and αSyn in a manner that does not require conserved AAA+ motifs. (**A**) WebLogo sequence logos demonstrating high conservation of Walker A, tyrosine-bearing pore loops, and Walker B motifs in both NBD1 and NBD2 across all Hsp104 homologs. (**B–J**) Spotting assays to define how mutations affect the ability of Hsp104 variants to suppress TDP-43 or αSyn toxicity. Within each panel, distinct yeast strains are spotted in rows and are labeled by the type of Hsp104 being expressed in each instance (Δhsp104, no Hsp104 being

*Figure 7 continued on next page*

Figure 7 continued

expressed; WT, Hsp104 variant with no additional mutations; ΔN, Hsp104 variant lacking an NTD; DWA(KA) and DWA(KT), Hsp104 variant with the indicated substitutions in the Walker A motifs; DPLA, Hsp104 variant with pore-loop tyrosines mutated to alanine; DWB(EA) and DWB(EQ), Hsp104 variant with the indicated substitutions in the Walker B motifs. (B) Spotting assay demonstrating that Hsp104$^{A503S}$ (A503S)-mediated suppression of TDP-43 (left) and αSyn (right) toxicity is inhibited by NTD deletion (ΔN) and mutations in Walker A (DWA(KA) and DWA(KT)), pore loop (DPLA), and Walker B (DWB(EA) and DWB(EQ)) motifs. (C) Spotting assay demonstrating that GsHsp104 (Gs)-mediated suppression of TDP-43 toxicity is resistant to NTD deletion (ΔN) as well as mutations in Walker A (DWA(KA) and DWA(KT)), pore loop (DPLA), and Walker B (DWB(EA) and DWB(EQ)) motifs. (D) Spotting assay demonstrating that MbHsp104 (Mb)-mediated suppression of TDP-43 toxicity is ablated by NTD deletion (ΔN) but is resistant to mutations in Walker A (DWA(KA) and DWA(KT)), pore loop (DPLA), and Walker B (DWB(EA) and DWB(EQ)) motifs. (E) Spotting assay demonstrating that CrHsp104 (Cr)-mediated suppression of TDP-43 toxicity is ablated by NTD deletion (ΔN) and mutations in Walker B (DWB(EA) and DWB(EQ)) motifs but is resistant to mutations in Walker A (DWA(KA) and DWA(KT)) and pore loop (DPLA) motifs. (F) Spotting assay demonstrating that PeHsp104 (Pe)-mediated suppression of TDP-43 toxicity is resistant to NTD deletion (ΔN) and mutations in pore loop (DPLA) motifs, but is partially sensitive to mutations in Walker A (i.e. suppression is ablated by DWA(KA) but not DWA(KT)) and Walker B (i.e. suppression is ablated by DWB(EA) but not DWB(EQ)) motifs. (G) Spotting assay demonstrating that SrHsp104 (Sr)-mediated suppression of TDP-43 is resistant to NTD deletion (ΔN) as well as mutations in Walker A (DWA(KA) and DWA(KT)), pore loop (DPLA), and Walker B (DWB(EA) and DWB(EQ)) motifs. (H) Spotting assay demonstrating that CtHsp104 (Ct)-mediated suppression of αSyn toxicity is resistant to NTD deletion (ΔN) as well as mutations in Walker A (DWA(KA) and DWA(KT)), pore loop (DPLA), and Walker B (DWB(EA) and DWB(EQ)) motifs. (I) Spotting assay demonstrating that TtHsp104 (Tt)-mediated suppression of αSyn toxicity is ablated by NTD deletion (ΔN) but is resistant to mutations in Walker A (DWA(KA) and DWA(KT)), pore loop (DPLA), and Walker B (DWB(EA) and DWB(EQ)) motifs. (J) Spotting assay demonstrating that TlHsp104 (Tl)-mediated suppression of αSyn toxicity is resistant to NTD deletion (ΔN) as well as mutations in Walker A (DWA(KA) and DWA(KT)), pore loop (DPLA), and Walker B (DWB(EA) and DWB(EQ)) motifs.

The online version of this article includes the following figure supplement(s) for figure 7:

**Figure supplement 1.** Hsp104 mutants are consistently expressed and are defective in thermotolerance.

We monitored expression of Hsp104 mutant proteins by Western blot, and found all mutants were expressed similarly to WT Hsp104, from either the galactose or native *HSP104* promoter (*Figure 7—figure supplement 1A–H*). Yeast expressing Walker A, Walker B, or pore-loop mutant proteins are all severely impaired in thermotolerance compared to WT controls, while ΔN mutants are only mildly impaired in thermotolerance compared to WT proteins (*Figure 7—figure supplement 1I*). These findings are similar to thermotolerance phenotypes of ScHsp104 mutants that have been previously reported (*DeSantis et al., 2012*; *Parsell et al., 1991*; *Sweeny et al., 2015*). The only exception is PeHsp104, where the WT protein confers no thermotolerance benefit over Δ*hsp104* cells, which precludes any conclusions being drawn about the impact of mutations on this protein (*Figure 5B* and *Figure 7—figure supplement 1I*). To confirm that the Walker A, Walker B, or pore-loop mutations inactivate disaggregase activity in Hsp104 homologs at the pure protein level, we purified CtHsp104 variants with these mutations and assessed their effect on luciferase disaggregase activity. As expected, mutation of Walker A, Walker B, or pore-loop motifs eliminate CtHsp104 disaggregase activity (*Figure 7—figure supplement 1J*). Thus, we confirm that these mutations have a conserved effect on Hsp104 disaggregase activity.

Next, we examined how these mutations affect the ability of different Hsp104 homologs to reduce TDP-43 or αSyn toxicity in yeast. TDP-43 or αSyn expression levels were largely unaffected by Hsp104 mutants, as assessed by Western blot (*Figure 7—figure supplement 1A–H*). All of the aforementioned mutations strongly impair the ability of a potentiated Hsp104 variant, ScHsp104$^{A503S}$, to suppress TDP-43 and αSyn toxicity (*Sweeny et al., 2015*; *Torrente et al., 2016*; *Figure 7B*). Thus, ScHsp104$^{A503S}$ suppresses TDP-43 and αSyn toxicity by a disaggregase-mediated mechanism that requires the NTD, ATP binding and hydrolysis, and substrate-engagement by conserved pore-loop tyrosines (*Sweeny et al., 2015*; *Torrente et al., 2016*). Remarkably, however, Hsp104 homolog-mediated suppression of TDP-43 or αSyn toxicity was largely unaffected by our specific alterations to Walker A, Walker B, or pore-loop residues (*Figure 7C–J*). For example, mutation of conserved pore-loop tyrosines to alanine had no effect on suppression of TDP-43 or αSyn toxicity (*Figure 7C–J*). Thus, canonical substrate translocation is likely not required for these Hsp104 homologs to mitigate TDP-43 or αSyn toxicity.

Likewise, in most cases, mutation of Walker A or Walker B motifs did not affect suppression of TDP-43 or αSyn toxicity (*Figure 7C–J*). There were, however, some exceptions. CrHsp104 was inhibited by mutations in the Walker B motifs (*Figure 7E*). Thus, CrHsp104 requires conserved Walker B motifs (but not conserved Walker A motifs) for optimal suppression of toxicity (*Figure 7E*). Notably, this requirement was not coupled to substrate translocation by conserved pore-loop

tyrosines (*Figure 7E*). Additionally, PeHsp104 was modestly inhibited by K to A but not K to T substitutions in the Walker A motif and E to A but not E to Q substitutions in the Walker B motifs (*Figure 7F*). Thus, PeHsp104 likely requires some level of ATPase activity for optimal suppression of toxicity. It may be that the K to T substitution in the Walker A motif and the E to Q substitution in the Walker B motif do not have such large effects in this specific homolog. Regardless, any requirement for ATPase activity was not coupled to substrate translocation by conserved pore-loop tyrosines in PeHsp104 (*Figure 7F*). Importantly, the remaining Hsp104 homologs tested here (GsHp104, MbHsp104, SrHsp104, CtHsp104, TtHsp104, and TlHsp104) suppressed TDP-43 or αSyn toxicity by a mechanism that does not require ATPase activity or conserved pore-loop tyrosines that engage substrate in the Hsp104 channel (*Figure 7C,D,H–J*). Mechanistically, these findings suggest that toxicity suppression by the majority of Hsp104 homologs tested here (GsHp104, MbHsp104, SrHsp104, CtHsp104, TtHsp104, and TlHsp104) is primarily due to ATPase-independent activity against specific substrates, such as TDP-43 or αSyn, and not due to ATPase-dependent disaggregase activity or chaperone activity.

## Suppression of TDP-43 toxicity by MbHsp104 and CrHsp104, and suppression of αSyn toxicity by TtHsp104 requires the NTD

Next, we sought to identify the requisite architecture of Hsp104 homologs that enables TDP-43 or αSyn proteotoxicity suppression. The engineered ScHsp104 variant, Hsp104[A503S], requires the NTD to mitigate TDP-43 and αSyn toxicity (*Figure 7B*; *Sweeny et al., 2015*). TDP-43 or αSyn proteotoxicity suppression by several Hsp104 homologs (GsHsp104, PeHsp104, SrHsp104, CtHsp104, and TlHsp104) is not greatly affected by NTD deletion (*Figure 7C,F,G,H,J*). Interestingly, MbHsp104[ΔN] and CrHsp104[ΔN] are unable to suppress TDP-43 toxicity (*Figure 7D,E*), and TtHsp104[ΔN] is unable to suppress αSyn toxicity (*Figure 7I*). Thus, proteotoxicity suppression by these specific Hsp104 homologs requires their NTDs.

We wondered whether the NTDs might drive the toxicity-suppression phenotypes for MbHsp104, CrHsp104, and TtHsp104. To test this possibility as well as additional questions concerning domain requirements, we made a series of Hsp104 chimeras (*Figures 8* and *9*) in which we systematically replaced domains of ScHsp104 with the homologous domains from other Hsp104 homologs (see *Figures 8A* and *9A* for illustration of domain boundaries and chimeras). Prior studies indicate that chimeric proteins formed between Hsp104 homologs can form hexamers that possess robust disaggregase activity in vitro and in vivo (*DeSantis and Shorter, 2012a*; *Miot et al., 2011*).

We found that replacing the NTD of ScHsp104 with the NTD of either MbHsp104 or CrHsp104 did not enable suppression of TDP-43 toxicity by the resulting chimeras (*Figure 8B,C*). Similarly, replacing the ScHsp104 NTD with the NTD from TtHsp104 did not enable suppression of αSyn toxicity by the resulting chimera (*Figure 9B*). Thus, proteotoxicity suppression by MbHsp104, CrHsp104, and TtHsp104 is not simply transmitted through or encoded by the NTD. Rather, the NTD must work together with neighboring domains of the same Hsp104 homolog to enable toxicity suppression.

## Suppression of TDP-43 toxicity is enabled by NBD1 and MD residues in Hsp104 homologs

Based on our observation that suppression of TDP-43 and αSyn toxicity by MbHsp104, CrHsp104, and TtHsp104 is not conferred solely by the NTD (*Figures 8B,C* and *9B*), and the observation that ΔN mutants of other homologs (GsHsp104, PeHsp104, SrHsp104, CtHsp104, and TlHsp104) retained their ability to reduce either TDP-43 or αSyn toxicity (*Figure 7C,F,G,H,J*), we reasoned that residues in other domains must also contribute to substrate selectivity. Within the ScHsp104 hexamer, the NTD interacts with NBD1 and the MD (*Sweeny et al., 2020*). Thus, we assessed additional chimeras by progressively replacing the NTD, NBD1 or MD of ScHsp104 for the homologous domain from another Hsp104 homolog (see *Figures 8A* and *9A* for illustrations). Generally, the chimeras expressed well in yeast (*Figure 8—figure supplement 1*). We observed that chimeras consisting of the NTD, NBD1, and MD from a TDP-43-selective homolog appended to the NBD2 and CTD from ScHsp104 phenocopied the TDP-43 suppression phenotype associated with the homolog itself (*Figure 8B–F*). These same chimeras do not reduce αSyn toxicity (*Figure 8—figure supplement 2A–E*). Thus, these chimeras encompass the sequence determinants that are crucial for passive

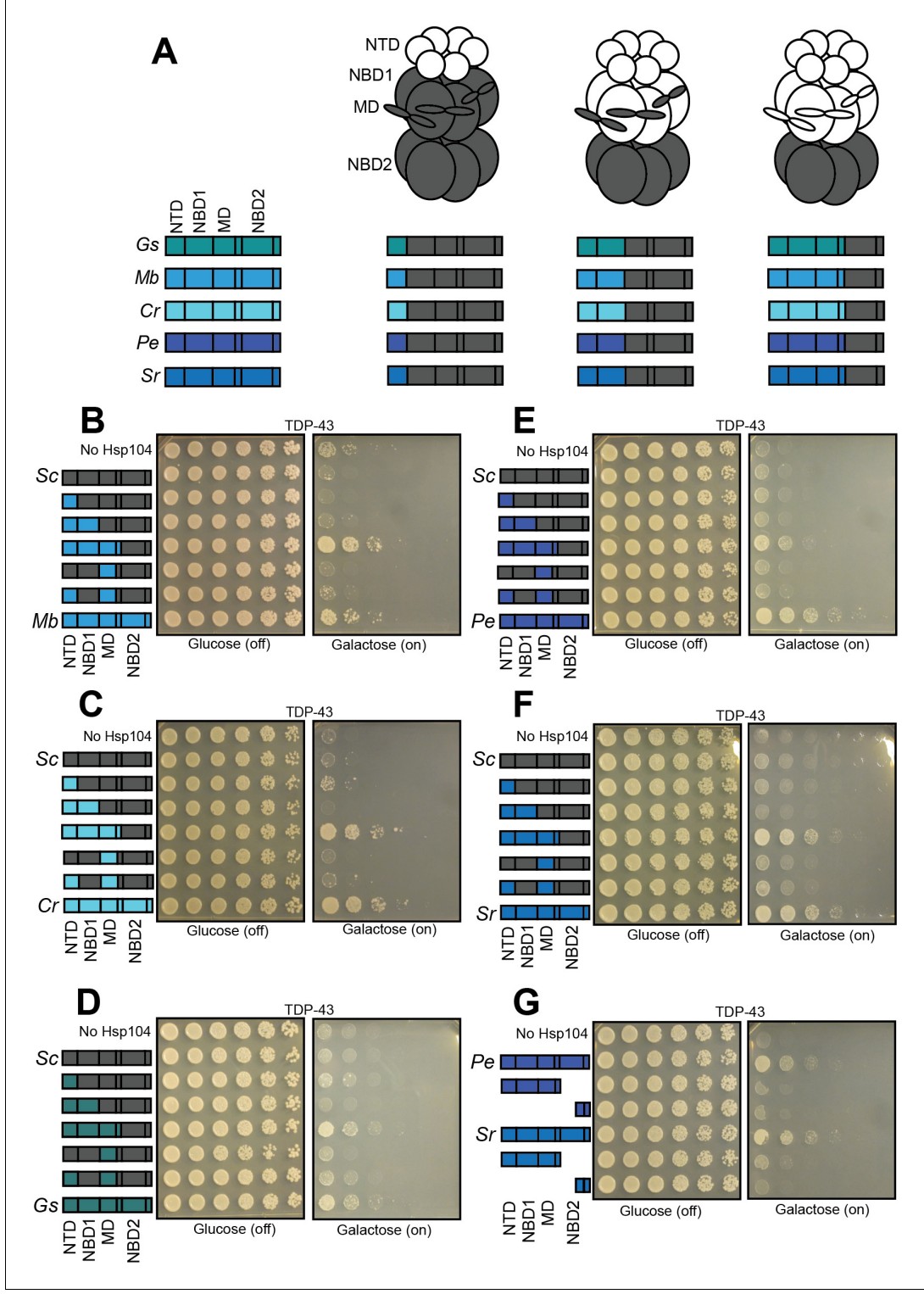

**Figure 8.** Interactions between the NTD, NBD1, and MD support TDP-43 toxicity suppression by Hsp104 homologs. (A) Color codes and domain boundaries and labels of Hsp104 homologs. (B) Spotting assay of Δ*hsp104* yeast coexpressing TDP-43 and the indicated chimeric Hsp104s between ScHsp104 and MbHsp104 illustrates that chimeras possessing the NTD, NBD1, and MD from MbHsp104 copy the TDP-43 toxicity-suppression phenotype of MbHsp104. (C) Spotting assay of Δ*hsp104* yeast coexpressing TDP-43 and the indicated chimeric Hsp104 between ScHsp104 and CrHsp104 illustrates that chimeras possessing the NTD, NBD1, and MD from CrHsp104 copy the TDP-43 toxicity-suppression phenotype of CrHsp104. (D) Spotting assay of Δ*hsp104* yeast coexpressing

*Figure 8 continued on next page*

*Figure 8 continued*

TDP-43 and the indicated chimeric Hsp104 between ScHsp104 and GsHsp104 illustrates that chimeras possessing the NTD, NBD1, and MD from GsHsp104 copy the TDP-43 toxicity-suppression phenotype of GsHsp104. (E) Spotting assay of Δ*hsp104* yeast coexpressing TDP-43 and the indicated chimeric Hsp104 between ScHsp104 and PeHsp104 illustrates that chimeras possessing the NTD, NBD1, and MD from PeHsp104 copy the TDP-43 toxicity-suppression phenotype of PeHsp104. (F) Spotting assay of Δ*hsp104* yeast coexpressing TDP-43 and the indicated chimeric Hsp104 between ScHsp104 and SrHsp104 illustrates that chimeras possessing the NTD, NBD1, and MD from SrHsp104 copy the TDP-43 toxicity-suppression phenotype of SrHsp104. (G) Spotting assay of Δ*hsp104* strains coexpressing TDP-43 and either full-length PeHsp104 or SrHsp104, or monomeric fragments derived from these homologs demonstrates that Hsp104-mediated toxicity suppression is an emergent property of hexameric Hsp104.

The online version of this article includes the following figure supplement(s) for figure 8:

**Figure supplement 1.** Chimeric Hsp104s and proteotoxic substrates are consistently expressed in yeast.
**Figure supplement 2.** Characterization of Hsp104 chimera specificity for TDP-43.
**Figure supplement 3.** Characterization of the intrinsic toxicity of Hsp104 chimeras.

---

inhibition of TDP-43 aggregation. Chimeras where cognate NTD:NBD1:MD networks were disrupted do not reduce TDP-43 toxicity (*Figure 8B–F*). We conclude that genetic variation in NBD1 and the MD of Hsp104 homologs also contributes to mitigation of TDP-43 proteotoxicity.

## Homolog-mediated suppression of TDP-43 toxicity requires the NBD2:CTD unit

Next, we tested whether NBD2 was required for suppression of TDP-43 toxicity by Hsp104 homologs, or whether expressing Hsp104 fragments encompassing the NTD, NBD1, and MD (which contain sequence determinants that reduce TDP-43 toxicity) would recapitulate the suppression of toxicity seen with the full-length homolog. We therefore co-expressed PeHsp104$^{1-541}$ and SrHsp104$^{1-551}$ with TDP-43 in yeast. Neither of these fragments reduce TDP-43 toxicity (*Figure 8G*; see *Figure 8—figure supplement 1L* for accompanying Western blot). Two additional fragments, PeHsp104$^{767-914}$ and SrHsp104$^{781-892}$, which correspond to the small domain of NBD2 and the CTD also did not reduce TDP-43 toxicity (*Figure 8G*). Since these Hsp104 fragments are likely monomeric (*Hattendorf and Lindquist, 2002*; *Mogk et al., 2003*), we conclude that proteotoxicity suppression is an emergent property of hexameric Hsp104 homologs or chimeras.

## The NBD2:CTD unit of TtHsp104 and TlHsp104 contribute to suppression of αSyn toxicity

For the αSyn-specific Hsp104 homologs, TtHsp104 and TlHsp104, chimeras consisting of NTD:NBD1:MD from TtHsp104 or TlHsp104 fused to NBD2:CTD from ScHsp104 were unable to reduce αSyn toxicity (*Figure 9B,C*). To test whether suppression of αSyn toxicity is encoded by residues within NBD2:CTD, we tested chimeras in which NTD:NBD1:MD were from ScHsp104 and NBD2:CTD was from either TtHsp104 or TlHsp104. However, these chimeras were also incapable of suppressing αSyn toxicity (*Figure 9D,E*; see *Figure 8—figure supplement 1J,K* for accompanying Western blots). Thus, these results suggest that additional residues or contacts in the NBD2:CTD unit are necessary but not sufficient for TtHsp104 and TlHsp104 to suppress αSyn toxicity.

## Hsp104 chimeras display reduced thermotolerance

We next characterized the thermotolerance activity of all chimeras (*Figure 9—figure supplement 1*) to understand how perturbing cognate interdomain interactions affects the disaggregase activity of each chimera. All Hsp104 homologs except PeHsp104 have substantial thermotolerance activity (*Figure 5A–B* and *Figure 9—figure supplement 1*). However, only GsHsp104 and TtHsp104 have thermotolerance activities that are statistically indistinguishable from that of ScHsp104 (*Figure 9—figure supplement 1*). Chimeras generated by swapping the NTD alone generally retain thermotolerance activity at a level comparable to the homologs from which the NTD is derived. Interestingly, Cr$^{NTD}$Sc$^{NBD1:MD:NBD2:CTD}$ and Ct$^{NTD}$Sc$^{NBD1:MD:NBD2:CTD}$ displayed enhanced thermotolerance relative to CrHsp104 and CtHsp104 (*Figure 9—figure supplement 1*). Thus, whereas CrHsp104 and CtHsp104 are deficient in thermotolerance activity relative to ScHsp104, Cr$^{NTD}$Sc$^{NBD1:MD:NBD2:CTD}$

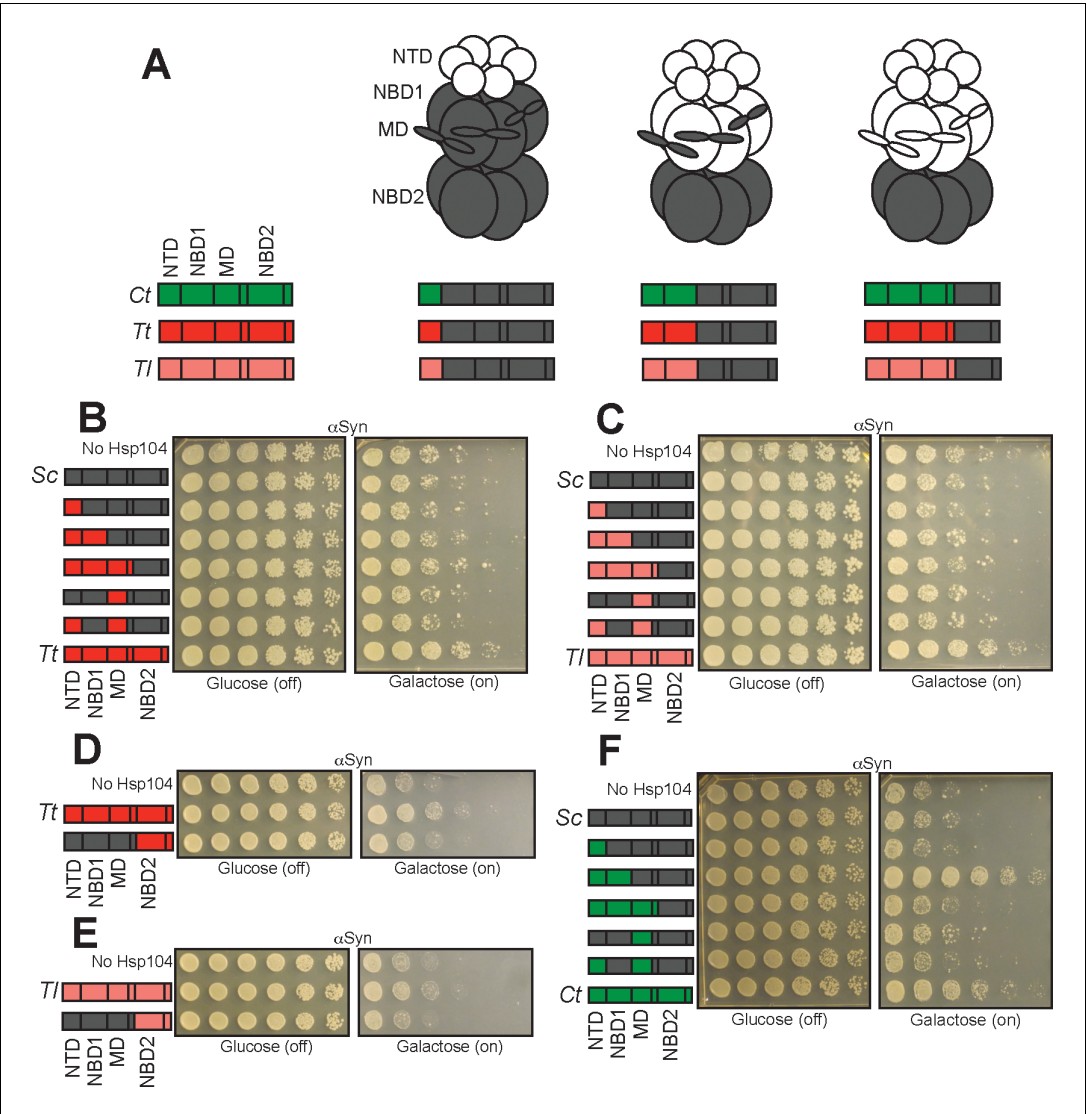

**Figure 9.** The NBD2:CTD unit of TtHsp104 and TlHsp104 contribute to suppression of αSyn toxicity. (**A**) Color codes and domain boundaries and labels of Hsp104 homologs. (**B**) Spotting assay of Δ*hsp104* strains coexpressing αSyn and the indicated chimeric Hsp104 between ScHsp104 and TtHsp104 illustrates that no chimeras between ScHsp104 and TtHsp104 replicate the αSyn toxicity-suppressing phenotype of TtHsp104. (**C**) Spotting assay of Δ*hsp104* strains coexpressing αSyn and the indicated chimeric Hsp104 between ScHsp104 and TlHsp104 illustrates that no chimeras between ScHsp104 and TlHsp104 replicate the αSyn toxicity-suppressing phenotype of TlHsp104. (**D, E**) Spotting assays of Δ*hsp104* strains coexpressing the indicated chimeric Hsp104 and αSyn illustrates that the NBD2:CTD unit from TtHsp104 (**D**) or TlHsp104 (**E**) is not sufficient to copy the αSyn toxicity-suppression phenotype of TtHsp104 or TlHsp104. (**F**) Spotting assay of Δ*hsp104* strains coexpressing the indicated chimeric Hsp104 and αSyn illustrates that chimeras possessing the NTD and NBD1 from CtHsp104 copies the αSyn toxicity-suppressing phenotype of CtHsp104$^{WT}$.

The online version of this article includes the following figure supplement(s) for figure 9:

**Figure supplement 1.** Thermotolerance activity of Hsp104 chimeras.

**Figure supplement 2.** Toxicity of select chimeras between CtHsp104 and ScHsp104.

performs significantly better than ScHsp104 and Ct$^{NTD}$Sc$^{NBD1:MD:NBD2:CTD}$ is indistinguishable from ScHsp104 (*Figure 9—figure supplement 1*). These results suggest that in some cases NTD swaps do not perturb Hsp104 activity. However, in other cases the NTD swap reduced Hsp104 activity as with Sr$^{NTD}$Sc$^{NBD1:MD:NBD2:CTD}$, which displayed minimal thermotolerance comparable to Δ*hsp104* cells (*Figure 9—figure supplement 1*).

Swaps of subsequent domains, either alone or in combination, impaired thermotolerance function (*Figure 9—figure supplement 1*). Some of the weak thermotolerance phenotypes may be attributable to low expression levels of certain chimeras from *pHSP104*. For instance, several chimeras between MbHsp104 and ScHsp104 are poorly expressed from *pHSP104* (*Figure 8—figure supplement 1C*). Nonetheless, expression levels of chimeras alone are insufficient to explain their thermotolerance phenotypes: even specific chimeras that reduce TDP-43 toxicity and express well from *pHSP104* (e.g. Cr$^{NTD:NBD1:MD}$Sc$^{NBD2:CTD}$) fail to confer thermotolerance (*Figure 9—figure supplement 1*). This observation provides further evidence that the mechanism that enables Hsp104 homologs to suppress TDP-43 toxicity is distinct from Hsp104 disaggregase activity.

## Non-cognate NTD:NBD1 units can yield toxic chimeras at 37°C

We next assessed the fitness effects intrinsic to the chimeras. We expressed all chimeras in Δ*hsp104* yeast at 37°C in the absence of any toxic substrate protein to observe any intrinsic toxicity associated with the chimeras themselves. We observed that NTD replacement alone does not cause toxicity (*Figure 8—figure supplement 3* and *Figure 9—figure supplement 2*, third row in all panels). By contrast, replacing both the NTD and NBD1 resulted in toxicity in several cases (e.g. for GsHsp104, CrHsp104, PeHsp104, SrHsp104, and CtHsp104; *Figure 8—figure supplement 3A,C,D,E* and *Figure 9—figure supplement 2A*, *boxed*), although TtHsp104 and TlHsp104 (αSyn-specific variants) were exceptions (*Figure 9—figure supplement 2B,C*). Interestingly, no toxicity was observed in chimeras where the NTD, NBD1, and MD were replaced together (*Figure 8—figure supplement 3* and *Figure 9—figure supplement 2*, fifth row), nor in chimeras where the MD was replaced alone (*Figure 8—figure supplement 3* and *Figure 9—figure supplement 2*, sixth row) or in combination with the NTD (*Figure 8—figure supplement 3* and *Figure 9—figure supplement 2*, seventh row). These findings suggest that altered non-cognate interactions between the transplanted NTD:NBD1 unit and the native ScMD can elicit off-target toxicity (*Sweeny et al., 2020*). Curiously, non-cognate interactions between a transplanted MD and the native ScNTD:NBD1 unit does not elicit that same effect. Thus, we suggest that the NTD:NBD1 unit plays a dominant role in regulating Hsp104.

## Select chimeras mimic potentiated ScHsp104 variants to suppress αSyn toxicity

Interestingly, two chimeras consisting of cognate NTD:NBD1 pairs fused to MD:NBD2:CTD from ScHsp104 unexpectedly suppressed αSyn toxicity. First, a chimera consisting of the CtHsp104 NTD and NBD1 fused to the ScHsp104 MD, NBD2, and CTD suppressed αSyn toxicity even more strongly than CtHsp104$^{WT}$ (*Figure 9F*). Similarly, a chimera consisting of the PeHsp104 NTD and NBD1 fused to the ScHsp104 MD, NBD2, and CTD reduced αSyn toxicity (*Figure 8—figure supplement 2D*). This finding was particularly unexpected because full-length PeHsp104 is specific to TDP-43, and this same chimera is inactive against TDP-43 (*Figure 8E*). Both of these chimeras were toxic to yeast when expressed alone at 37°C (*Figure 8—figure supplement 3D* and *Figure 9—figure supplement 2A*, boxed). Thus, in these cases, disruption of interaction between the transplanted NTD:NBD1 unit and the ScMD appears to mimic potentiated ScHsp104 variants and enables suppression of αSyn toxicity. Indeed, mutations in ScHsp104 NTD and NBD1 that disrupt interactions with the MD can potentiate activity (*Mack et al., 2020*; *Sweeny et al., 2020*; *Tariq et al., 2019*; *Ye et al., 2020*).

## Discussion

Here, we used yeast toxicity models to identify naturally occurring Hsp104 homologs, from diverse hosts, capable of buffering proteotoxicity of several proteins implicated in human neurodegenerative diseases. Among the prokaryotic homologs we tested, ClpB and hyperactive variants were inutile, whereas ClpG mitigated TDP-43, FUS, and αSyn toxicity. By contrast, eukaryotic Hsp104 homologs emerged that selectively suppressed TDP-43 or αSyn toxicity in yeast. Excitingly, eukaryotic Hsp104 homologs that selectively suppress TDP-43 toxicity in yeast also suppress TDP-43 aggregation in human cells. Likewise, eukaryotic Hsp104 homologs that selectively suppress αSyn toxicity in yeast also suppress αSyn toxicity-induced neurodegeneration in *C. elegans*. Thus, we suggest that, like previously-defined potentiated Hsp104 variants (*Jackrel et al., 2014*; *Mack et al., 2020*), these naturally-occurring Hsp104 variants may be able to mitigate proteotoxicity in a wide variety of circumstances, including in metazoan systems.

Several features of the eukaryotic Hsp104 homologs presented here contrast with potentiated Hsp104 variants. Eukaryotic Hsp104 homologs are substrate-specific and typically only suppress either TDP-43 or αSyn toxicity (the exception is CtHsp104). Indeed, we did not isolate any eukaryotic Hsp104 homologs capable of suppressing FUS toxicity. In contrast, potentiated Hsp104 variants typically are able to suppress toxicity of multiple toxic substrates (*Jackrel et al., 2014*; *Jackrel and Shorter, 2014*; *Jackrel et al., 2015*; *Tariq et al., 2019*; *Tariq et al., 2018*). Natural Hsp104 homologs also displayed no intrinsic toxicity (with the exception of PfHsp104), even when expressed at elevated temperatures. Thus, extant Hsp104 homologs have likely been filtered through natural selection to avoid deleterious and destabilizing sequences. In contrast, potentiated Hsp104 variants have predominantly been engineered by destabilizing the NBD1:MD interface (*Jackrel et al., 2014*; *Jackrel et al., 2015*; *Sweeny et al., 2020*; *Tariq et al., 2019*; *Tariq et al., 2018*; *Ye et al., 2020*). This difference is further reflected in fundamental mechanistic differences between how the eukaryotic Hsp104 homologs described here and potentiated Hsp104 variants operate to antagonize proteotoxic misfolding. Engineered Hsp104 variants are generally enhanced disaggregases (*Jackrel et al., 2014*; *Jackrel and Shorter, 2014*; *Jackrel et al., 2015*; *Tariq et al., 2018*). We hypothesize that in some cases these enhanced disaggregase and unfoldase activities may come at the cost of substrate specificity (e.g. by mistargeting natively-folded complexes for disassembly), analogous to trade-offs between speed and fidelity observed in other NTPase molecular machines, such as RNA polymerases (*Fitzsimmons et al., 2018*). However, the eukaryotic Hsp104 homologs presented here are not similarly enhanced disaggregases, and in fact do not require disaggregase activity to antagonize proteotoxic misfolding of TDP-43 or αSyn. Rather, they appear to act in an ATP-independent manner to inhibit protein aggregation and suppress toxicity of specific substrates.

The dual role of Hsp104 as a molecular chaperone capable of preventing aggregation in addition to a disaggregase capable of reversing protein aggregation has long been appreciated (*Shorter and Lindquist, 2004*; *Shorter and Lindquist, 2006*). Both of these activities require ATPase activity for maximum effect, although disaggregase activity is much more sensitive to reduced ATPase activity than inhibition of aggregation (*Shorter and Lindquist, 2004*; *Shorter and Lindquist, 2006*). Here, we find that diverse eukaryotic Hsp104 homologs suppress toxicity of select substrates in a manner that requires neither active ATPase domains nor substrate engagement by the canonical pore loops. Rather, we have uncovered genetic variation outside of these core AAA+ features that enables molecular recognition of specific substrates. Thus, in future work it will be important to tune this genetic variation to create designer disaggregases with highly specific molecular recognition. Indeed, we envision making modular disaggregases by combining potentiating mutations with molecular-recognition motifs to generate Hsp104 variants with enhanced specificity *and* disaggregase activity.

What sequence determinants of eukaryotic Hsp104 homologs enable their toxicity-suppression phenotypes? Hsp104 homologs that selectively suppress TDP-43 toxicity or selectively suppress αSyn toxicity are more similar to each other than between groups (average sequence identities of 56%, 76%, and 44%, respectively; *Supplementary file 2* and *Figure 2—figure supplement 2B*). To further address this question, we first tested the effect of deleting the NTD on the toxicity-suppression phenotypes associated with Hsp104 homologs (*Figure 7*). We observed that suppression of TDP-43 toxicity by MbHsp104 and CrHsp104, and suppression of αSyn toxicity by TtHsp104, depends on the presence of the NTD of these homologs (*Figure 7*). The ScHsp104 NTD enables many aspects of ScHsp104 function, including hexamer cooperativity, substrate binding, amyloid dissolution, and proteotoxicity suppression by potentiated Hsp104 variants (*Sweeny et al., 2015*; *Sweeny et al., 2020*). We suggest that the NTD-dependent rescue phenotypes we observed may reflect a role of the NTD in collaboration with other cognate domains to enable effective substrate engagement.

While the NTD is necessary for proteotoxicity suppression under some circumstances, it is not universally required. Three out of five TDP-43-specific homologs do not require the NTD to suppress TDP-43 toxicity, and TlHsp104 does not require the NTD to mitigate αSyn toxicity. Furthermore, simply replacing the ScHsp104 NTD with the NTD from MbHsp104 or CrHsp104 does not enable suppression of TDP-43 toxicity, nor does replacing the ScHsp104 NTD with the NTD from TtHsp104 impart suppression of αSyn toxicity. Thus, we reasoned that additional domains must be required. Indeed, for TDP-43, we found that replacing the NTD:NBD1:MD unit of the ScHsp104 hexamer with the homologous domains from any TDP-43-rescuing homolog enables suppression of TDP-43

toxicity by the resulting chimera. The NBD2 and CTD domains are required to facilitate suppression of toxicity, as fragments of PeHsp104 and SrHsp104 encompassing NTD:NBD1:MD but lacking NBD2 and the CTD fail to suppress toxicity. However, NBD2:CTD could be from ScHsp104. In contrast, NBD2:CTD could not be from ScHsp104 for suppression of αSyn toxicity by TtHsp104 and TlHsp104. We propose two explanations for these divergent observations. The first is that, particularly for the homologs where NTD deletion does not diminish suppression of toxicity (e.g. GsHsp104, PeHsp104, and SrHsp104 for TDP-43 and TlHsp104 for αSyn), there are important sequence determinants throughout the other domains of the protein that enable each homolog to produce a protective phenotype. The second possible explanation is that specific domains contribute to suppression of toxicity indirectly by stabilizing the overall protein architecture of the chimeras. There are no obvious trends in the primary sequence of the homologs, however, that would clarify these two competing possibilities. Indeed, further work is needed to gain a complete understanding of how homologous Hsp104 sequences confer variable substrate-specific proteotoxicity suppression.

That nearly half of the eukaryotic Hsp104 homologs we tested (8/17) were able to antagonize toxicity of either TDP-43 or αSyn (or both, in the case of CtHsp104) was unanticipated. There are no clear TDP-43 or αSyn homologs in the host species from which we selected Hsp104 homologs to test. Our observation that, for a given Hsp104 homolog, there was not a predictable relationship between suppression of TDP-43 or αSyn toxicity and thermotolerance, which may more closely reflect the primary in vivo function of these Hsp104 homologs, suggests one interpretation involving cryptic genetic variation (*Paaby and Rockman, 2014*). Variation that is neutral or nearly neutral with respect to thermotolerance function, but which enables suppression of TDP-43 or αSyn, may have accumulated in these lineages in the absence of selective pressure and was only revealed by our synthetic experimental paradigm. Alternatively, it is possible that the Hsp104 homologs we tested are adapted to host proteomes that have higher-than-average content of TDP-43-like or αSyn-like motifs, which is reflected in their ability to selectively antagonize misfolding of similar motifs in TDP-43 or αSyn. Nevertheless, the fact that Hsp104 homologs that are active against at least one of TDP-43 or αSyn were so widespread among diverse eukaryotic lineages, and the fact that we have now identified ~100 potentiating mutations in Hsp104 that enable similar toxicity-suppression phenotypes (albeit by different mechanisms) suggests that Hsp104, and possibly other AAA+ proteins (e.g. Skd3, VCP), are uniquely poised to buffer deleterious protein misfolding and aggregation (*Cupo and Shorter, 2020*; *Darwich et al., 2020*; *March et al., 2019*; *Wang et al., 2019*).

## Materials and methods

### Key resources table

| Reagent type (species) or resource | Designation | Source or reference | Identifiers | Additional information |
|---|---|---|---|---|
| Strain (*Saccharomyces cerevisiae*) | W303a (*MATa; can1-100; his3-11,15; leu2-3,112; trp1-1; ura3-1; ade2-1*) | *Schirmer et al., 2004* | N/A | |
| Strain (*S. cerevisiae*) | W303aΔ hsp104 (*MATa; can1-100; his3-11,15; leu2-3,112; trp1-1; ura3-1; ade2-1; hsp104::KanMX*) | *Schirmer et al., 2004* | A3224 | |
| Strain (*Escherichia coli*) | BL21-CodonPlus (DE3)-RIL | Agilent | 2302545 | |
| Strain (*Caenorhabditis elegans*) | UA44 (*baIn11* [$P_{dat-1}$: :α-syn, $P_{dat-1}$: :GFP]) | *Cao et al., 2005* | UA44 | Full description can be found in Materials and methods: Generation of transgenic *C. elegans* and neurodegeneration analysis |
| Strain (*C. elegans*) | UA381 (*baIn11* [$P_{dat-1}$: :α-syn, $P_{dat-1}$: :GFP]; *baEx210* [$P_{dat-1}$:: CtHsp104, rol-6]) | This paper | UA381 | Full description can be found in Materials and methods: Generation of transgenic *C. elegans* and neurodegeneration analysis |

*Continued on next page*

*Continued*

| Reagent type (species) or resource | Designation | Source or reference | Identifiers | Additional information |
|---|---|---|---|---|
| Strain (*C. elegans*) | UA382 (*baln11* [P*~dat-1~*: :α-syn, P*~dat-1~*: :GFP]; *baEx211* [P*~dat-1~*:: *TtHsp104*, *rol-6*]) | This paper | UA382 | Full description can be found in Materials and methods: Generation of transgenic *C. elegans* and neurodegeneration analysis |
| Strain (*C. elegans*) | UA383 (*baln11* [P*~dat-1~*: :α-syn, P*~dat-1~*: :GFP]; *baEx212* [P*~dat-1~*:: *TlHsp104*, *rol-6*]) | This paper | UA383 | Full description can be found in Materials and methods: Generation of transgenic *C. elegans* and neurodegeneration analysis |
| Strain (*C. elegans*) | UA403 (*vtIs7* [P*~dat-1~*::GFP]; *baEx223* [P*~dat-1~*::*CtHSP104*, *rol-6*]) | This paper | UA403 | Full description can be found in Materials and methods: Generation of transgenic *C. elegans* and neurodegeneration analysis |
| Strain (*C. elegans*) | UA404 (*vtIs7* [P*~dat-1~*::GFP]; *baEx224* [P*~dat-1~*:: *TtHSP104*, *rol-6*]) | This paper | UA404 | Full description can be found in Materials and methods: Generation of transgenic *C. elegans* and neurodegeneration analysis |
| Strain (*C. elegans*) | UA405 (*vtIs7* [P*~dat-1~*::GFP]; *baEx225* [P*~dat-1~*:: *TlHSP104*, *rol-6*]) | This paper | UA405 | Full description can be found in Materials and methods: Generation of transgenic *C. elegans* and neurodegeneration analysis |
| Cell line (*Homo sapiens*) | HEK293T | ATCC | Cat# CRL-3216 RRID:CVCL_0063 | |
| Antibody | Mouse monoclonal anti-FLAG M2 | Sigma-Aldrich | Cat# F1804; RRID:AB_262044 | (1:1000 dilution) |
| Antibody | Rabbit polyclonal anti-TDP-43 | Proteintech | Cat#10782; RRID:AB_615042 | (1:1000 dilution) |
| Antibody | Rabbit polyclonal anti-GFP | Sigma-Aldrich | Cat# G1544; RRID:AB_439690 | (1:2500 dilution) |
| Antibody | Mouse monoclonal anti-3-phosphoglycerate kinase | Novex | Cat# 459250; RRID:AB_221541 | (1:1000 dilution) |
| Antibody | Rat monoclonal anti-tubulin | Abcam | Cat# ab6160; RRID:AB_305328 | (1:1000 dilution) |
| Antibody | IRDye 680RD Goat anti-Rabbit IgG secondary antibody | Li-Cor | Cat# 926–68071; RRID:AB_10956166 | (1:2500 dilution) |
| Antibody | IRDye 800CW Goat anti-Mouse IgG secondary antibody | Li-Cor | Cat# 926–32210; RRID:AB_621842 | (1:5000 dilution) |
| Antibody | IRDye 800CW Goat anti-Rat IgG secondary antibody | Li-Cor | Cat# 926–32219; RRID:AB_1850025 | (1:2500 dilution) |
| Recombinant DNA reagent | pAG416GAL-ccdB | *Alberti et al., 2007* | N/A | |
| Recombinant DNA reagent | pRS313HSE-ccdB | *Gates et al., 2017* | N/A | |
| Recombinant DNA reagent | pMCSG | *Kim et al., 2011* | N/A | |
| Recombinant DNA reagent | pDAT-ccdB | *Jackrel et al., 2014* | N/A | |
| Recombinant DNA reagent | pInducer20-ccdB | *Meerbrey et al., 2011* | N/A | |
| Recombinant DNA reagent | pE-SUMO | Lifesensors | N/A | |
| Recombinant DNA reagent | pAG416GAL-ScHsp104-FLAG | *Michalska et al., 2019* | N/A | |
| Recombinant DNA reagent | pRS313HSE-ScHsp104-FLAG | *Michalska et al., 2019* | N/A | |
| Recombinant DNA reagent | pNOTAG-ScHsp104 | *Jackrel et al., 2014* | N/A | |

*Continued on next page*

*Continued*

| Reagent type (species) or resource | Designation | Source or reference | Identifiers | Additional information |
|---|---|---|---|---|
| Recombinant DNA reagent | pAG416GAL-ScHsp104$^{A503V}$-FLAG | *Michalska et al., 2019* | N/A | |
| Recombinant DNA reagent | pAG416GAL-ScHsp104$^{A503S}$-FLAG | *Michalska et al., 2019* | N/A | |
| Recombinant DNA reagent | pNOTAG-ScHsp104$^{A503S}$ | *Jackrel et al., 2014* | N/A | |
| Recombinant DNA reagent | pMCSG-CtHsp104 | *Michalska et al., 2019* | N/A | |
| Recombinant DNA reagent | pAG416GAL-CtHsp104-FLAG | *Michalska et al., 2019* | N/A | |
| Recombinant DNA reagent | pRS313HSE-CtHsp104-FLAG | *Michalska et al., 2019* | N/A | |
| Recombinant DNA reagent | pDAT-CtHsp104 | This paper | N/A | Full description can be found in Materials and methods: Plasmids |
| Recombinant DNA reagent | pInducer20-CtHsp104-FLAG | This paper | N/A | Full description can be found in Materials and methods: Plasmids |
| Recombinant DNA reagent | pAG416GAL-CtHsp104$^{DN}$-FLAG | This paper | N/A | Encodes CtHsp104$^{158-882}$; full description can be found in Materials and methods: Plasmids |
| Recombinant DNA reagent | pRS313HSE-CtHsp104$^{DN}$-FLAG | This paper | N/A | Encodes CtHsp104$^{158-882}$; full description can be found in Materials and methods: Plasmids |
| Recombinant DNA reagent | pAG416GAL-CtHsp104$^{DWA(KA)}$-FLAG | This paper | N/A | Encodes CtHsp104$^{K211A:K612A}$; full description can be found in Materials and methods: Plasmids |
| Recombinant DNA reagent | pRS313HSE-CtHsp104$^{DWA(KA)}$ FLAG | This paper | N/A | Encodes CtHsp104$^{K211A:K612A}$; full description can be found in Materials and methods: Plasmids |
| Recombinant DNA reagent | pAG416GAL-CtHsp104$^{DWA(KT)}$-FLAG | This paper | N/A | Encodes CtHsp104$^{K211T:K612T}$; full description can be found in Materials and methods: Plasmids |
| Recombinant DNA reagent | pRS313HSE-CtHsp104$^{DWA(KT)}$ FLAG | This paper | N/A | Encodes CtHsp104$^{K211T:K612T}$; full description can be found in Materials and methods: Plasmids |
| Recombinant DNA reagent | pAG416GAL-CtHsp104$^{DPLA}$-FLAG | This paper | N/A | CtHsp104$^{Y249A:Y654A}$; full description can be found in Materials and methods: Plasmids |
| Recombinant DNA reagent | pRS313HSE-CtHsp104$^{DPLA}$-FLAG | This paper | N/A | CtHsp104$^{Y249A:Y654A}$; full description can be found in Materials and methods: Plasmids |
| Recombinant DNA reagent | pAG416GAL-CtHsp104$^{DWB(EA)}$-FLAG | This paper | N/A | CtHsp104$^{E275A:E679A}$; full description can be found in Materials and methods: Plasmids |
| Recombinant DNA reagent | pRS313HSE-CtHsp104$^{DWB(EA)}$ FLAG | This paper | N/A | CtHsp104$^{E275A:E679A}$; full description can be found in Materials and methods: Plasmids |
| Recombinant DNA reagent | pAG416GAL-CtHsp104$^{DWB(EQ)}$-FLAG | This paper | N/A | CtHsp104$^{E275Q:E679Q}$; full description can be found in Materials and methods: Plasmids |
| Recombinant DNA reagent | pRS313HSE-CtHsp104$^{DWB(EQ)}$-FLAG | This paper | N/A | CtHsp104$^{E275Q:E679Q}$; full description can be found in Materials and methods: Plasmids |
| Recombinant DNA reagent | pAG416GAL-CaSSS-FLAG | This paper | N/A | Chimera sequence available in *Supplementary file 2*; full description can be found in Materials and methods: Plasmids |
| Recombinant DNA reagent | pRS313HSE-CaSSS-FLAG | This paper | N/A | Chimera sequence available in *Supplementary file 2*; full description can be found in Materials and methods: Plasmids |

*Continued on next page*

*Continued*

| Reagent type (species) or resource | Designation | Source or reference | Identifiers | Additional information |
|---|---|---|---|---|
| Recombinant DNA reagent | pAG416GAL-CaCaSS-FLAG | This paper | N/A | Chimera sequence available in *Supplementary file 2*; full description can be found in Materials and methods: Plasmids |
| Recombinant DNA reagent | pRS313HSE-CaCaSS-FLAG | This paper | N/A | Chimera sequence available in *Supplementary file 2*; full description can be found in Materials and methods: Plasmids |
| Recombinant DNA reagent | pAG416GAL-CaCaCaS-FLAG | This paper | N/A | Chimera sequence available in *Supplementary file 2*; full description can be found in Materials and methods: Plasmids |
| Recombinant DNA reagent | pRS313HSE-CaCaCaS-FLAG | This paper | N/A | Chimera sequence available in *Supplementary file 2*; full description can be found in Materials and methods: Plasmids |
| Recombinant DNA reagent | pAG416GAL-SSCaS-FLAG | This paper | N/A | Chimera sequence available in *Supplementary file 2*; full description can be found in Materials and methods: Plasmids |
| Recombinant DNA reagent | pRS313HSE-SSCaS-FLAG | This paper | N/A | Chimera sequence available in *Supplementary file 2*; full description can be found in Materials and methods: Plasmids |
| Recombinant DNA reagent | pAG416GAL-CaSCaS-FLAG | This paper | N/A | Chimera sequence available in *Supplementary file 2*; full description can be found in Materials and methods: Plasmids |
| Recombinant DNA reagent | pRS313HSE-CaSCaS-FLAG | This paper | N/A | Chimera sequence available in *Supplementary file 2*; full description can be found in Materials and methods: Plasmids |
| Recombinant DNA reagent | pAG416GAL-GsHsp104-FLAG | This paper | N/A | Full description can be found in Materials and methods: Plasmids |
| Recombinant DNA reagent | pRS313HSE-GsHsp104-FLAG | This paper | N/A | Full description can be found in Materials and methods: Plasmids |
| Recombinant DNA reagent | pAG416GAL-GsHsp104$^{DN}$-FLAG | This paper | N/A | Encodes GsHsp104$^{158-922}$; full description can be found in Materials and methods: Plasmids |
| Recombinant DNA reagent | pRS313HSE-GsHsp104$^{DN}$-FLAG | This paper | N/A | Encodes GsHsp104$^{158-922}$; full description can be found in Materials and methods: Plasmids |
| Recombinant DNA reagent | pAG416GAL-GsHsp104$^{DWA(KA)}$-FLAG | This paper | N/A | Encodes GsHsp104$^{K211A:K621A}$; full description can be found in Materials and methods: Plasmids |
| Recombinant DNA reagent | pRS313HSE-GsHsp104$^{DWA(KA)}$ FLAG | This paper | N/A | Encodes GsHsp104$^{K211A:K621A}$; full description can be found in Materials and methods: Plasmids |
| Recombinant DNA reagent | pAG416GAL-GsHsp104$^{DWA(KT)}$-FLAG | This paper | N/A | Encodes GsHsp104$^{K211T:K621T}$; full description can be found in Materials and methods: Plasmids |
| Recombinant DNA reagent | pRS313HSE-GsHsp104$^{DWA(KT)}$ FLAG | This paper | N/A | Encodes GsHsp104$^{K211T:K621T}$; full description can be found in Materials and methods: Plasmids |
| Recombinant DNA reagent | pAG416GAL-GsHsp104$^{DPLA}$-FLAG | This paper | N/A | Encodes GsHsp104$^{Y249A:Y663A}$; full description can be found in Materials and methods: Plasmids |

*Continued on next page*

*Continued*

| Reagent type (species) or resource | Designation | Source or reference | Identifiers | Additional information |
|---|---|---|---|---|
| Recombinant DNA reagent | pRS313HSE-GsHsp104$^{DPLA}$-FLAG | This paper | N/A | Encodes GsHsp104$^{Y249A:Y663A}$; full description can be found in Materials and methods: Plasmids |
| Recombinant DNA reagent | pAG416GAL-GsHsp104$^{DWB(EA)}$-FLAG | This paper | N/A | Encodes GsHsp104$^{E277A:E688A}$; full description can be found in Materials and methods: Plasmids |
| Recombinant DNA reagent | pRS313HSE-GsHsp104$^{DWB(EA)}$ FLAG | This paper | N/A | Encodes GsHsp104$^{E277A:E688A}$; full description can be found in Materials and methods: Plasmids |
| Recombinant DNA reagent | pAG416GAL-GsHsp104$^{DWB(EQ)}$-FLAG | This paper | N/A | Encodes GsHsp104$^{E277Q:E688Q}$; full description can be found in Materials and methods: Plasmids |
| Recombinant DNA reagent | pRS313HSE-GsHsp104$^{DWB(EQ)}$-FLAG | This paper | N/A | Encodes GsHsp104$^{E277Q:E688Q}$; full description can be found in Materials and methods: Plasmids |
| Recombinant DNA reagent | pAG416GAL-GSSS-FLAG | This paper | N/A | Chimera sequence available in *Supplementary file 2*; full description can be found in Materials and methods: Plasmids |
| Recombinant DNA reagent | pRS313HSE-GSSS-FLAG | This paper | N/A | Chimera sequence available in *Supplementary file 2*; full description can be found in Materials and methods: Plasmids |
| Recombinant DNA reagent | pAG416GAL-GGSS-FLAG | This paper | N/A | Chimera sequence available in *Supplementary file 2*; full description can be found in Materials and methods: Plasmids |
| Recombinant DNA reagent | pRS313HSE-GGSS-FLAG | This paper | N/A | Chimera sequence available in *Supplementary file 2*; full description can be found in Materials and methods: Plasmids |
| Recombinant DNA reagent | pAG416GAL-GGGS-FLAG | This paper | N/A | Chimera sequence available in *Supplementary file 2*; full description can be found in Materials and methods: Plasmids |
| Recombinant DNA reagent | pRS313HSE-GGGS-FLAG | This paper | N/A | Chimera sequence available in *Supplementary file 2*; full description can be found in Materials and methods: Plasmids |
| Recombinant DNA reagent | pAG416GAL-SSGS-FLAG | This paper | N/A | Chimera sequence available in *Supplementary file 2*; full description can be found in Materials and methods: Plasmids |
| Recombinant DNA reagent | pRS313HSE-SSGS-FLAG | This paper | N/A | Chimera sequence available in *Supplementary file 2*; full description can be found in Materials and methods: Plasmids |
| Recombinant DNA reagent | pAG416GAL-GSGS-FLAG | This paper | N/A | Chimera sequence available in *Supplementary file 2*; full description can be found in Materials and methods: Plasmids |
| Recombinant DNA reagent | pRS313HSE-GSGS-FLAG | This paper | N/A | Chimera sequence available in *Supplementary file 2*; full description can be found in Materials and methods: Plasmids |
| Recombinant DNA reagent | pNOTAG-MbHsp104 | This paper | N/A | Full description can be found in Materials and methods: Plasmids |
| Recombinant DNA reagent | pAG416GAL-MbHsp104-FLAG | This paper | N/A | Full description can be found in Materials and methods: Plasmids |

*Continued on next page*

*Continued*

| Reagent type (species) or resource | Designation | Source or reference | Identifiers | Additional information |
|---|---|---|---|---|
| Recombinant DNA reagent | pRS313HSE-MbHsp104-FLAG | This paper | N/A | Full description can be found in Materials and methods: Plasmids |
| Recombinant DNA reagent | pAG416GAL-MbHsp104$^{DN}$-FLAG | This paper | N/A | Encodes MbHsp104$^{160-889}$; full description can be found in Materials and methods: Plasmids |
| Recombinant DNA reagent | pRS313HSE-MbHsp104$^{DN}$-FLAG | This paper | N/A | Encodes MbHsp104$^{160-889}$; full description can be found in Materials and methods: Plasmids |
| Recombinant DNA reagent | pAG416GAL-MbHsp104$^{DWA(KA)}$-FLAG | This paper | N/A | Encodes MbHsp104$^{K213A:K623A}$; full description can be found in Materials and methods: Plasmids |
| Recombinant DNA reagent | pRS313HSE-MbHsp104$^{DWA(KA)}$ FLAG | This paper | N/A | Encodes MbHsp104$^{K213A:K623A}$; full description can be found in Materials and methods: Plasmids |
| Recombinant DNA reagent | pAG416GAL-MbHsp104$^{DWA(KT)}$-FLAG | This paper | N/A | Encodes MbHsp104$^{K213T:K623T}$; full description can be found in Materials and methods: Plasmids |
| Recombinant DNA reagent | pRS313HSE-MbHsp104$^{DWA(KT)}$ FLAG | This paper | N/A | Encodes MbHsp104$^{K213T:K623T}$; full description can be found in Materials and methods: Plasmids |
| Recombinant DNA reagent | pAG416GAL-MbHsp104$^{DPLA}$-FLAG | This paper | N/A | Encodes MbHsp104$^{Y251A:Y665A}$; full description can be found in Materials and methods: Plasmids |
| Recombinant DNA reagent | pRS313HSE-MbHsp104$^{DPLA}$-FLAG | This paper | N/A | Encodes MbHsp104$^{Y251A:Y665A}$; full description can be found in Materials and methods: Plasmids |
| Recombinant DNA reagent | pAG416GAL-MbHsp104$^{DWB(EA)}$-FLAG | This paper | N/A | Encodes MbHsp104$^{E279A:E690A}$; full description can be found in Materials and methods: Plasmids |
| Recombinant DNA reagent | pRS313HSE-MbHsp104$^{DWB(EA)}$ FLAG | This paper | N/A | Encodes MbHsp104$^{E279A:E690A}$; full description can be found in Materials and methods: Plasmids |
| Recombinant DNA reagent | pAG416GAL-MbHsp104$^{DWB(EQ)}$-FLAG | This paper | N/A | Encodes MbHsp104$^{E279Q:E690Q}$; full description can be found in Materials and methods: Plasmids |
| Recombinant DNA reagent | pRS313HSE-MbHsp104$^{DWB(EQ)}$-FLAG | This paper | N/A | Encodes MbHsp104$^{E279Q:E690Q}$; full description can be found in Materials and methods: Plasmids |
| Recombinant DNA reagent | pAG416GAL-MSSS-FLAG | This paper | N/A | Chimera sequence available in *Supplementary file 2*; full description can be found in Materials and methods: Plasmids |
| Recombinant DNA reagent | pRS313HSE-MSSS-FLAG | This paper | N/A | Chimera sequence available in *Supplementary file 2*; full description can be found in Materials and methods: Plasmids |
| Recombinant DNA reagent | pAG416GAL-MMSS-FLAG | This paper | N/A | Chimera sequence available in *Supplementary file 2*; full description can be found in Materials and methods: Plasmids |
| Recombinant DNA reagent | pRS313HSE-MMSS-FLAG | This paper | N/A | Chimera sequence available in *Supplementary file 2*; full description can be found in Materials and methods: Plasmids |
| Recombinant DNA reagent | pAG416GAL-MMMS-FLAG | This paper | N/A | Chimera sequence available in *Supplementary file 2*; full description can be found in Materials and methods: Plasmids |

*Continued*

| Reagent type (species) or resource | Designation | Source or reference | Identifiers | Additional information |
|---|---|---|---|---|
| Recombinant DNA reagent | pRS313HSE-MMMS-FLAG | This paper | N/A | Chimera sequence available in *Supplementary file 2*; full description can be found in Materials and methods: Plasmids |
| Recombinant DNA reagent | pAG416GAL-SSMS-FLAG | This paper | N/A | Chimera sequence available in *Supplementary file 2*; full description can be found in Materials and methods: Plasmids |
| Recombinant DNA reagent | pRS313HSE-SSMS-FLAG | This paper | N/A | Chimera sequence available in *Supplementary file 2*; full description can be found in Materials and methods: Plasmids |
| Recombinant DNA reagent | pAG416GAL-MSMS-FLAG | This paper | N/A | Chimera sequence available in *Supplementary file 2*; full description can be found in Materials and methods: Plasmids |
| Recombinant DNA reagent | pRS313HSE-MSMS-FLAG | This paper | N/A | Chimera sequence available in *Supplementary file 2*; full description can be found in Materials and methods: Plasmids |
| Recombinant DNA reagent | pNOTAG-CrHsp104 | This paper | N/A | Full description can be found in Materials and methods: Plasmids |
| Recombinant DNA reagent | pAG416GAL-CrHsp104-FLAG | This paper | N/A | Full description can be found in Materials and methods: Plasmids |
| Recombinant DNA reagent | pRS313HSE-CrHsp104-FLAG | This paper | N/A | Full description can be found in Materials and methods: Plasmids |
| Recombinant DNA reagent | pAG416GAL-CrHsp104$^{DN}$-FLAG | This paper | N/A | Encodes CrHsp104$^{165-925}$; full description can be found in Materials and methods: Plasmids |
| Recombinant DNA reagent | pRS313HSE-CrHsp104$^{DN}$-FLAG | This paper | N/A | Encodes CrHsp104$^{165-925}$; full description can be found in Materials and methods: Plasmids |
| Recombinant DNA reagent | pAG416GAL-CrHsp104$^{DWA(KA)}$-FLAG | This paper | N/A | Encodes CrHsp104$^{K216A:K614A}$; full description can be found in Materials and methods: Plasmids |
| Recombinant DNA reagent | pRS313HSE-CrHsp104$^{DWA(KA)}$ FLAG | This paper | N/A | Encodes CrHsp104$^{K216A:K614A}$; full description can be found in Materials and methods: Plasmids |
| Recombinant DNA reagent | pAG416GAL-CrHsp104$^{DWA(KT)}$-FLAG | This paper | N/A | Encodes CrHsp104$^{K216T:K614T}$; full description can be found in Materials and methods: Plasmids |
| Recombinant DNA reagent | pRS313HSE-CrHsp104$^{DWA(KT)}$ FLAG | This paper | N/A | Encodes CrHsp104$^{K216T:K614T}$; full description can be found in Materials and methods: Plasmids |
| Recombinant DNA reagent | pAG416GAL-CrHsp104$^{DPLA}$-FLAG | This paper | N/A | Encodes CrHsp104$^{Y255A:Y656A}$; full description can be found in Materials and methods: Plasmids |
| Recombinant DNA reagent | pRS313HSE-CrHsp104$^{DPLA}$-FLAG | This paper | N/A | Encodes CrHsp104$^{Y255A:Y656A}$; full description can be found in Materials and methods: Plasmids |
| Recombinant DNA reagent | pAG416GAL-CrHsp104$^{DWB(EA)}$-FLAG | This paper | N/A | Encodes CrHsp104$^{E283A:E681A}$; full description can be found in Materials and methods: Plasmids |
| Recombinant DNA reagent | pRS313HSE-CrHsp104$^{DWB(EA)}$ FLAG | This paper | N/A | Encodes CrHsp104$^{E283A:E681A}$; full description can be found in Materials and methods: Plasmids |

*Continued on next page*

*Continued*

| Reagent type (species) or resource | Designation | Source or reference | Identifiers | Additional information |
|---|---|---|---|---|
| Recombinant DNA reagent | pAG416GAL-CrHsp104$^{DWB(EQ)}$-FLAG | This paper | N/A | Encodes CrHsp104$^{E283Q:E681Q}$; full description can be found in Materials and methods: Plasmids |
| Recombinant DNA reagent | pRS313HSE-CrHsp104$^{DWB(EQ)}$-FLAG | This paper | N/A | Encodes CrHsp104$^{E283Q:E681Q}$; full description can be found in Materials and methods: Plasmids |
| Recombinant DNA reagent | pAG416GAL-CSSS-FLAG | This paper | N/A | Chimera sequence available in *Supplementary file 2*; full description can be found in Materials and methods: Plasmids |
| Recombinant DNA reagent | pRS313HSE-CSSS-FLAG | This paper | N/A | Chimera sequence available in *Supplementary file 2*; full description can be found in Materials and methods: Plasmids |
| Recombinant DNA reagent | pAG416GAL-CCSS-FLAG | This paper | N/A | Chimera sequence available in *Supplementary file 2*; full description can be found in Materials and methods: Plasmids |
| Recombinant DNA reagent | pRS313HSE-CCSS-FLAG | This paper | N/A | Chimera sequence available in *Supplementary file 2*; full description can be found in Materials and methods: Plasmids |
| Recombinant DNA reagent | pAG416GAL-CCCS-FLAG | This paper | N/A | Chimera sequence available in *Supplementary file 2*; full description can be found in Materials and methods: Plasmids |
| Recombinant DNA reagent | pRS313HSE-CCCS-FLAG | This paper | N/A | Chimera sequence available in *Supplementary file 2*; full description can be found in Materials and methods: Plasmids |
| Recombinant DNA reagent | pAG416GAL-SSCS-FLAG | This paper | N/A | Chimera sequence available in *Supplementary file 2*; full description can be found in Materials and methods: Plasmids |
| Recombinant DNA reagent | pRS313HSE-SSCS-FLAG | This paper | N/A | Chimera sequence available in *Supplementary file 2*; full description can be found in Materials and methods: Plasmids |
| Recombinant DNA reagent | pAG416GAL-CSCS-FLAG | This paper | N/A | Chimera sequence available in *Supplementary file 2*; full description can be found in Materials and methods: Plasmids |
| Recombinant DNA reagent | pRS313HSE-CSCS-FLAG | This paper | N/A | Chimera sequence available in *Supplementary file 2*; full description can be found in Materials and methods: Plasmids |
| Recombinant DNA reagent | pAG416GAL-PeHsp104-FLAG | This paper | N/A | Full description can be found in Materials and methods: Plasmids |
| Recombinant DNA reagent | pRS313HSE-PeHsp104-FLAG | This paper | N/A | Full description can be found in Materials and methods: Plasmids |
| Recombinant DNA reagent | pInducer20-PeHsp104-FLAG | This paper | N/A | Full description can be found in Materials and methods: Plasmids |
| Recombinant DNA reagent | pAG416GAL-PeHsp104$^{DN}$-FLAG | This paper | N/A | Encodes PeHsp104$^{163-914}$; full description can be found in Materials and methods: Plasmids |
| Recombinant DNA reagent | pRS313HSE-PeHsp104$^{DN}$-FLAG | This paper | N/A | Encodes PeHsp104$^{163-914}$; full description can be found in Materials and methods: Plasmids |

*Continued on next page*

Continued

| Reagent type (species) or resource | Designation | Source or reference | Identifiers | Additional information |
|---|---|---|---|---|
| Recombinant DNA reagent | pAG416GAL-PeHsp104$^{DWA(KA)}$-FLAG | This paper | N/A | Encodes PeHsp104$^{K214A:K613A}$; full description can be found in Materials and methods: Plasmids |
| Recombinant DNA reagent | pRS313HSE-PeHsp104$^{DWA(KA)}$ FLAG | This paper | N/A | Encodes PeHsp104$^{K214A:K613A}$; full description can be found in Materials and methods: Plasmids |
| Recombinant DNA reagent | pAG416GAL-PeHsp104$^{DWA(KT)}$-FLAG | This paper | N/A | Encodes PeHsp104$^{K214T:K613T}$; full description can be found in Materials and methods: Plasmids |
| Recombinant DNA reagent | pRS313HSE-PeHsp104$^{DWA(KT)}$ FLAG | This paper | N/A | Encodes PeHsp104$^{K214T:K613T}$; full description can be found in Materials and methods: Plasmids |
| Recombinant DNA reagent | pAG416GAL-PeHsp104$^{DPLA}$-FLAG | This paper | N/A | Encodes PeHsp104$^{Y253A:Y655A}$; full description can be found in Materials and methods: Plasmids |
| Recombinant DNA reagent | pRS313HSE-PeHsp104$^{DPLA}$-FLAG | This paper | N/A | Encodes PeHsp104$^{Y253A:Y655A}$; full description can be found in Materials and methods: Plasmids |
| Recombinant DNA reagent | pAG416GAL-PeHsp104$^{DWB(EA)}$-FLAG | This paper | N/A | Encodes PeHsp104$^{E281A:E680A}$; full description can be found in Materials and methods: Plasmids |
| Recombinant DNA reagent | pRS313HSE-PeHsp104$^{DWB(EA)}$ FLAG | This paper | N/A | Encodes PeHsp104$^{E281A:E680A}$; full description can be found in Materials and methods: Plasmids |
| Recombinant DNA reagent | pAG416GAL-PeHsp104$^{DWB(EQ)}$-FLAG | This paper | N/A | Encodes PeHsp104$^{E281Q:E680Q}$; full description can be found in Materials and methods: Plasmids |
| Recombinant DNA reagent | pRS313HSE-PeHsp104$^{DWB(EQ)}$-FLAG | This paper | N/A | Encodes PeHsp104$^{E281Q:E680Q}$; full description can be found in Materials and methods: Plasmids |
| Recombinant DNA reagent | pAG416GAL-PSSS-FLAG | This paper | N/A | Chimera sequence available in *Supplementary file 2*; full description can be found in Materials and methods: Plasmids |
| Recombinant DNA reagent | pRS313HSE-PSSS-FLAG | This paper | N/A | Chimera sequence available in *Supplementary file 2*; full description can be found in Materials and methods: Plasmids |
| Recombinant DNA reagent | pAG416GAL-PPSS-FLAG | This paper | N/A | Chimera sequence available in *Supplementary file 2*; full description can be found in Materials and methods: Plasmids |
| Recombinant DNA reagent | pRS313HSE-PPSS-FLAG | This paper | N/A | Chimera sequence available in *Supplementary file 2*; full description can be found in Materials and methods: Plasmids |
| Recombinant DNA reagent | pAG416GAL-PPPS-FLAG | This paper | N/A | Chimera sequence available in *Supplementary file 2*; full description can be found in Materials and methods: Plasmids |
| Recombinant DNA reagent | pRS313HSE-PPPS-FLAG | This paper | N/A | Chimera sequence available in *Supplementary file 2*; full description can be found in Materials and methods: Plasmids |
| Recombinant DNA reagent | pAG416GAL-SSPS-FLAG | This paper | N/A | Chimera sequence available in *Supplementary file 2*; full description can be found in Materials and methods: Plasmids |

*Continued on next page*

Continued

| Reagent type (species) or resource | Designation | Source or reference | Identifiers | Additional information |
|---|---|---|---|---|
| Recombinant DNA reagent | pRS313HSE-SSPS-FLAG | This paper | N/A | Chimera sequence available in *Supplementary file 2*; full description can be found in Materials and methods: Plasmids |
| Recombinant DNA reagent | pAG416GAL-PSPS-FLAG | This paper | N/A | Chimera sequence available in *Supplementary file 2*; full description can be found in Materials and methods: Plasmids |
| Recombinant DNA reagent | pRS313HSE-PSPS-FLAG | This paper | N/A | Chimera sequence available in *Supplementary file 2*; full description can be found in Materials and methods: Plasmids |
| Recombinant DNA reagent | pAG416GAL-SrHsp104-FLAG | This paper | N/A | Full description can be found in Materials and methods: Plasmids |
| Recombinant DNA reagent | pRS313HSE-SrHsp104-FLAG | This paper | N/A | Full description can be found in Materials and methods: Plasmids |
| Recombinant DNA reagent | pInducer20-SrHsp104-FLAG | This paper | N/A | Full description can be found in Materials and methods: Plasmids |
| Recombinant DNA reagent | pAG416GAL-SrHsp104$^{DN}$-FLAG | This paper | N/A | Encodes SrHsp104$^{160-892}$; full description can be found in Materials and methods: Plasmids |
| Recombinant DNA reagent | pRS313HSE-SrHsp104$^{DN}$-FLAG | This paper | N/A | Encodes SrHsp104$^{160-892}$; full description can be found in Materials and methods: Plasmids |
| Recombinant DNA reagent | pAG416GAL-SrHsp104$^{DWA(KA)}$-FLAG | This paper | N/A | Encodes SrHsp104$^{K213A:K624A}$; full description can be found in Materials and methods: Plasmids |
| Recombinant DNA reagent | pRS313HSE-SrHsp104$^{DWA(KA)}$ FLAG | This paper | N/A | Encodes SrHsp104$^{K213A:K624A}$; full description can be found in Materials and methods: Plasmids |
| Recombinant DNA reagent | pAG416GAL-SrHsp104$^{DWA(KT)}$-FLAG | This paper | N/A | Encodes SrHsp104$^{K213T:K624T}$; full description can be found in Materials and methods: Plasmids |
| Recombinant DNA reagent | pRS313HSE-SrHsp104$^{DWA(KT)}$ FLAG | This paper | N/A | Encodes SrHsp104$^{K213T:K624T}$; full description can be found in Materials and methods: Plasmids |
| Recombinant DNA reagent | pAG416GAL-SrHsp104$^{DPLA}$-FLAG | This paper | N/A | Encodes SrHsp104$^{Y251A:Y666A}$; full description can be found in Materials and methods: Plasmids |
| Recombinant DNA reagent | pRS313HSE-SrHsp104$^{DPLA}$-FLAG | This paper | N/A | Encodes SrHsp104$^{Y251A:Y666A}$; full description can be found in Materials and methods: Plasmids |
| Recombinant DNA reagent | pAG416GAL-SrHsp104$^{DWB(EA)}$-FLAG | This paper | N/A | Encodes SrHsp104$^{E279A:E691A}$; full description can be found in Materials and methods: Plasmids |
| Recombinant DNA reagent | pRS313HSE-SrHsp104$^{DWB(EA)}$ FLAG | This paper | N/A | Encodes SrHsp104$^{E279A:E691A}$; full description can be found in Materials and methods: Plasmids |
| Recombinant DNA reagent | pAG416GAL-SrHsp104$^{DWB(EQ)}$-FLAG | This paper | N/A | Encodes SrHsp104$^{E279Q:E691Q}$; full description can be found in Materials and methods: Plasmids |
| Recombinant DNA reagent | pRS313HSE-SrHsp104$^{DWB(EQ)}$ FLAG | This paper | N/A | Encodes SrHsp104$^{E279Q:E691Q}$; full description can be found in Materials and methods: Plasmids |

*Continued on next page*

*Continued*

| Reagent type (species) or resource | Designation | Source or reference | Identifiers | Additional information |
|---|---|---|---|---|
| Recombinant DNA reagent | pAG416GAL-RSSS-FLAG | This paper | N/A | Chimera sequence available in *Supplementary file 2*; full description can be found in Materials and methods: Plasmids |
| Recombinant DNA reagent | pRS313HSE-RSSS-FLAG | This paper | N/A | Chimera sequence available in *Supplementary file 2*; full description can be found in Materials and methods: Plasmids |
| Recombinant DNA reagent | pAG416GAL-RRSS-FLAG | This paper | N/A | Chimera sequence available in *Supplementary file 2*; full description can be found in Materials and methods: Plasmids |
| Recombinant DNA reagent | pRS313HSE-RRSS-FLAG | This paper | N/A | Chimera sequence available in *Supplementary file 2*; full description can be found in Materials and methods: Plasmids |
| Recombinant DNA reagent | pAG416GAL-RRRS-FLAG | This paper | N/A | Chimera sequence available in *Supplementary file 2*; full description can be found in Materials and methods: Plasmids |
| Recombinant DNA reagent | pRS313HSE-RRRS-FLAG | This paper | N/A | Chimera sequence available in *Supplementary file 2*; full description can be found in Materials and methods: Plasmids |
| Recombinant DNA reagent | pAG416GAL-SSRS-FLAG | This paper | N/A | Chimera sequence available in *Supplementary file 2*; full description can be found in Materials and methods: Plasmids |
| Recombinant DNA reagent | pRS313HSE-SSRS-FLAG | This paper | N/A | Chimera sequence available in *Supplementary file 2*; full description can be found in Materials and methods: Plasmids |
| Recombinant DNA reagent | pAG416GAL-RSRS-FLAG | This paper | N/A | Chimera sequence available in *Supplementary file 2*; full description can be found in Materials and methods: Plasmids |
| Recombinant DNA reagent | pRS313HSE-RSRS-FLAG | This paper | N/A | Chimera sequence available in *Supplementary file 2*; full description can be found in Materials and methods: Plasmids |
| Recombinant DNA reagent | pMCSG-TtHsp104 | This paper | N/A | Full description can be found in Materials and methods: Plasmids |
| Recombinant DNA reagent | pAG416GAL-TtHsp104-FLAG | This paper | N/A | Full description can be found in Materials and methods: Plasmids |
| Recombinant DNA reagent | pDAT-TtHsp104 | This paper | N/A | Full description can be found in Materials and methods: Plasmids |
| Recombinant DNA reagent | pAG416GAL-TtHsp104$^{DN}$-FLAG | This paper | N/A | Encodes TtHsp104$^{173-923}$; full description can be found in Materials and methods: Plasmids |
| Recombinant DNA reagent | pAG416GAL-TtHsp104$^{DWA(KA)}$-FLAG | This paper | N/A | Encodes TtHsp104$^{K226A:K637A}$; full description can be found in Materials and methods: Plasmids |
| Recombinant DNA reagent | pAG416GAL-TtHsp104$^{DWA(KT)}$-FLAG | This paper | N/A | Encodes TtHsp104$^{K226T:K637T}$; full description can be found in Materials and methods: Plasmids |
| Recombinant DNA reagent | pAG416GAL-TtHsp104$^{DPLA}$-FLAG | This paper | N/A | Encodes TtHsp104$^{Y265A:Y679A}$; full description can be found in Materials and methods: Plasmids |

*Continued on next page*

*Continued*

| Reagent type (species) or resource | Designation | Source or reference | Identifiers | Additional information |
|---|---|---|---|---|
| Recombinant DNA reagent | pAG416GAL-TtHsp104$^{DWB(EA)}$-FLAG | This paper | N/A | Encodes TtHsp104$^{E293A:E704A}$; full description can be found in Materials and methods: Plasmids |
| Recombinant DNA reagent | pAG416GAL-TtHsp104$^{DWB(EQ)}$ FLAG | This paper | N/A | Encodes TtHsp104$^{E293Q:E704Q}$; full description can be found in Materials and methods: Plasmids |
| Recombinant DNA reagent | pAG416GAL-TtSSS-FLAG | This paper | N/A | Chimera sequence available in *Supplementary file 2*; full description can be found in Materials and methods: Plasmids |
| Recombinant DNA reagent | pAG416GAL-TtTtSS-FLAG | This paper | N/A | Chimera sequence available in *Supplementary file 2*; full description can be found in Materials and methods: Plasmids |
| Recombinant DNA reagent | pAG416GAL-TtTtTtS-FLAG | This paper | N/A | Chimera sequence available in *Supplementary file 2*; full description can be found in Materials and methods: Plasmids |
| Recombinant DNA reagent | pAG416GAL-SSTtS-FLAG | This paper | N/A | Chimera sequence available in *Supplementary file 2*; full description can be found in Materials and methods: Plasmids |
| Recombinant DNA reagent | pAG416GAL-TtSTtS-FLAG | This paper | N/A | Chimera sequence available in *Supplementary file 2*; full description can be found in Materials and methods: Plasmids |
| Recombinant DNA reagent | pMCSG-TlHsp104 | This paper | N/A | Full description can be found in Materials and methods: Plasmids |
| Recombinant DNA reagent | pAG416GAL-TlHsp104-FLAG | This paper | N/A | Full description can be found in Materials and methods: Plasmids |
| Recombinant DNA reagent | pRS313HSE-TlHsp104-FLAG | This paper | N/A | Full description can be found in Materials and methods: Plasmids |
| Recombinant DNA reagent | pDAT-TlHsp104 | This paper | N/A | Full description can be found in Materials and methods: Plasmids |
| Recombinant DNA reagent | pAG416GAL-TlHsp104$^{DN}$-FLAG | This paper | N/A | Encodes TlHsp104$^{173-922}$; full description can be found in Materials and methods: Plasmids |
| Recombinant DNA reagent | pRS313HSE-TlHsp104$^{DN}$-FLAG | This paper | N/A | Encodes TlHsp104$^{173-922}$; full description can be found in Materials and methods: Plasmids |
| Recombinant DNA reagent | pAG416GAL-TlHsp104$^{DWA(KA)}$-FLAG | This paper | N/A | Encodes TlHsp104$^{K226A:K638A}$; full description can be found in Materials and methods: Plasmids |
| Recombinant DNA reagent | pRS313HSE-TlHsp104$^{DWA(KA)}$ FLAG | This paper | N/A | Encodes TlHsp104$^{K226A:K638A}$; full description can be found in Materials and methods: Plasmids |
| Recombinant DNA reagent | pAG416GAL-TlHsp104$^{DWA(KT)}$-FLAG | This paper | N/A | Encodes TlHsp104$^{K226T:K638T}$; full description can be found in Materials and methods: Plasmids |
| Recombinant DNA reagent | pRS313HSE-TlHsp104$^{DWA(KT)}$ FLAG | This paper | N/A | Encodes TlHsp104$^{K226T:K638T}$; full description can be found in Materials and methods: Plasmids |
| Recombinant DNA reagent | pAG416GAL-TlHsp104$^{DPLA}$-FLAG | This paper | N/A | Encodes TlHsp104$^{Y265A:Y680A}$; full description can be found in Materials and methods: Plasmids |

*Continued on next page*

*Continued*

| Reagent type (species) or resource | Designation | Source or reference | Identifiers | Additional information |
|---|---|---|---|---|
| Recombinant DNA reagent | pRS313HSE-TlHsp104$^{DPLA}$-FLAG | This paper | N/A | Encodes TlHsp104$^{Y265A:Y680A}$; full description can be found in Materials and methods: Plasmids |
| Recombinant DNA reagent | pAG416GAL-TlHsp104$^{DWB(EA)}$-FLAG | This paper | N/A | Encodes TlHsp104$^{E293A:E705A}$; full description can be found in Materials and methods: Plasmids |
| Recombinant DNA reagent | pRS313HSE-TlHsp104$^{DWB(EA)}$-FLAG | This paper | N/A | Encodes TlHsp104$^{E293A:E705A}$; full description can be found in Materials and methods: Plasmids |
| Recombinant DNA reagent | pAG416GAL-TlHsp104$^{DWB(EQ)}$-FLAG | This paper | N/A | Encodes TlHsp104$^{E293Q:E705Q}$; full description can be found in Materials and methods: Plasmids |
| Recombinant DNA reagent | pRS313HSE-TlHsp104$^{DWB(EQ)}$-FLAG | This paper | N/A | Encodes TlHsp104$^{E293Q:E705Q}$; full description can be found in Materials and methods: Plasmids |
| Recombinant DNA reagent | pAG416GAL-TlSSS-FLAG | This paper | N/A | Chimera sequence available in *Supplementary file 2*; full description can be found in Materials and methods: Plasmids |
| Recombinant DNA reagent | pRS313HSE-TlSSS-FLAG | This paper | N/A | Chimera sequence available in *Supplementary file 2*; full description can be found in Materials and methods: Plasmids |
| Recombinant DNA reagent | pAG416GAL-TlTlSS-FLAG | This paper | N/A | Chimera sequence available in *Supplementary file 2*; full description can be found in Materials and methods: Plasmids |
| Recombinant DNA reagent | pRS313HSE-TlTlSS-FLAG | This paper | N/A | Chimera sequence available in *Supplementary file 2*; full description can be found in Materials and methods: Plasmids |
| Recombinant DNA reagent | pAG416GAL-TlTlTlS-FLAG | This paper | N/A | Chimera sequence available in *Supplementary file 2*; full description can be found in Materials and methods: Plasmids |
| Recombinant DNA reagent | pRS313HSE-TlTlTlS-FLAG | This paper | N/A | Chimera sequence available in *Supplementary file 2*; full description can be found in Materials and methods: Plasmids |
| Recombinant DNA reagent | pAG416GAL-SSTlS-FLAG | This paper | N/A | Chimera sequence available in *Supplementary file 2*; full description can be found in Materials and methods: Plasmids |
| Recombinant DNA reagent | pRS313HSE-SSTlS-FLAG | This paper | N/A | Chimera sequence available in *Supplementary file 2*; full description can be found in Materials and methods: Plasmids |
| Recombinant DNA reagent | pAG416GAL-TlSTlS-FLAG | This paper | N/A | Chimera sequence available in *Supplementary file 2*; full description can be found in Materials and methods: Plasmids |
| Recombinant DNA reagent | pRS313HSE-TlSTlS-FLAG | This paper | N/A | Chimera sequence available in *Supplementary file 2*; full description can be found in Materials and methods: Plasmids |
| Recombinant DNA reagent | pAG416GAL-ClpB-FLAG | This paper | N/A | Full description can be found in Materials and methods: Plasmids |
| Recombinant DNA reagent | pAG416GAL-ClpB$^{K476C}$-FLAG | This paper | N/A | Full description can be found in Materials and methods: Plasmids |

*Continued on next page*

*Continued*

| Reagent type (species) or resource | Designation | Source or reference | Identifiers | Additional information |
|---|---|---|---|---|
| Recombinant DNA reagent | pAG416GAL-ClpB$^{Y503D}$-FLAG | This paper | N/A | Full description can be found in Materials and methods: Plasmids |
| Recombinant DNA reagent | pRS313HSE-ClpB-FLAG | This paper | N/A | Full description can be found in Materials and methods: Plasmids |
| Recombinant DNA reagent | pAG416GAL-ClpG$_{GI}$-FLAG | This paper | N/A | Full description can be found in Materials and methods: Plasmids |
| Recombinant DNA reagent | pAG416GAL-DdHsp104-FLAG | This paper | N/A | Full description can be found in Materials and methods: Plasmids |
| Recombinant DNA reagent | pRS313HSE-DdHsp104-FLAG | This paper | N/A | Full description can be found in Materials and methods: Plasmids |
| Recombinant DNA reagent | pAG416GAL-AtHsp104-FLAG | This paper | N/A | Full description can be found in Materials and methods: Plasmids |
| Recombinant DNA reagent | pRS313HSE-AtHsp104-FLAG | This paper | N/A | Full description can be found in Materials and methods: Plasmids |
| Recombinant DNA reagent | pAG416GAL-ChtHsp104-FLAG | This paper | N/A | Full description can be found in Materials and methods: Plasmids |
| Recombinant DNA reagent | pRS313HSE-ChtHsp104-FLAG | This paper | N/A | Full description can be found in Materials and methods: Plasmids |
| Recombinant DNA reagent | pAG416GAL-LtHsp104-FLAG | This paper | N/A | Full description can be found in Materials and methods: Plasmids |
| Recombinant DNA reagent | pRS313HSE-LtHsp104-FLAG | This paper | N/A | Full description can be found in Materials and methods: Plasmids |
| Recombinant DNA reagent | pAG416GAL-MtHsp104-FLAG | This paper | N/A | Full description can be found in Materials and methods: Plasmids |
| Recombinant DNA reagent | pRS313HSE-MtHsp104-FLAG | This paper | N/A | Full description can be found in Materials and methods: Plasmids |
| Recombinant DNA reagent | pAG416GAL-StHsp104-FLAG | This paper | N/A | Full description can be found in Materials and methods: Plasmids |
| Recombinant DNA reagent | pRS313HSE-StHsp104-FLAG | This paper | N/A | Full description can be found in Materials and methods: Plasmids |
| Recombinant DNA reagent | pAG416GAL-TaHsp104-FLAG | This paper | N/A | Full description can be found in Materials and methods: Plasmids |
| Recombinant DNA reagent | pRS313HSE-TaHsp104-FLAG | This paper | N/A | Full description can be found in Materials and methods: Plasmids |
| Recombinant DNA reagent | pAG416GAL-PfHsp104-FLAG | This paper | N/A | Full description can be found in Materials and methods: Plasmids |
| Recombinant DNA reagent | pAG303GAL-TDP43 | *Johnson et al., 2009* | N/A | |
| Recombinant DNA reagent | pAG303GAL-TDP43-GFPS11 | *Jackrel et al., 2014* | N/A | |
| Recombinant DNA reagent | pAG305GAL-GFPS1-10 | *Jackrel et al., 2014* | N/A | |
| Recombinant DNA reagent | pAG303GAL-FUS | *Sun et al., 2011* | N/A | |
| Recombinant DNA reagent | pAG303GAL-aSyn-YFP | *Gitler et al., 2008* | N/A | |
| Recombinant DNA reagent | pAG304GAL-aSyn-YFP | *Gitler et al., 2008* | N/A | |
| Recombinant DNA reagent | pE-SUMO-Ssa1 | *Michalska et al., 2019* | N/A | |
| Recombinant DNA reagent | pE-SUMO-Hsc70 | *Michalska et al., 2019* | N/A | |
| Recombinant DNA reagent | pE-SUMO-Sis1 | *Michalska et al., 2019* | N/A | |
| Recombinant DNA reagent | pE-SUMO-Ydj1 | *Michalska et al., 2019* | N/A | |
| Recombinant DNA reagent | pE-SUMO-Hdj1 | *Michalska et al., 2019* | N/A | |
| Recombinant DNA reagent | pE-SUMO-Hdj2 | *Michalska et al., 2019* | N/A | |
| Sequence-based reagent | CtHsp104 forward | This paper | qPCR primer | GACGAAGCGTGTGCCAATAC |
| Sequence-based reagent | CtHsp104 reverse | This paper | qPCR primer | CACTTCCTGGAGCCGCTG |

*Continued on next page*

*Continued*

| Reagent type (species) or resource | Designation | Source or reference | Identifiers | Additional information |
|---|---|---|---|---|
| Sequence-based reagent | TtHsp104 forward | This paper | qPCR primer | CAACTACTTCCTGCCCGAG |
| Sequence-based reagent | TtHsp104 reverse | This paper | qPCR primer | ATCTGGACGTTGCGGTCGT |
| Sequence-based reagent | TlHsp104 forward | This paper | qPCR primer | AACCGTCTCACCAAGCGTG |
| Sequence-based reagent | TlHsp104 reverse | This paper | qPCR primer | GCCTCTCCGAGATAGTCCT |
| Sequence-based reagent | aSyn forward | This paper | qPCR primer | ATGTAGGCTCCAAAACCAAGG |
| Sequence-based reagent | aSyn reverse | This paper | qPCR primer | ACTGCTCCTCCAACATTTGTC |
| Sequence-based reagent | snb-1 forward | This paper | qPCR primer | CCGGATAAGACCATCTTGACG |
| Sequence-based reagent | snb-1 reverse | This paper | qPCR primer | GACGACTTCATCAACCTGAGC |
| Sequence-based reagent | cdc-42 forward | This paper | qPCR primer | CCGAGAAAAATGGGTGCCTG |
| Sequence-based reagent | cdc-42 reverse | This paper | qPCR primer | TTCTCGAGCATTCCTGGATCAT |
| Sequence-based reagent | tba-1 forward | This paper | qPCR primer | ATCTCTGCTGACAAGGCTTAC |
| Sequence-based reagent | tba-1 reverse | This paper | qPCR primer | GTACAAGAGGCAAACAGCCAT |
| Peptide, recombinant protein | ScHsp104 | *Jackrel et al., 2014* | N/A | |
| Peptide, recombinant protein | ScHsp104$^{A503S}$ | *Jackrel et al., 2014* | N/A | |
| Peptide, recombinant protein | MbHsp104 | This paper | N/A | Full description can be found in Materials and methods: Protein expression and purification |
| Peptide, recombinant protein | CrHsp104 | This paper | N/A | Full description can be found in Materials and methods: Protein expression and purification |
| Peptide, recombinant protein | His$_6$-(TevC)-CtHsp104 | *Michalska et al., 2019* | N/A | |
| Peptide, recombinant protein | His$_6$-(TevC)-TtHsp104 | This paper | N/A | Full description can be found in Materials and methods: Protein expression and purification |
| Peptide, recombinant protein | His$_6$-(TevC)-TlHsp104 | This paper | N/A | Full description can be found in Materials and methods: Protein expression and purification |
| Peptide, recombinant protein | His$_6$-SUMO-Ssa1 | *Michalska et al., 2019* | N/A | |
| Peptide, recombinant protein | His$_6$-SUMO-Hsc70 | *Michalska et al., 2019* | N/A | |
| Peptide, recombinant protein | His$_6$-SUMO-Sis1 | *Michalska et al., 2019* | N/A | |
| Peptide, recombinant protein | His$_6$-SUMO-Ydj1 | *Michalska et al., 2019* | N/A | |
| Peptide, recombinant protein | His$_6$-SUMO-Hdj1 | *Michalska et al., 2019* | N/A | |
| Peptide, recombinant protein | His$_6$-SUMO-Hdj2 | *Michalska et al., 2019* | N/A | |
| Peptide, recombinant protein | Firefly luciferase | Sigma-Aldrich | L9506 | |
| Peptide, recombinant protein | MBP-(TevC)-TDP43 | This paper | N/A | Full description can be found in Materials and methods: Protein expression and purification |
| Commercial assay or kit | PiColorLock Phosphate Detection | Innova | Cat# 601–0120 | |
| Commercial assay or kit | Luciferase assay reagent | Promega | E1483 | |
| Chemical compound, drug | Creatine phosphate | Roche | 10621722001 | |
| Chemical compound, drug | ATP | Sigma-Aldrich | A3377 | |

## Bioinformatic analyses

Multiple sequence alignments of Hsp104 homologs were generated with Clustal Omega (*Sievers et al., 2011*). Multiple sequence alignments were visualized in JalView (*Waterhouse et al., 2009*). The phylogenetic tree in *Figure 1* was generated using EMBL-EBI Simple Phylogeny tool (*Madeira et al., 2019*). Pairwise sequence identities in *Supplementary file 2* and *Figure 2—figure*

*supplement 2B* were calculated using UniProt Align tool. Divergence times shown in *Figure 5* were from TimeTree (*Kumar et al., 2017*).

## Yeast strains and media

All yeast were WT W303a (*MATa, can1-100, his3-11,15, leu2-3,112, trp1-1, ura3-1, ade2-1*) or the isogenic strain W303aΔ*hsp104* (*Schirmer et al., 2004*). Yeast were grown in rich medium (YPD) or in the appropriate synthetic selection media. Media was supplemented with 2% glucose, raffinose, or galactose.

## Plasmids

Yeast expression vectors encoding TDP-43 (pAG303GAL-TDP-43), FUS (pAG303GAL-FUS), and αSyn-GFP (pAG303GAL-αSyn-GFP and pAG304GAL-αSyn-GFP) were from Aaron Gitler (*Gitler et al., 2008*; *Johnson et al., 2008*; *Sun et al., 2011*). pRS313HSE-ScHsp104^WT-FLAG, pAG416GAL-ScHsp104^WT-FLAG, pAG416GAL-ScHsp104^A503S-FLAG, and pAG416GAL-CtHsp104-FLAG have been described previously (*Michalska et al., 2019*). cDNAs encoding some Hsp104 homologs were kind gifts from Adrian Tsang (Concordia University; TtHsp104, TlHsp104, ChtHsp104, LtHsp104, MtHsp104, StHsp104, and TaHsp104), Susan Lindquist (Whitehead Institute; AtHsp104), and Simon Alberti (Technische Universität Dresden; DdHsp104), while yeast codon-optimized cDNAs encoding other Hsp104s were synthesized by Invitrogen (MbHsp104 and PfHsp104) or Genscript (GsHsp104, CrHsp104, SrHsp104, and PeHsp104). Hsp104 cDNAs codon-optimized for expression in mammalian cells were used for experiments shown in *Figure 3*. Amino acid sequences of all Hsp104 homologs are included in *Supplementary file 1*. Gateway BP reactions were used to shuttle Hsp104 genes into a Gateway entry vector, pDONR221-ccdB. The entry clones were then used to shuttle Hsp104 ORFs into suitable yeast (pAG416GAL-ccdB or pAG413GAL-ccdB or pRS313HSE-ccdB), *C. elegans* (pDAT-ccdB), or mammalian (pInducer20-ccdB) expression vectors via LR reactions. MbHsp104 and CrHsp104 were cloned into pNOTAG for bacterial expression through NdeI and SacI sites. Hsp104 mutants (e.g. ΔN, DWA, DPLA, and DWB) were obtained by Quikchange mutagenesis. The specific mutants are listed in Key Resources Table. Hsp104 chimeras were cloned by overlap-extension PCR. Sequences of all Hsp104 chimeras are included in *Supplementary file 3*.

Doxycycline-inducible TDP-43ΔNLS mammalian expression vector (*Winton et al., 2008*) was modified as follows. Human wild-type TDP-43 was amplified in two separate PCR reactions excluding the NLS and reassembled using Gibson cloning (NEB) into a Doxycycline-inducible expression vector containing an N-terminal mClover3 tag. PCR primers used to generate TRE-mClover3-linker-TDP-43deltaNLS: 5′-GGATCCGGAAGTGGCTCAAGCGGAATGTCTGAATATATTCGGGT-3′; 5′-TGCTGCTGCCACTGCCACTGCTGATGAAGCATCTGTCTCATCCATTGCTGCTGCGTTAT CTTTTGGATAGTTGACA-3′; 5′- ATCAGCAGTGGCAGTGGCAGCAGCAGCAGTCCAGAAAACATCCGA-3′; 5′- AAGTTTGTTGCGCCGGATCC CATTCCCCAGCCAGAAGACT-3′.

## Yeast transformation and spotting assays

Yeast were transformed according to standard protocols using polyethylene glycol and lithium acetate (*Gietz and Schiestl, 2007*). For spotting assays, yeast were grown to saturation overnight in synthetic raffinose dropout media at 30℃. Cultures were normalized and serially diluted 5-fold and spotted in duplicate onto synthetic dropout media containing glucose or galactose. Plates were analyzed after growth for 2–3 days at 30℃.

## Assessing toxicity of Hsp104 homologs and chimeras

W303aΔ*hsp104* yeast were transformed with the indicated Hsp104 plasmid. Transformants were grown to saturation overnight in synthetic raffinose media. Cultures were normalized, spotted onto synthetic dropout media containing glucose or galactose, and incubated at 30℃ or 37℃ for 2–3 d before growth was documented.

## Western blots

Yeast were grown in galactose-containing media to induce protein expression for 5 hr (for strains expressing Hsp104s alone or with TDP-43 or FUS) or 8 hr (for strains expressing αSyn). Cultures

were normalized to $OD_{600}$ = 0.6, and 6 mL of cells were harvested. For heat-shock controls, samples were incubated at 37°C for 30 min prior to processing. Yeast lysates were extracted by incubation with 0.1M NaOH at room temperature for 5 min. Lysates were mixed with SDS sample buffer, boiled for 5 min, and subjected to Tris-HCl SDS-PAGE (4–20% gradient, Bio-Rad) followed by transfer to a PVDF membrane (Millipore). Membranes were blocked in Odyssey blocking buffer (LI-COR) for 1 hr at room temperature or overnight at 4°C. Primary antibody incubations were performed at 4°C overnight or at room temperature for 2 hr. After washing with PBST, membranes were incubated with fluorescently labeled secondary antibodies at room temperature for 1 hr, followed by washing with PBST. Proteins were detected using an Odyssey Fc Dual-Mode Imaging system. Primary antibodies used: mouse monoclonal anti-FLAG M2 (Sigma-Aldrich), rabbit polyclonal anti-TDP-43 (Proteintech), rabbit polyclonal anti-FUS (Bethyl Labs, Cat #A300), rabbit polyclonal anti-GFP (Sigma-Aldrich), mouse monoclonal 3-phosphoglycerate kinase (Novex), rabbit polyclonal anti-Hsp26 (Johannes Buchner, TU-Munich), fluorescently labeled anti-mouse and anti-rabbit secondary antibodies (Li-Cor).

## Fluorescence microscopy

To visualize changes in TDP-43 and αSyn localization in response to coexpression of various Hsp104s, yeast strains expressing fluorescently-tagged TDP-43 and αSyn were previously generated and described (*Jackrel et al., 2014*). For microscopy, these strains were grown for 5 hr (TDP-43) or 8 hr (αSyn) in galactose-containing media at 30°C. For TDP-43 microscopy, cells were harvested and fixed with ice-cold 70% ethanol, washed three times with ice-cold PBS, and stained with 4',6-diamidino-2-phenylindole (DAPI) in Vectashield mounting medium (Vector Laboratories) to visualize nuclei. For αSyn, imaging was performed with live cells. Images were collected at 100x magnification using a Leica-DMIRBE microscope and processed using ImageJ software (NIH).

## HEK-293T cell culture and transfections

HEK293T cells were from ATCC (Cat# CRL-3216 RRID:CVCL_0063) and have been authenticated by the vendor and were not contaminated by mycoplasma. HEK-293T cells were maintained in Dulbecco's modified Eagle's medium, high glucose (Gibco) containing 10% fetal bovine serum (Life Sciences), 1% non-essential amino acids (Gibco), and 1% antibiotic-antimyotic (Gibco). Cells were plated in gelatin-coated 6-well plates at a density of $3 \times 10^6$ cells/plate 24 hr before transfection. Cells were transfected with 2 µg total DNA and 7.35 µl polyethylenimine HCl MAX transfection reagent (Polysciences, Inc). Wells co-transfected with mClover3-TDPΔNLS and HSP104 variants received 1 µg of each plasmid. Media was changed 6 hr post-transfection to media containing 1 µg/ml of doxycycline hyclate (Sigma-Aldrich) to induce transgene expression. Transfected cells were lifted every 24 hr over 2 days, at which point cells were analyzed by FACS (FACSAria Fusion BD). Cells were gated to have a narrow range of FCS and SSC values and to be fluorescence positive. TDP-43 aggregation was quantified by comparing the height (FITC-H) to the width (FITC-W) of the fluorescence channel using 488 nm laser and FITC filters.

## Generation of transgenic *C. elegans* and neurodegeneration analysis

Nematodes were maintained through well-established methods (*Brenner, 1974*). Constructs were injected into animals to create transgenic lines using previously-described methods (*Berkowitz et al., 2008*). Strains UA381 (*baIn11* [$P_{dat-1}$: :α-syn, $P_{dat-1}$: :GFP]; *baEx210* [$P_{dat-1}$:: *CtHsp104*, *rol-6*]), UA382 (*baIn11* [$P_{dat-1}$: :α-syn, $P_{dat-1}$: :GFP]; *baEx211* [$P_{dat-1}$::*TtHsp104*, *rol-6*]), UA383 (*baIn11* [$P_{dat-1}$: :α-syn, $P_{dat-1}$: :GFP]; *baEx212* [$P_{dat-1}$::*TlHsp104*, *rol-6*]) were generated by injecting 50 ng/µl of corresponding plasmid construct into UA44 (*baIn11* [$P_{dat-1}$: :α-syn, $P_{dat-1}$: :GFP]) with phenotypic marker (*rol-6*, 50 ng/µl, for roller expression). Strains UA403 (*vtIs7* [$P_{dat-1}$::GFP]; *baEx223* [$P_{dat-1}$::*CtHSP104*, *rol-6*]), UA404 (*vtIs7* [$P_{dat-1}$::GFP]; *baEx224* [$P_{dat-1}$::*TtHSP104*, *rol-6*]), UA405 (*vtIs7* [$P_{dat-1}$::GFP]; *baEx225* [$P_{dat-1}$::*TiHSP104*, *rol-6*]) were generated by injecting 50 ng/µl of corresponding plasmid construct into BY250 (*vtIs7* [$P_{dat-1}$::GFP]) with phenotypic marker (*rol-6*, 50 ng/µl, for roller expression). Three independent stable lines were created for each group. For dopaminergic neurodegeneration analyses, the transgenic animals were scored as described previously (*Hamamichi et al., 2008*). Briefly, on the day of analysis, the six anterior dopaminergic neurons [four CEP (cephalic) and two ADE (anterior deirid)] were examined in 30 animals randomly selected for each trial worms that express the roller marker in the body wall muscle cells. Neurons were analyzed

for any degenerative phenotypes, such as a missing dendritic process, cell body loss, or a blebbing neuronal process. Each animal was scored as having normal or wild-type neurons when none of the degenerative phenotypes were present in any anterior dopaminergic neurons. Three independent transgenic worm lines were analyzed per genetic background and an average of total percentage of worms with normal neurons was reported in the study. One-way ANOVA, followed by a Dunnett's multiple comparisons *post hoc* test, was performed for statistical analysis using GraphPad Prism Software.

## Quantitative real-time PCR

RNA isolation and RT-qPCR was performed on worms using previously published methods (*Knight et al., 2014*). Briefly, total RNA was isolated from 100 young adult (day 4 post-hatching) nematodes from corresponding transgenic lines using TRI reagent (Molecular Research Center). The genomic DNA contamination was removed with 1 μl of DNaseI (Promega) treatment for 60 min at 37°C, then with DNase Stop solution for 10 min at 65°C. 1 μg of RNA was used for cDNA synthesis using the iScript Reverse Transcription Supermix for RT-qPCR (Bio-Rad) following the manufacturer's protocol. RT-qPCR was performed using IQ-SYBR Green Supermix (Bio-Rad) with the Bio-Rad CFX96 Real-Time System. Each reaction contained 7.5 μl of the IQ-SYBR Green Supermix, 200 nM of forward and reverse primers and 5 ng of cDNA, to a final volume of 15 μl. The cycling conditions were as follows: polymerase activation and DNA denaturation at 95°C for 3 min, followed by 35 cycles of 10 s at 95°C, 30 s at 60°C. After the final cycle, a melting curve analysis was performed using the default setting of CFX96 Real-Time System. A single melt peak for each targeted gene was observed and no non-specific amplification was detected in each reaction mixture by agarose gel electrophoresis. PCR efficiency was calculated from standard curves that were generated using serial dilution ($E_{\alpha\text{-syn}}$ = 98.8%, $E_{TtHsp104}$ = 101.0%, $E_{TlHsp104}$ = 97.4%, $E_{tba\text{-}1}$ = 98.8%, $E_{cdc\text{-}42}$ = 98.4%, $E_{snb\text{-}1}$ = 95.3%). The expression levels of α-syn and Hsp104 variants were normalized to three reference genes (*snb-1*, *cdc-42*, and *tba-1*). No amplification was detected in NTC and NRT controls. The reference target stability was analyzed by GeNorm and passed for all reference genes listed above. All target genes were measured in triplicates for three independent transgenic lines for each sample in this study. The data analysis was performed by the Gene Expression Module of CFX Manager software.

The following primers were used for the assays:

> *CtHsp104* Forward: GACGAAGCGTGTGCCAATAC
> *CtHsp104* Reverse: CACTTCCTGGAGCCGCTG
> *TtHsp104* Forward: CAACTACTTCCTGCCCGAG
> *TtHsp104* Reverse: ATCTGGACGTTGCGGTCGT
> *TlHsp104* Forward: AACCGTCTCACCAAGCGTG
> *TlHsp104* Reverse: GCCTCTCCGAGATAGTCCT
> *α-syn* Forward: ATGTAGGCTCCAAAACCAAGG
> *α-syn* Reverse: ACTGCTCCTCCAACATTTGTC
> *snb-1* Forward: CCGGATAAGACCATCTTGACG
> *snb-1* Reverse: GACGACTTCATCAACCTGAGC
> *cdc-42* Forward: CCGAGAAAAATGGGTGCCTG
> *cdc-42* Reverse: TTCTCGAGCATTCCTGGATCAT
> *tba-1* Forward: ATCTCTGCTGACAAGGCTTAC
> *tba-1* Reverse: GTACAAGAGGCAAACAGCCAT

## Thermotolerance

W303aΔ*hsp104* yeast were transformed with plasmids bearing either Hsp104 from *S. cerevisiae,* the indicated Hsp104 homolog, or Hsp104 mutant under the native *HSP104* promoter (except for TtHsp104 and derivative mutants, which were expressed from p*GAL*), or an empty vector control. Transformants were selected, grown to saturation in yeast minimal media (SD-His), and then diluted to $OD_{600}$ = 0.2 in fresh SD-His (except for TtHsp104 and derivative mutants, which were grown to saturation in SRaff-Ura and diluted in SGal-Ura). Yeast were allowed to double at 30°C, after which cultures were normalized and pretreated at 37°C for 30 min to induce expression of heat-shock proteins (samples were taken here and processed for Western blot to assess Hsp104 and Hsp26 expression). Cells were then heat shocked at 50°C for the indicated time and cooled for 2 min on ice.

Cultures were then diluted 1000-fold, plated on SD-His (except for TtHsp104 and derivative mutants, which were plated on SD-Ura), and plates were incubated at 30°C for 2–3 days to observe viable colonies.

## Protein expression and purification

TEV protease and Ulp1 were purified via standard protocols. Untagged ScHsp104 was expressed from the pNOTAG-ScHsp104 vector (*Hattendorf and Lindquist, 2002*) and purified as previously described (*DeSantis et al., 2014*; *Torrente et al., 2016*). Briefly, pNOTAG-ScHsp104 was used to transform BL21(DE3)RIL *E. coli*. Transformed cells were grown in 2xYT broth supplemented with 25 µg/ml chloramphenicol and 100 µg/ml ampicillin at 37°C until an $OD_{600}$ of 0.4–0.6 was reached, at which point cells were cooled to 15°C. Expression was induced by addition of 1 mM isopropyl 1-thio-β-D-galactopyranoside for 15–18 hr. Cells were harvested by centrifugation (4000 × g, 4°C, 25 min), resuspended in lysis buffer (50 mM Tris-HCl, pH 8.0, 10 mM $MgCl_2$, 2.5% glycerol, 2 mM β-mercaptoethanol, 5 µM pepstatin, c0mplete EDTA-free protease inhibitors (Roche)). Cells were treated on ice with 20 mg lysozyme per 1l culture and lysed by sonication. Cell debris was removed by centrifugation at 16,000 × g at 4°C for 20 min, and the supernatant was applied to Affi-Gel Blue resin (Bio-Rad). Resin was incubated with the lysates for 4 hr at 4°C with slow rotation. Resin was then washed four times with wash buffer (50 mM Tris-HCl, pH 8.0, 10 mM $MgCl_2$, 100 mM KCl, 2.5% glycerol, 2 mM β-mercaptoethanol). ScHsp104 was eluted with wash buffer supplemented with 1 M KCl. The protein was then exchanged into running buffer Q (20 mM Tris-HCl pH 8.0, 0.5 mM EDTA, 5 mM $MgCl_2$, 50 mM NaCl), further purified by ResourceQ anion exchange chromatography, and eluted with a linear salt gradient (50 mM-1 M NaCl). Eluted protein was then exchanged into storage buffer (40 mM HEPES-KOH pH 7.4, 150 mM KCl, 20 mM $MgCl_2$, 10% glycerol, 1 mM DTT), snap-frozen in liquid $N_2$, and stored at −80°C until use.

MbHsp104 and CrHsp104 were expressed as untagged proteins by subcloning the MbHsp104 and CrHsp104 ORFs into pNOTAG through NdeI and SacI sites. Protein expression and lysis were carried out as for ScHsp104. However, MbHsp104 and CrHsp104 bound poorly to Affi-Blue resin, even after extended incubation times (data not shown). Instead, MbHsp104 and CrHsp104 were precipitated from clarified bacterial lysates by addition of solid ammonium sulfate to 40% of saturation. Precipitates were collected by centrifugation (16,000 × g at 4°C, 20 min) and resuspended in buffer containing 50 mM Tris-HCl, pH 8.0, 10 mM $MgCl_2$, 100 mM KCl, 2.5% glycerol, 2 mM β-mercaptoethanol. To resolubilize precipitates, ammonium sulfate was removed by dialysis against this buffer, with three buffer changes. The dialyzate was filtered and applied to a 5 ml HiTrapQ column and purified with running buffer Q (20 mM Tris pH 8.0, 0.5 mM EDTA, 5 mM $MgCl_2$, 50 mM NaCl) and eluted with a linear gradient of buffer Q+ (20 mM Tris pH 8.0, 0.5 mM EDTA, 5 mM $MgCl_2$, 1 M NaCl) over 40 column volumes. Peak fractions were collected, pooled, and exchanged into size-exclusion buffer (40 mM HEPES-KOH pH 7.4, 150 mM KCl, 10 mM $MgCl_2$ 1 mM DTT) and further purified by size-exclusion chromatography on a pre-equilibrated Superdex 200 column (GE healthcare). Protein in size-exclusion buffer was concentrated to ~10 mg/ml, supplemented with 10% glycerol, snap-frozen in liquid $N_2$ and stored at −80°C until use.

CtHsp104, TtHsp104, and TlHsp104 were expressed from pMCSG68 vector as TEV protease-cleavable $His_6$-tagged fusion proteins. Protein expression and lysis were carried out as for other Hsp104s, except in this case lysis buffer consisted of 40 mM HEPES-KOH pH 7.4, 500 mM KCl, 20 mM $MgCl_2$, 2.5% glycerol, 20 mM imidazole, 2 mM β-mercaptoethanol supplemented with 5 µM pepstatin A and complete protease inhibitor tablets. Clarified lysate was loaded onto Ni-NTA resin. The resin was washed with 10 volumes of wash buffer (same formulation as lysis buffer except without protease inhibitors) and eluted in wash buffer supplemented with 350 mM imidazole. TEV protease was added to the eluted protein, and the sample was dialyzed against wash buffer containing no imidazole for 4 hr at room temperature followed by ~16 hr at 4°C. After dialysis and cleavage, the protein was loaded onto a second Ni-NTA column to remove the $His_6$ tag and uncleaved protein. Eluted protein was pooled, concentrated, and exchanged into high salt storage buffer (40 mM HEPES-KOH pH 7.4, 500 mM KCl, 20 mM $MgCl_2$, 10% glycerol, and 1 mM DTT), snap-frozen in liquid $N_2$, and stored at −80°C until further use.

Ssa1, Hsc70, Sis1, Ydj1, Hdj1, and Hdj2 (in pESUMO (Life Sensors)) were expressed as N-terminally $His_6$-SUMO-tagged proteins in BL21(DE3)RIL cells. Transformed cells were grown at 37°C in Luria broth supplemented with 25 µg/ml chloramphenicol and 100 µg/ml ampicillin to an

OD$_{600}$ ~0.5. Cultures were cooled to 15°C, and expression was induced with 1 mM IPTG for 16 hr. Cells were harvested, resuspended in lysis buffer (50 mM HEPES pH 7.5, 750 mM KCl, 5 mM MgCl$_2$, 10% glycerol, 20 mM imidazole, 2 mM β-mercaptoethanol, 5 µM pepstatin A, and c0mplete protease inhibitor (Roche)), and lysed by treatment with lysozyme and sonication. Lysates were clarified by centrifugation (16,000 × g, 20 min, 4°C), and incubated with Ni-NTA resin for 90 min at 4°C. Resin was washed with 10 column volumes of wash buffer (50 mM HEPES pH 7.5, 750 mM KCl, 10 mM MgCl$_2$, 10% glycerol, 20 mM imidazole, 1 mM ATP, 2 mM β-mercaptoethanol) and eluted with two column volumes of elution buffer (wash buffer+300 mM imidazole). To cleave the His$_6$-SUMO tag, Ulp1 was added at a 1:100 molar ratio, and imidazole was removed by dialysis against wash buffer. After dialysis, protein was loaded onto a 5 ml HisTrap column (GE Healthcare) and eluted with a linear imidazole gradient (20–350 mM) over 40 column volumes. Fractions containing cleaved protein were pooled, concentrated, and purified further by Resource Q (Ssa1, Hsc70, Ydj1, and Hdj2) or Resource S (Sis1 and Hdj1) ion exchange chromatography. Ssa1, Hsc70, Ydj1, Hdj2, Sis1, and Hdj1 were snap-frozen in liquid N$_2$ and stored at −80°C in elution buffer supplemented with 10% glycerol.

TDP-43 was expressed with an N-terminal, TEV-cleavable maltose binding protein (MBP) tag in BL21(DE3)RIL cells. Transformed cells were grown at 37°C in Luria broth supplemented with 25 µg/ml chloramphenicol and 100 µg/ml ampicillin to an OD$_{600}$ ~0.8. Cultures were cooled to 15°C, and expression was induced with 1 mM IPTG for 16 hr. Cells were harvested, resuspended in lysis buffer (50 mM HEPES-KOH pH 7.5, 500 mM NaCl, 10% glycerol, 2 mM EDTA, 2 mM DTT, and c0mplete protease inhibitor (Roche)), and lysed by treatment with lysozyme and sonication. Lysates were clarified by centrifugation (16,000 × g, 20 min, 4°C). Cleared lysate was added to a 50% slurry of amylose resin (New England Biolabs) in lysis buffer and incubated at 4°C for 30 min to bind MBP-TDP-43. The resin was washed with lysis buffer, and eluted MBP-TEV with elution buffer (50 mM HEPES-KOH pH 7.5, 500 mM NaCl, 10% glycerol, 2 mM EDTA, 2 mM DTT, 10 mM maltose). Purified fractions were pooled, snap-frozen in liquid N$_2$, and stored at −80°C until use.

## Size-exclusion chromatography with multiangle light scattering (SEC-MALS)

Molecular weights of Hsp104 homologs (30 µM monomer) were determined using multi-angle light scattering coupled with refractive interferometric detection (Wyatt Technology Corporation) and a Superdex 200 10/300 size-exclusion column. The column was equilibrated with 40 mM HEPES-KOH, 140 mM KCl, and 20 mM MgCl$_2$ at room temperature and elution of Hsp104 was monitored by both absorbance at 280 nm and refractive index. For TtHsp104, Superdex 200 5/150 column was used, and 1 mM ATP was added to the running buffer to promote oligomerization. For TlHsp104, the presence of higher order species at the leading edge of the elution peak raised the observed molecular weight, although elution times for TlHsp104 are consistent with hexamers observed for other Hsp104 homologs under identical conditions.

## ATPase activity

Hsp104 (0.25 µM monomer) was incubated with ATP (1 mM) for 5 min at the indicated temperatures in buffer (25 mM HEPES-KOH pH 7.4, 150 mM potassium acetate, 10 mM magnesium acetate, 10 mM DTT). ATPase activity was determined by inorganic phosphate release using a malachite green phosphate detection kit (Innova). Background hydrolysis was determined at time zero and subtracted.

## Luciferase disaggregation and reactivation

Luciferase disaggregation and reactivation was performed as described (*DeSantis et al., 2012*). Aggregated firefly luciferase (50 nM) was incubated with Hsp104 (0.167 µM hexamer) with ATP (5 mM) and an ATP regeneration system (ARS; 1 mM creatine phosphate, 0.25 µM creatine kinase) plus or minus 0.167 µM Hsp70 (Ssa1 or Hsc70) and 0.167 µM Hsp40 (Sis1, Ydj1, Hdj1, or Hdj2, as indicated). Luciferase activity was assessed by luminescence on a TECAN Safire II plate reader.

## Semen-derived enhancer of virus infection (SEVI) remodeling

SEVI remodeling was performed as previously described (*Castellano et al., 2015*). SEVI fibrils (20 µM monomer) were incubated with Hsp104 homologs (3 µM hexamer) in LRB buffer in the presence

of ATP (5 mM) and an ATP regeneration system (0.1 mM ATP, 0.02 mg/ml creatine kinase, 10 mM creatine phosphate). Samples were incubated at 37°C for the duration of the experiments. At various time points, aliquots were removed, added to a 96-well plate containing a solution of 25 µM ThT in LRB buffer. ThT fluorescence characteristics were measured on a Tecan Safire[2] microplate reader with excitation and emission filters set to 440 nm and 482 nm, respectively. To assess fibril morphology by negative stain EM, reaction aliquots were spotted on Formvar carbon-coated grids (EM Sciences) and stained with 2% uranyl acetate. Samples were visualized using a JEOL-1010 electron microscope.

### TDP-43 aggregation experiments

TDP-43 aggregation was initiated by incubating 3 µM MBP-TEV-TDP-43 with TEV protease in assembly buffer (40 mM HEPES-KOH pH 7.5, 150 mM NaCl, 10 mM $MgCl_2$, 1 mM DTT) without agitation. Aggregation was monitored by turbidity ($A_{395\ nm}$) in a TECAN Infinite M1000 plate reader. In some reactions, either ScHsp104, MbHsp104, or CrHsp104 were added (6 µM) with or without ATP (5 mM) and regeneration system (1 mM creatine phosphate and 0.25 µM creatine kinase). We verified that neither Hsp104 nor ATP affected cleavage of the MBP tag by Western blot (see *Figure 6—figure supplement 3*).

## Acknowledgements

We thank Ryan Cupo and Charlotte Fare for comments on the manuscript, Kushol Gupta for help with SEC-MALS, and Oliver King for help with the phylogenetic analyses. We also thank Susan Lindquist, Aaron Gitler, Simon Alberti, Adrian Tsang, and Johannes Buchner for generous provision of reagents. This work was supported by a Muscular Dystrophy Association Research Award (JS), an ALS Association Award (JS), the Life Extension Foundation (JS), a Linda Montague Pechenik Research Award (JS), the Packard Center for ALS Research at Johns Hopkins University (JS), Target ALS (MEJ and JS), an NSF Graduate Research Fellowship DGE-0822 (LMC), an Alzheimer's Association Research Fellowship (JL), a Warren Alpert Foundation Distinguished Scholars Fellowship (JL), a Blavatnik Family Fellowship (EC), and NIH grants T32GM071399 (ZMM), F31NS101807 (ZMM), T32GM008076 (EC), and R01GM099836 (JS).

## Additional information

### Competing interests

James Shorter: JS is a consultant for Dewpoint Therapeutics and Maze Therapeutics. The other authors declare that no competing interests exist.

### Funding

| Funder | Grant reference number | Author |
| --- | --- | --- |
| NIH | R01GM099836 | James Shorter |
| Muscular Dystrophy Association | | James Shorter |
| ALS Association | | James Shorter |
| Life Extension Foundation | | James Shorter |
| Linda Montague Pechenik | | James Shorter |
| Johns Hopkins University | | James Shorter |
| Target ALS | | Meredith E Jackrel James Shorter |
| National Science Foundation | DGE-0822 | Laura M Castellano |
| Alzheimer's Association | Research Fellowship | JiaBei Lin |
| Warren Alpert Foundation | Distinguished Scholars Fellowship | JiaBei Lin |

| Blavatnik Family Foundation | | Edward Chuang |
| --- | --- | --- |
| NIH | T32GM071399 | Zachary M March |
| NIH | F31NS101807 | Zachary M March |
| NIH | T32GM008076 | Edward Chuang |

The funders had no role in study design, data collection and interpretation, or the decision to submit the work for publication.

### Author contributions
Zachary M March, Conceptualization, Data curation, Formal analysis, Funding acquisition, Validation, Investigation, Visualization, Methodology, Writing - original draft, Writing - review and editing; Katelyn Sweeney, Validation, Investigation; Hanna Kim, Xiaohui Yan, Validation, Investigation, Writing - review and editing; Laura M Castellano, Funding acquisition, Investigation; Meredith E Jackrel, JiaBei Lin, Funding acquisition, Investigation, Writing - review and editing; Edward Chuang, Resources, Funding acquisition, Methodology, Writing - review and editing; Edward Gomes, Resources, Investigation, Methodology; Corey W Willicott, Investigation; Karolina Michalska, Andrzej Joachimiak, Ophir Shalem, Resources, Supervision, Methodology, Writing - review and editing; Robert P Jedrzejczak, Resources, Methodology; Kim A Caldwell, Guy A Caldwell, Resources, Supervision, Methodology, Writing - original draft, Writing - review and editing; James Shorter, Conceptualization, Resources, Formal analysis, Supervision, Funding acquisition, Writing - original draft, Project administration, Writing - review and editing

### Author ORCIDs
Zachary M March https://orcid.org/0000-0003-2441-899X
Meredith E Jackrel https://orcid.org/0000-0003-4406-9504
Kim A Caldwell http://orcid.org/0000-0003-1580-6122
James Shorter https://orcid.org/0000-0001-5269-8533

### Decision letter and Author response
Decision letter https://doi.org/10.7554/eLife.57457.sa1
Author response https://doi.org/10.7554/eLife.57457.sa2

# Additional files

## Supplementary files
• Supplementary file 1. Amino acid sequences of Hsp104 homologs. The primary sequence of each Hsp104 homolog is provided.

• Supplementary file 2. Pairwise sequence identity between Hsp104 homologs. The pairwise sequence identities between Hsp104 homologs were calculated using UniProt Align tool.

• Supplementary file 3. Amino acid sequences of Hsp104 chimeras. The primary sequence of each Hsp104 chimera is provided.

• Transparent reporting form

## Data availability
All data generated or analysed during this study are included in the manuscript.

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
