## [Decision Letter]

**Acceptance summary:**

Formation of fibrillar (amyloid) aggregates is associated with neurodegenerative diseases such as Parkinson's disease and amyotrophic lateral sclerosis (ALS). Bacteria and fungi, but not mammalian cells, contain specialized AAA+ chaperones of the Hsp104 type that have the ability to dissociate fibrillar aggregates. Here, the authors investigated natural homologues of Hsp104 from species across multiple kingdoms that can suppress cellular toxicity arising from aggregation of disease-associated proteins, such as TDP-43, α-synuclein and FUS. Using natural variants of Hsp104, they identified homologs that prevented aggregation without showing previously observed off-target effects. These findings may help in developing new therapeutic strategies.

**Decision letter after peer review:**

Thank you for submitting your article "Therapeutic genetic variation revealed in diverse Hsp104 homologs" for consideration by *eLife*. Your article has been reviewed by three peer reviewers, one of whom is a member of our Board of Reviewing Editors, and the evaluation has been overseen by Huda Zoghbi as the Senior Editor. The following individual involved in review of your submission has agreed to reveal their identity: David Li-Kroeger (Reviewer #3).

The reviewers have discussed the reviews with one another and the Reviewing Editor has drafted this decision to help you prepare a revised submission.

The reviewers and the reviewing editor agree that this study should be considered for publication in *eLife*, subject to the following revisions:

Essential revisions:

1) The authors need to more carefully interpret protein levels in their experiments. Throughout the text, the authors conclude that the levels of Hsp104, ɑ-synuclein, and TDP-43 do not change enough to affect their preferred interpretations. However, levels of the Hsp104 variants vary over at least an order of magnitude (Figure 1D, Figure 2C, Figure 3B, Figure 5C). The authors need to consider Hsp104 variant levels when arguing that one Hsp104 variant is more effective than another (e.g. subsection “Diverse Hsp104 homologs selectively suppress TDP-43 toxicity and aggregation in yeast”).

Furthermore, some Hsp104 variants reduce expression of toxic proteins (i.e., A503S in Figure 1D, Figure 2D, Figure 3B). The text needs to properly acknowledge that alterations in toxic protein expression could be a mechanism by which some Hsp104 variants suppress toxicity (e.g. subsection “Hsp104 homologs prevent TDP-43 aggregation in human cells”, sentence beginning with, "We also monitored mClover3-TDP-43∆NLS…").

2) Some experiments lack important controls. In Figure 4, the authors should also analyze worms lacking both ɑ-synuclein and Hsp104 as well as worms expressing the Hsp104 variants alone. This control will help the reader to understand: (a) whether the loss of DA neurons is ɑ-synuclein-specific, and (b) whether the Hsp104-dependent rescue of DA neurons is ɑ-synuclein-specific. In Figure 5G,H, the authors should include un-aggregated luciferase controls so the reader can better understand the magnitude of luciferase refolding induced by these Hsp104 variants.

3) The result that these Hsp104 variants could suppress toxicity by a disaggregase-independent mechanism is a compelling finding that requires an additional control. Because many of these Hsp104 variants have not been biochemically characterized before this work, the authors should demonstrate that the canonical walker A, pore loop, and walker B mutations used in Figure 6 disrupt disaggregase activity in the expected manner. The authors could use the same luciferase assay from Figure 5G,5H with wild-type, walker A mutant, pore loop mutant, and walker B mutant variants of *S. cerevisiae* Hsp104 as well as Hsp104 from one of the more exotic species.

4) Subsection "Hsp104 homologs prevent TDP-43 aggregation in human cells":

a) Regarding the observation that ScHsp104WT also reduced the proportion of HEK293T cells with TDP-43 aggregation at 24-hours post transfection while it did not in yeast cells (Figure 3C vs. Figure 1G), the authors did not address if this is just a protein artefact in utilizing mClover3-TDP-43ΔNLS instead of TDP-43-GFP, or if it reflects differences in cellular environments or factors related to Hsp104 activity that are unaccounted for. It should be emphasized that the latter possibility requires attention as it raises the problem of relevance in using the yeast model to approximate the human cell, despite the authors' justification by referencing such practice by other groups. It is thus prudent that the authors repeat some of the TDP-43 experiments for ScHsp104WT in yeast cells using mClover3-TDP-43ΔNLS instead of TDP-43-GFP, for example.

b) In tracking the TDP-43 foci formation at 24-hours versus 48-hours post transfection, the authors did not address as to why the magnitude of increase in % cells with foci from 24-hours to 48-hours displayed by no-Hsp104 (vector-only) control is drastically lower compared to all other cells co-expressing a Hsp104 homologue. One explanation is the unaccounted protein overexpression-induced stress / toxicity experienced by cells co-transfected with an Hsp104 homologue that is absent in the vector-only (no Hsp104) control, which will likely make the results more difficult to interpret.

c) Western blots for samples at 48-hours post transfection is missing.

5) Subsection "αSyn-selective Hsp104 homologs prevent dopaminergic neurodegeneration in *C. elegans*":

a) Regarding the interpretation of lack of neuroprotection by CtHsp104 due to target promiscuity, this argument will be convincing once CtHsp104 is shown to also fail in providing neuroprotection against TDP-43 toxicity compared to Hsp104 homologues that are specific against TDP-43 using the *C. elegans* model (TDP-43 toxicity in *C. elegans* model: https://www.ncbi.nlm.nih.gov/pubmed/20530643). This data is currently missing from the manuscript.

6) Subsection "Differential suppression of proteotoxicity by Hsp104 homologs is not due to changes in disaggregase activity":

a) Regarding the authors' statement on potentiated Hsp104 disaggregases having elevated ATPase and disaggregase activities: Even though this has been shown previously, ScHsp104(A503S) or another potentiated mutant should be included in the biochemistry assays presented as a positive control, which will validate the experimental procedures employed and importantly, to provide conclusive evidence that the natural Hsp104 homologues are indeed "potentiated" differently compared to the artificially potentiated mutants.

b) In Figure 5C, the authors should also include Western blots showing normal induction of Hsp26 or other heat shock-inducible proteins as confirmation that the yeast strains used are not compromised in heat shock response.

c) In Figure 5E, the authors should label the individual rescuer and non-rescuer Hsp104 homologues; the current figure is not very informative.

d) Although the authors presented extensive biochemistry data showing the ATPase and disaggregase activity of Hsp104 homologues, there is no discussion in regard to the oligomerization state, which can be assessed using a variety of experimental techniques, including native PAGE, size exclusion chromatography, analytical ultracentrifugation, negative-stain electron microscopy, etc. In fact, examination of oligomeric state becomes particularly important in the authors' investigation of Hsp104 domains that contribute to suppression of aggregate toxicity, as chimeric proteins or isolated cognate domains may not form the expected hexameric form. Furthermore, data pertaining to the particular oligomeric state of the chimeras / cognate domains are expected to provide valuable information in establishing the molecular / structural basis of any retention of loss of toxicity suppression activity.

7) Subsection "ATPase-independent passive chaperone activity underlies suppression of TDP-43 and αSyn toxicity by Hsp104 homologs":

a) Regarding the authors' claim of Hsp104 homologues in suppressing aggregate toxicity via passive chaperone activity: Although the authors have provided extensive, thorough yeast genetics data showing that neither the ATPase nor substrate translocation activity is required for aggregate toxicity suppression, to claim passive chaperone activity being the molecular basis requires additional experimental proof, such as comparing the refolding rate of thermally denatured reporter proteins (e.g. luciferase) in the presence or absence of Hsp104 homologues, their respective chimeric proteins or isolated cognate domains.

8) In subsection “Diverse Hsp104 homologs selectively suppress TDP-43 toxicity and aggregation in yeast” and subsection “Distinct Hsp104 homologs selectively suppress αSyn toxicity and inclusion formation in yeast”, referring to the data In Figure 1—figure supplement 3: Suppression of TDP-43 or αSyn toxicity by select Hsp104 homologs is a substrate-specific effect (related to Figure 1 and Figure 2). The authors make the claim that "MbHsp104, SrHsp104, CrHsp104, PeHsp104, and GsHsp104, which all suppress TDP-43 toxicity, do not suppress αSyn nor FUS toxicity" (panel A) and that "TtHsp104 and TlHsp104 do not suppress TDP43 nor FUS toxicity" panel B and that "other Hsp104s investigated in this study do not suppress toxicity of TDP-43 (left), αSyn (middle), or FUS (right)" (panel C). While most of the data here support the authors conclusions, the wording "does not suppress" is a very strong statement. The data is not completely convincing. For example, in panel C, for the FUS overexpression (right panel) ChtHsp104 and StHsp140 seem to show some suppression of toxicity. Similarly, MtHsp104, StHsp104 and TaHsp104 for α-Syn maybe show some slight suppression. I don't think this changes the nature of the conclusion; however, the authors should be a little more cautious in their choice of wording and/or better discuss the data if they feel that there is compelling evidence that absolutely no suppression is shown.

9) In subsection “Diverse Hsp104 homologs selectively suppress TDP-43 toxicity and aggregation in yeast”: the authors state "Hsp104 homologs were consistently expressed,.… (Figure 1D)". The data on the Western Blot for Hsp104 (anti-flag in the blot in Figure 1D) is not convincing. The levels of protein look variable. Perhaps a quantification of replicates could be added to show whether Hsp104::flag protein levels are indeed consistent? Similarly, in Figure 2 panel C the Western blot could also be more convincing with replication and quantification. In Figure 5C, the Western Blot and the text that refer to it show variation in the levels of expression as well. For example, ScHsp104, TtHsp104 and GsHsp104 show much higher protein levels than StHsp104, MtHsp104 and ChtHsp104 yet the text claims: "These differences are not due to differences in expression, as all homologs were consistently expressed, as confirmed by Western blot (Figure 5C)". ScHsp104, TtHsp104 and GsHsp104 are among the highest rescue of thermotolerance. Contrast this with Figure 7—figure supplement 1, where the expression levels look much more consistent, with only CtHsp104 ∆N from the HSP104 promoter (panel F) showing lowered expression. The chimeras presented in Figure 8—figure supplement 1 also show some variation in levels. Here, the claim in the text is not as strong ("Generally, the chimeras expressed well in yeast"), however the figure caption states "Chimeric Hsp104s and proteotoxic substrates are consistently expressed. In addition, the authors discuss the variation in regard to the results (subsection “Hsp104 chimeras display reduced thermotolerance”: "But, expression levels of chimeras alone are insufficient to explain their thermotolerance phenotypes: even specific chimeras that reduce TDP-43 toxicity and express well from pHSP104 (e.g. Cr NTD:NBD2:MDScNBD2:CTD) fail to confer thermotolerance"). In summary, the differences in expression levels for these experiments should be better quantified if possible and should be discussed in more depth regarding possible effects on the data, especially given the reliance on ectopic expression throughout the manuscript.

---

## [Author Response]

Essential revisions:1) The authors need to more carefully interpret protein levels in their experiments. Throughout the text, the authors conclude that the levels of Hsp104, ɑ-synuclein, and TDP-43 do not change enough to affect their preferred interpretations. However, levels of the Hsp104 variants vary over at least an order of magnitude (Figure 1D, Figure 2C, Figure 3B, Figure 5C). The authors need to consider Hsp104 variant levels when arguing that one Hsp104 variant is more effective than another (e.g. subsection “Diverse Hsp104 homologs selectively suppress TDP-43 toxicity and aggregation in yeast”).Furthermore, some Hsp104 variants reduce expression of toxic proteins (i.e., A503S in Figure 1D, Figure 2D, Figure 3B). The text needs to properly acknowledge that alterations in toxic protein expression could be a mechanism by which some Hsp104 variants suppress toxicity (e.g. subsection “Hsp104 homologs prevent TDP-43 aggregation in human cells”, sentence beginning with, "We also monitored mClover3-TDP-43∆NLS…").

We thank the reviewers for raising this concern regarding how varying protein levels affect our interpretations. We have quantified three independent replicates of the blots shown in Figure 1D and Figure 2C. For the blot shown in Figure 1D, we find that neither Hsp104-FLAG nor TDP-43 levels vary significantly across strains. We have added bar charts showing this quantification to the manuscript as Figure 1E, F. Similarly, for the blot shown in Figure 2C, neither Hsp104-FLAG nor αSyn-GFP levels vary significantly across strains. We have added bar charts reflecting this quantification to the manuscript as Figure 2D, E.

Similarly, we also quantified three independent replicates of the blot shown in Figure 3B (which now includes samples taken 24hours and 48hours post-transfection) for Hsp104-FLAG and mClover3-TDP43∆NLS. We find that no sample (with the exception of no Hsp104 control) is significantly deficient in Hsp104 expression, although PeHsp104 expression is significantly increased at both day 1 and day 2 post-transfection. We also find that at day 1 post-transfection, none of the conditions have significantly different TDP-43 expression. Likewise, at day 2, none of the conditions shows diminished TDP-43 expression, although some conditions (*Sc*, *Pe*, and *Sr*) have increased TDP-43 expression. These findings suggest that Hsp104 homologs do not decrease TDP-43 foci formation by lowering TDP-43 expression. We have added these data to Figure 3C, D.

We have also quantified three independent replicates of the blot shown in Figure 5C (now Figure 5E) and added this quantification to the manuscript as Figure 5G. Here, we find that some Hsp104 homologs are expressed at significantly lower levels than ScHsp104-FLAG. However, we find that Hsp104-FLAG levels are a poor predictor of thermotolerance performance (R^2^=0.24; see Figure 5H).

Taken together, our data suggest that variation in protein levels (of either Hsp104 homologs or substrates) does not contribute significantly to the observed phenotypes.

2) Some experiments lack important controls. In Figure 4, the authors should also analyze worms lacking both ɑ-synuclein and Hsp104 as well as worms expressing the Hsp104 variants alone. This control will help the reader to understand: (a) whether the loss of DA neurons is ɑ-synuclein-specific, and (b) whether the Hsp104-dependent rescue of DA neurons is ɑ-synuclein-specific.

We have added data in Figure 4—figure supplement 1 showing that expression of either GFP alone or GFP plus Hsp104 homologs alone in dopaminergic neurons is not intrinsically toxic (i.e. they do not induce dopaminergic neuron degeneration when expressed alone in the absence of α-synuclein). Therefore, the dopaminergic neurodegeneration phenotype is α-synuclein-specific and Hsp104 homologs mitigate this α-synuclein-mediated neurodegeneration.

In Figure 5G,H, the authors should include un-aggregated luciferase controls so the reader can better understand the magnitude of luciferase refolding induced by these Hsp104 variants.

We now note in subsection “Differential suppression of proteotoxicity by Hsp104 homologs is not due to changes in disaggregase activity” that when combined with Hsp70 and Hsp40, ScHsp104 typically recovers ~10-30% of the unaggregated, native luciferase activity in these experiments (Cupo and Shorter, 2020; Glover and Lindquist, 1998; Shorter, 2011).

3) The result that these Hsp104 variants could suppress toxicity by a disaggregase-independent mechanism is a compelling finding that requires an additional control. Because many of these Hsp104 variants have not been biochemically characterized before this work, the authors should demonstrate that the canonical walker A, pore loop, and walker B mutations used in Figure 6 disrupt disaggregase activity in the expected manner. The authors could use the same luciferase assay from Figure 5G,5H with wild-type, walker A mutant, pore loop mutant, and walker B mutant variants of S. cerevisiae Hsp104 as well as Hsp104 from one of the more exotic species.

The inactivating effect of Walker A, pore-loop, and Walker B mutations on *S. cerevisiae* Hsp104 disaggregase activity has been exhaustively characterized in previous work, see for example: (DeSantis et al., 2012; Torrente et al., 2016). Furthermore, this loss of disaggregase activity is reflected by thermotolerance defects in vivo (DeSantis et al., 2012). Thus, defective thermotolerance is a reliable reporter of mutations that impair disaggregase activity, which is why we employed it in Figure 7—figure supplement 1I to show that Walker A, Walker B, and pore-loop mutants shown in Figure 6 (now Figure 7) are defective in disaggregase activity. To address the reviewers’ concern that we characterize the biochemical effects of these mutants directly, we purified Walker A, Walker B, and pore-loop mutants for the Hsp104 homolog from *Calcarisporiella thermophila* and characterized their in vitro disaggregase activity against luciferase aggregates. As expected, Walker A, Walker B, and pore-loop mutants were all severely impaired in disaggregase activity. We have added these data to Figure 7—figure supplement 1J.

4) Subsection "Hsp104 homologs prevent TDP-43 aggregation in human cells":a) Regarding the observation that ScHsp104WT also reduced the proportion of HEK293T cells with TDP-43 aggregation at 24-hours post transfection while it did not in yeast cells (Figure 3C vs. Figure 1G), the authors did not address if this is just a protein artefact in utilizing mClover3-TDP-43ΔNLS instead of TDP-43-GFP, or if it reflects differences in cellular environments or factors related to Hsp104 activity that are unaccounted for. It should be emphasized that the latter possibility requires attention as it raises the problem of relevance in using the yeast model to approximate the human cell, despite the authors' justification by referencing such practice by other groups. It is thus prudent that the authors repeat some of the TDP-43 experiments for ScHsp104WT in yeast cells using mClover3-TDP-43ΔNLS instead of TDP-43-GFP, for example.

We have shown that ScHsp104 has limited ability to antagonize TDP-43 aggregation in yeast (Figure 1G, I) (Jackrel et al., 2014; Jackrel and Shorter, 2014; Jackrel et al., 2015; Sweeny et al., 2015; Tariq et al., 2019; Tariq et al., 2018), in human cells after 48hours of expression (Figure 3E), and at the pure protein level (Figure 6D, E) (Jackrel et al., 2014). These data from multiple settings (using different TDP-43 constructs) make us confident that ScHsp104 does not effectively antagonize TDP-43 aggregation. The reviewers draw attention to the reduced proportion of cells with TDP-43 aggregates after 24hours when ScHsp104 is expressed (Figure 3E). This finding is interesting, but ultimately after longer incubation times, ScHsp104 is unable to antagonize TDP-43 aggregation in human cells (Figure 3E) just as it has limited ability to antagonize TDP-43 aggregation in yeast (Figure 1G, I) or at the pure protein level (Figure 6D, E). Thus, we suggest that there is no reason to doubt the relevance of the yeast model for human cells. Importantly, our yeast model (originally developed by Aaron Gitler and Susan Lindquist) of TDP-43 aggregation and toxicity has empowered the discovery of several key genetic modifiers of TDP-43 aggregation and toxicity, which have translated to fly, mouse, and neuronal models of ALS/FTD (Armakola et al., 2012; Becker et al., 2017; Elden et al., 2010; Kim et al., 2014). Indeed, this yeast model sparked the discovery of intermediate-length, polyglutamine expansions (~27-33 glutamines) in ataxin 2 as a common risk factor for ALS (Auburger et al., 2017; Elden et al., 2010; Lee et al., 2011a; Lee et al., 2011b; Yu et al., 2011). Based on these findings that originated from our yeast model, ataxin 2 antisense oligonucleotides are now in clinical development by Biogen/Ionis to treat ALS (ION541: https://ir.ionispharma.com/news-releases/news-release-details/ionis-third-novel-antisense-medicine-als-its-first-designed?fbclid=IwAR1VYI00Fkb4QZMVMxpKh517RXZI9dxFkkdUZj-7UR1dZeVnPjpifE-LcD4)

b) In tracking the TDP-43 foci formation at 24-hours versus 48-hours post transfection, the authors did not address as to why the magnitude of increase in % cells with foci from 24-hours to 48-hours displayed by no-Hsp104 (vector-only) control is drastically lower compared to all other cells co-expressing a Hsp104 homologue. One explanation is the unaccounted protein overexpression-induced stress / toxicity experienced by cells co-transfected with an Hsp104 homologue that is absent in the vector-only (no Hsp104) control, which will likely make the results more difficult to interpret.

In response to the reviewers' comment, we reanalyzed the raw data (which we now present in Figure 3—figure supplement 1) for Figure 3E to ensured that we gated the data to account for any shifts in expression. Hence, the graphs in Figure 3E have now been updated. We note that reviewers are not fully correct, the increase in % cells with TDP-43 aggregates from 24-hours to 48-hours displayed by no-Hsp104 (vector-only) control is not drastically lower compared to all other cells co-expressing a Hsp104 homologue. For example, for the no Hsp104 control the increase is ~3.7%, which is comparable to PeHsp104 and Hsp104^A503S^, which show increases of ~3.7% and ~4.7%, respectively. It is unclear why the increase is more pronounced from 24hours to 48hours for DPLA:DWB (~12.2%), ScHsp104 (~8.3%), and SrHsp104 (~6.5%), but we note that we have observed no overt toxicity from expressing TDP-43 or the Hsp104 variants in HEK293T cells over this short timeframe. This finding corroborates several prior studies, which found that expression of Hsp104 in mammalian cells is not toxic (Bao et al., 2002; Carmichael et al., 2000; Mosser et al., 2004; Yasuda et al., 2017). Indeed, Hsp104 expression can be neuroprotective in mammalian models of neurodegeneration (Lo Bianco et al., 2008; Vacher et al., 2005). In the present study, we have shown that expression of Hsp104 homologues in dopaminergic neurons of *C. elegans* is not toxic (Figure 4—figure supplement 1). Regardless, a key result from our experiments is that Hsp104^A503S^ and PeHsp104 consistently and significantly reduce TDP-43 aggregation in human cells (HEK293T cells) (Figure 3E).

c) Western blots for samples at 48-hr post transfection is missing.

We have added Western blots from samples at 48hours post-transfection, along with quantification of the blots for Hsp104 and mClover3-TDP-43∆NLS (at both 24hours and 48hours post-transfection) to Figure 3B. Please refer to our response to point 1 above for further discussion of these data.

5) Subsection "αSyn-selective Hsp104 homologs prevent dopaminergic neurodegeneration in C. elegans":a) Regarding the interpretation of lack of neuroprotection by CtHsp104 due to target promiscuity, this argument will be convincing once CtHsp104 is shown to also fail in providing neuroprotection against TDP-43 toxicity compared to Hsp104 homologues that are specific against TDP-43 using the C. elegans model (TDP-43 toxicity in C. elegans model: https://www.ncbi.nlm.nih.gov/pubmed/20530643). This data is currently missing from the manuscript.

We thank the reviewers for suggesting we consider how CtHsp104 behaves against TDP-43 in a worm model. However, it is not trivial for us to establish and characterize a new worm model, especially under the prevailing conditions of the pandemic. Nonetheless, to address this issue we have instead assessed CtHsp104 activity against TDP-43 aggregation in HEK293T cells using the same experimental protocol as in Figure 3. We find that CtHsp104 does not inhibit TDP-43 aggregation in HEK293T cells and in fact exacerbates aggregation. We have added these data to a new Figure 4—figure supplement 3. Our data show that CtHsp104, which is not substrate-specific, may have limited efficacy in animal model systems.

6) Subsection "Differential suppression of proteotoxicity by Hsp104 homologs is not due to changes in disaggregase activity":a) Regarding the authors' statement on potentiated Hsp104 disaggregases having elevated ATPase and disaggregase activities: Even though this has been shown previously, ScHsp104(A503S) or another potentiated mutant should be included in the biochemistry assays presented as a positive control, which will validate the experimental procedures employed and importantly, to provide conclusive evidence that the natural Hsp104 homologues are indeed "potentiated" differently compared to the artificially potentiated mutants.

We have added data comparing ATPase activity and luciferase disaggregation activity of WT ScHsp104 and ScHsp104^A503S^ (Figure 5—figure supplement 1) showing that ScHsp104^A503S^ is enhanced in ATPase and disaggregase activity compared to WT ScHsp104 as established previously (Jackrel et al., 2014). We note, however, that we have already provided conclusive genetic evidence that the natural Hsp104 homologs are potentiated differently than the artificially potentiated ScHsp104 variants (Figure 7). Thus, DWA, DWB, or DPLA mutations eliminate the ability of ScHsp104^A503S^ to rescue TDP-43 and a-synuclein toxicity in yeast (Figure 7B), whereas these mutations typically do not affect rescue of TDP-43 and a-synuclein toxicity by Hsp104 homologues in yeast (Figure 7C-J). Since DWA mutations eliminate ATP binding, DWB mutations eliminate ATP hydrolysis, and DPLA eliminates pore-loop binding to substrate, these results strongly suggest that the Hsp104 homologs rescue TDP-43 and a-synuclein toxicity by a different mechanism than ScHsp104^A503S^.

b) In Figure 5C, the authors should also include Western blots showing normal induction of Hsp26 or other heat shock-inducible proteins as confirmation that the yeast strains used are not compromised in heat shock response.

We thank the reviewers for this suggestion and have added an Hsp26 blot to Figure 5E (and quantification in Figure 5F). These data show that the yeast strains we use for thermotolerance induce Hsp26 after incubation at 37°C and thus are able to mount a robust heat shock response.

c) In Figure 5E, the authors should label the individual rescuer and non-rescuer Hsp104 homologues; the current figure is not very informative.

We thank the reviewers for bringing up this issue of clarity. We have amended Figure 5E (now Figure 5D) so that each data point is now color-coded by the Hsp104 homolog it represents.

d) Although the authors presented extensive biochemistry data showing the ATPase and disaggregase activity of Hsp104 homologues, there is no discussion in regard to the oligomerization state, which can be assessed using a variety of experimental techniques, including native PAGE, size exclusion chromatography, analytical ultracentrifugation, negative-stain electron microscopy, etc.

Hsp104 is only functional as an ATPase or disaggregase as a hexamer (Mackay et al., 2008; Parsell et al., 1994a; Parsell et al., 1994b; Schirmer et al., 1998; Schirmer et al., 2001). Thus, ATPase activity and disaggregase activity serve as strong evidence that the various Hsp104 homologues form functional hexamers (Figure 6A-C). We have previously shown that CtHsp104 forms hexamers under these conditions (Michalska et al., 2019). We have now added size-exclusion chromatography coupled to multi-angle light Scattering (SEC-MALS) data to the manuscript (in Figure 6—figure supplement 1) demonstrating that the other Hsp104 homologs we purified (i.e. MbHsp104, CrHsp104, TtHsp104, and TlHsp104) are likely hexameric.

In fact, examination of oligomeric state becomes particularly important in the authors' investigation of Hsp104 domains that contribute to suppression of aggregate toxicity, as chimeric proteins or isolated cognate domains may not form the expected hexameric form. Furthermore, data pertaining to the particular oligomeric state of the chimeras / cognate domains are expected to provide valuable information in establishing the molecular / structural basis of any retention of loss of toxicity suppression activity.

As noted in the manuscript, the isolated domains of Hsp104 and homologs are all anticipated to be monomeric based on previous studies (Hattendorf and Lindquist, 2002; Mogk et al., 2003). We have not purified the very many chimeric proteins assessed in these studies, as this would be a very large endeavor. However, prior studies indicate that chimeric proteins formed between Hsp104 and ClpB (*E. coli* Hsp104) can form hexamers that possess robust disaggregase activity in vitro and in vivo (Miot et al., 2011). Moreover, we have assessed the thermotolerance activity of all the chimeras (Figure 9—figure supplement 1), which provides an accurate proxy for disaggregase activity. In many cases, the chimeras display partial thermotolerance activity (e.g. all Tt-Sc chimeras), but in other cases they do not, which indicates impaired functionality with respect to thermotolerance. Possible causes of reduced functionality are dysregulated interdomain interactions, reduced hexamerization, reduced ATPase activity, or reduced ability to collaborate with yeast Hsp70. We suspect that the latter possibility applies to several chimeras (e.g. Sr-Sc chimeras and Ct-Sc chimeras) that lose thermotolerance activity when they do not possess the ScHsp104 middle domain, the primary site of Hsp70 interaction (Lee et al., 2013). In some cases, chimeras are robustly expressed, provide no thermotolerance but rescue TDP-43 toxicity (e.g. Cr^NTD:NBD1:MD^Sc^NBD2:CTD^). Importantly, these findings provide further evidence that rescue of TDP-43 toxicity by this chimera does not depend on disaggregase activity.

7) Subsection "ATPase-independent passive chaperone activity underlies suppression of TDP-43 and αSyn toxicity by Hsp104 homologs":a) Regarding the authors' claim of Hsp104 homologues in suppressing aggregate toxicity via passive chaperone activity: Although the authors have provided extensive, thorough yeast genetics data showing that neither the ATPase nor substrate translocation activity is required for aggregate toxicity suppression, to claim passive chaperone activity being the molecular basis requires additional experimental proof, such as comparing the refolding rate of thermally denatured reporter proteins (e.g. luciferase) in the presence or absence of Hsp104 homologues, their respective chimeric proteins or isolated cognate domains.

By passive chaperone activity we simply meant passive inhibition of aggregation. Our data (Figure 6D, E) suggest that Hsp104 homologs inhibit aggregation of TDP-43. However, we do not believe that the homologs are necessarily refolding TDP-43. We appreciate that “chaperone activity” may suggest assisting native folding, whereas we simply meant inhibition of aggregation. To avoid any confusion, we have removed the “passive chaperone” terminology from the manuscript and now simply refer to passive inhibition of aggregation.

8) In subsection “Diverse Hsp104 homologs selectively suppress TDP-43 toxicity and aggregation in yeast” and subsection “Distinct Hsp104 homologs selectively suppress αSyn toxicity and inclusion formation in yeast”, referring to the data In Figure 1—figure supplement 3: Suppression of TDP-43 or αSyn toxicity by select Hsp104 homologs is a substrate-specific effect (related to Figure 1 and Figure 2). The authors make the claim that "MbHsp104, SrHsp104, CrHsp104, PeHsp104, and GsHsp104, which all suppress TDP-43 toxicity, do not suppress αSyn nor FUS toxicity" (panel A) and that "TtHsp104 and TlHsp104 do not suppress TDP43 nor FUS toxicity" panel B and that "other Hsp104s investigated in this study do not suppress toxicity of TDP-43 (left), αSyn (middle), or FUS (right)" (panel C). While most of the data here support the authors conclusions, the wording "does not suppress" is a very strong statement. The data is not completely convincing. For example, in panel C, for the FUS overexpression (right panel) ChtHsp104 and StHsp140 seem to show some suppression of toxicity. Similarly, MtHsp104, StHsp104 and TaHsp104 for α-Syn maybe show some slight suppression. I don't think this changes the nature of the conclusion; however, the authors should be a little more cautious in their choice of wording and/or better discuss the data if they feel that there is compelling evidence that absolutely no suppression is shown.

We have altered the wording of the manuscript at these junctures from “do not suppress” to “have minimal effect on”.

9) In subsection “Diverse Hsp104 homologs selectively suppress TDP-43 toxicity and aggregation in yeast”: the authors state "Hsp104 homologs were consistently expressed,.… (Figure 1D)". The data on the Western Blot for Hsp104 (anti-flag in the blot in Figure 1D) is not convincing. The levels of protein look variable. Perhaps a quantification of replicates could be added to show whether Hsp104::flag protein levels are indeed consistent? Similarly, in Figure 2 panel C the Western blot could also be more convincing with replication and quantification. In Figure 5C, the Western Blot and the text that refer to it show variation in the levels of expression as well. For example, ScHsp104, TtHsp104 and GsHsp104 show much higher protein levels than StHsp104, MtHsp104 and ChtHsp104 yet the text claims: "These differences are not due to differences in expression, as all homologs were consistently expressed, as confirmed by Western blot (Figure 5C)". ScHsp104, TtHsp104 and GsHsp104 are among the highest rescue of thermotolerance. Contrast this with Figure 7—figure supplement 1, where the expression levels look much more consistent, with only CtHsp104 ∆N from the HSP104 promoter (panel F) showing lowered expression. The chimeras presented in Figure 8—figure supplement 1 also show some variation in levels. Here, the claim in the text is not as strong ("Generally, the chimeras expressed well in yeast"), however the figure caption states "Chimeric Hsp104s and proteotoxic substrates are consistently expressed. In addition, the authors discuss the variation in regard to the results (subsection “Hsp104 chimeras display reduced thermotolerance”: "But, expression levels of chimeras alone are insufficient to explain their thermotolerance phenotypes: even specific chimeras that reduce TDP-43 toxicity and express well from pHSP104 (e.g. Cr NTD:NBD2:MDScNBD2:CTD) fail to confer thermotolerance"). In summary, the differences in expression levels for these experiments should be better quantified if possible and should be discussed in more depth regarding possible effects on the data, especially given the reliance on ectopic expression throughout the manuscript.

We have quantified blots in Figure 1, Figure 2, and Figure 5 to be able to more substantively demonstrate that varying Hsp104 levels are not a significant contributor to our observed phenotypes (see also the response above to point 1). These findings suggest that the minor variations in chimera expression are also not a major factor.